# ON THE MODE-SEEKING PROPERTIES OF LANGEVIN DYNAMICS

## ABSTRACT

The Langevin Dynamics framework, which aims to generate samples from the score function of a probability distribution, is widely used for analyzing and interpreting score-based generative modeling. While the convergence behavior of Langevin Dynamics under unimodal distributions has been extensively studied in the literature, in practice the data distribution could consist of multiple distinct modes. In this work, we investigate Langevin Dynamics in producing samples from multimodal distributions and theoretically study its mode-seeking properties. We prove that under a variety of sub-Gaussian mixtures, Langevin Dynamics is unlikely to find all mixture components within a sub-exponential number of steps in the data dimension. To reduce the mode-seeking tendencies of Langevin Dynamics, we propose *Chained Langevin Dynamics*, which divides the data vector into patches of constant size and generates every patch sequentially conditioned on the previous patches. We perform a theoretical analysis of Chained Langevin Dynamics by reducing it to sampling from a constant-dimensional distribution. We present the results of several numerical experiments on synthetic and real image datasets, supporting our theoretical results on the iteration complexities of sample generation from mixture distributions using the chained and vanilla Langevin Dynamics.

## 1 INTRODUCTION

Langevin dynamics, a.k.a. Langevin Monte Carlo, is a well-known Markov Chain Monte Carlo (MCMC) sampling methodology that has been widely used to implement and interpret score-based generative modeling. It can produce samples from the (Stein) score function of a probability density, i.e., the gradient of the log probability density function with respect to data. It has been widely recognized that a pitfall of Langevin dynamics is its slow mixing rate (Wooddard et al., 2009; Raginsky et al., 2017; Lee et al., 2018). Specifically, Song & Ermon (2019) shows that under a multi-modal data distribution, the samples from Langevin dynamics may have an incorrect relative density across the modes. Based on this finding, Song & Ermon (2019) proposes *anneal Langevin dynamics*, which injects different levels of Gaussian noise into the data distribution and samples with Langevin dynamics on the perturbed distribution. While outputting the correct relative density across modes can be challenging for Langevin dynamics, a natural question is whether Langevin dynamics would be able to find all the modes of a multi-modal distribution.

In this work, we study this question by analyzing the mode-seeking properties of Langevin dynamics. The notion of mode-seekingness (Bishop, 2006; Ke et al., 2021; Li & Farnia, 2023; Li et al., 2024a) refers to the property that a generative model captures only a subset of the modes of a multi-modal distribution. We note that a similar problem, known as metastability, has been studied in the context of Langevin diffusion, a continuous-time version of Langevin dynamics described by stochastic differential equation (SDE) (Bovier et al., 2002; 2004; Gayrard et al., 2005). Specifically, Bovier et al. (2002) gave a sharp bound on the mean hitting time of Langevin diffusion and proved that it may require exponential (in the space dimensionality $d$) time for transition between modes. Regarding discrete Langevin dynamics, Lee et al. (2018) constructed a probability distribution whose density is close to a mixture of two well-separated isotropic Gaussians, and proved that Langevin dynamics could not find one of the two modes within an exponential number of steps. However, further exploration of the mode-seeking tendencies of Langevin dynamics and its variants for general distributions is still lacking in the literature.

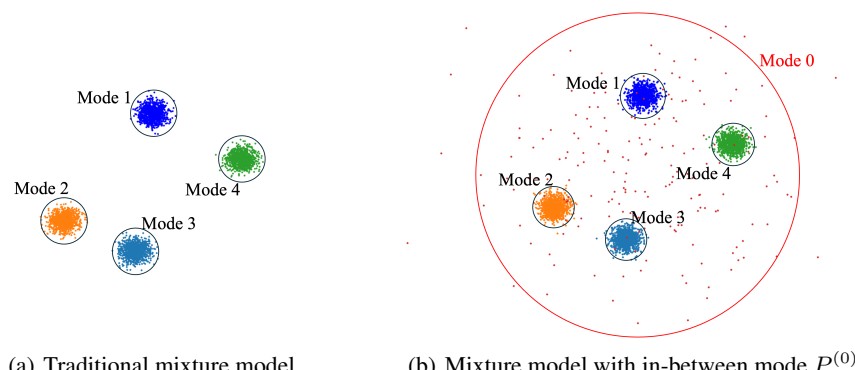

(a) Traditional mixture model      (b) Mixture model with in-between mode $P^{(0)}$

Figure 1: Traditional mixture models studied in the literature vs. our analyzed mixture distribution possessing the in-between mode $P^{(0)}$. $P^{(0)}$ is supposed to contain a minor probability mass, yet with a significantly higher variance than the other modes $P^{(1)}, \ldots, P^{(k)}$.

In this work, we study Langevin dynamics under multi-modal distributions in a slightly different setting from the standard theory literature on sampling. As illustrated in Figure 1, the existing theoretical literature commonly considers a mixture of well-separated modes with bounded variance. On the other hand, in our analysis, we consider a low-density high-variance mode (referred to as Mode 0 or $P^{(0)}$) surrounding the other modes and filling up the space between the modes. Note that Langevin dynamics relies on the score function (i.e., the gradient of log-pdf) to search for the modes. For a mixture model with no in-between modes (Figure 1.a), Langevin dynamics could carry information about the direction towards the closest mode, even if they are far from all the modes. However, assuming an in-between mode $P^{(0)}$ with high variance (Figure 1.b), the gradient information outside the support set of the low-variance modes will be dominated by $P^{(0)}$, despite the minor overall mass of $P^{(0)}$. As a result, one can expect that the dynamics would randomly explore a large volume in $\mathbb{R}^d$ until finding a low-variance mode, which can take a significant time.

To theoretically formulate and demonstrate the potential mode-seeking tendency of Langevin dynamics, we begin by analyzing the convergence for mixture distributions of Gaussian modes, under which Langevin dynamics could fail to visit all the mixture components within sub-exponential steps (in the data dimension). Subsequently, we generalize this result to mixture distributions of sub-Gaussian modes. This generalization extends our earlier result on Gaussian mixtures to a significantly larger family of mixture models, as the sub-Gaussian family includes any distribution over an $\ell_2$-norm-bounded support set.

To reduce Langevin dynamics' large iteration complexity shown under a high-dimensional input vector, we propose *Chained Langevin Dynamics (Chained-LD)*. Since Langevin dynamics could suffer from the curse of dimensionality, we decompose the sample $\mathbf{x} \in \mathbb{R}^d$ into $d/Q$ patches $\mathbf{x}^{(1)}, \cdots, \mathbf{x}^{(d/Q)}$, each of constant size $Q$, and sequentially generate every patch $\mathbf{x}^{(q)}$ for all $q \in [d/Q]$ statistically conditioned on previous patches, i.e., $P(\mathbf{x}^{(q)} \mid \mathbf{x}^{(0)}, \cdots \mathbf{x}^{(q-1)})$. The combination of all patches generated from the conditional distribution faithfully follows the probability density $P(\mathbf{x})$ due to the chain rule, while drawing samples from each patch requires less cost due to the reduced dimension. We also provide a theoretical analysis of Chained-LD by reducing the convergence of a $d$-dimensional sample to the convergence of each patch.

Finally, we present the results of several numerical experiments to validate our theoretical findings. For synthetic experiments, we consider moderately high-dimensional Gaussian mixture models, where the vanilla Langevin dynamics could not find all the components within a million steps, while Chained-LD could capture all the components with correct frequencies in $\mathcal{O}(10^4)$ steps. For experiments on real image datasets, we consider a mixture of two modes by using the original images from MNIST/Fashion-MNIST training dataset (black background and white digits/objects) as the first mode and constructing the second mode by i.i.d. flipping the images (white background and black digits/objects) with probability 0.5. Following from Song & Ermon (2019), we trained a Noise Conditional Score Network (NCSN) to estimate the score function. Our numerical results indicate

that Chained-LD was capable of finding both modes regardless of initialization. We summarize the contributions of this work as follows:

- Theoretically studying the mode-seeking properties of vanilla Langevin dynamics,
- Proposing Chained Langevin Dynamics (Chained-LD), which decomposes the sample into patches and sequentially generates each patch conditioned on previous patches,
- Providing a theoretical analysis of the convergence behavior of Chained-LD,
- Numerically comparing the mode-seeking properties of vanilla and chained Langevin dynamics.

**Notations:** We use $[n]$ to denote the set $\{1, 2, \cdots, n\}$. Also, in the paper, $\|\cdot\|$ refers to the $\ell_2$ norm. We use $\mathbf{0}_n$ and $\mathbf{1}_n$ to denote a 0-vector and 1-vector of length $n$. We use $\boldsymbol{I}_n$ to denote the identity matrix of size $n \times n$. In the text, TV stands for the total variation distance.

## 2 RELATED WORKS

**Langevin Dynamics:** The convergence guarantees for Langevin diffusion, a continuous version of Langevin dynamics, are classical results extensively studied in the literature (Bhattacharya, 1978; Roberts & Tweedie, 1996; Bakry & Émery, 1983; Bakry et al., 2008). Langevin dynamics, also known as Langevin Monte Carlo, is a discretization of Langevin diffusion typically modeled as a Markov Chain Monte Carlo (Welling & Teh, 2011). For unimodal distributions, e.g., the probability density function that is log-concave or satisfies log-Sobolev inequality, the convergence of Langevin dynamics is provably fast (Dalalyan, 2017; Durmus & Moulines, 2017; Vempala & Wibisono, 2019). However, for multimodal distributions, the non-asymptotic convergence analysis is much more challenging (Cheng et al., 2018). Raginsky et al. (2017) gave an upper bound on the convergence time of Langevin dynamics for arbitrary non-log-concave distributions with certain regularity assumptions, which, however, could be exponentially large without imposing more restrictive assumptions. Lee et al. (2018) studied the special case of a mixture of Gaussians of equal variance and provided an analysis of sampling from general non-log-concave distributions.

**Mode-Seekingness of Langevin Dynamics:** The investigation of the mode-seekingness of generative models starts with different generative adversarial network (GAN) (Goodfellow et al., 2014) model formulations and divergence measures, from both the practical (Goodfellow, 2016; Poole et al., 2016) and theoretical (Shannon et al., 2020; Li & Farnia, 2023; Li et al., 2024a) perspectives. In the context of Langevin dynamics, mode-seekingness is closely related to a lower bound on the transition time between two modes, e.g., two local maximums. Bovier et al. (2002; 2004); Gayrard et al. (2005) studied the mean hitting time of the continuous Langevin diffusion. Lee et al. (2018) proved the existence of a mixture of two Gaussian distributions whose covariance matrices differ by a constant factor, Langevin dynamics cannot find both modes in polynomial time.

**Score-based Generative Modeling:** A central task in unsupervised learning involves learning the underlying probability distribution of training data and efficiently generating new samples from the distribution. Since Song et al. (2020a) proposed sliced score matching which can train deep models to learn the score functions of implicit probability distributions on high-dimensional data, score-based generative modeling (SGM) has been going through a spurt of growth. Annealed Langevin dynamics (Song & Ermon, 2019) estimates the noise score of the probability density perturbed by Gaussian noise and utilizes Langevin dynamics to generate samples from a sequence of decreasing noise scales. Song & Ermon (2020) conducted an analysis of the effect of noise levels on the performance of annealed Langevin dynamics. Denoising diffusion probabilistic model (DDPM) (Ho et al., 2020) incorporates a step-by-step introduction of random noise into data, followed by learning to reverse this diffusion process in order to generate desired data samples from the noise. Song et al. (2020b) unified anneal Langevin dynamics and DDPM via a stochastic differential equation. A recent line of work focuses on the non-asymptotic convergence guarantees for SGM with an imperfect score estimation under various assumptions on the data distribution (Block et al., 2020; De Bortoli et al., 2021; Lee et al., 2022; Chen et al., 2023; Benton et al., 2023; Li et al., 2023; 2024b). Conforti et al. (2023) also investigated the KL convergence guarantees for score-based diffusion models. We highlight that a key difference between SGM and our theoretical analysis is that we assume the sampler has direct access to the true score function, whereas SGM typically focuses on learning the score function from training data.

## 3 PRELIMINARIES

### 3.1 LANGEVIN DYNAMICS

Generative modeling aims to produce samples such that their distribution is close to the underlying true distribution $P$. For a continuously differentiable probability density $P(\mathbf{x})$ on $\mathbb{R}^d$, its score function is defined as the gradient of the log probability density function (PDF) $\nabla_{\mathbf{x}} \log P(\mathbf{x})$. Langevin diffusion is a stochastic process defined by the stochastic differential equation (SDE)

$$\mathrm{d}\mathbf{x}_t = \nabla_{\mathbf{x}} \log P(\mathbf{x}_t)\,\mathrm{d}t + \sqrt{2}\,\mathrm{d}\mathbf{w}_t,$$

where $\mathbf{w}_t$ is the Wiener process on $\mathbb{R}^d$. Langevin dynamics, a discretization of the SDE for $T$ iterations, is applied to generate samples. Each iteration of Langevin dynamics is defined as

$$\mathbf{x}_t = \mathbf{x}_{t-1} + \frac{\delta_t}{2}\nabla_{\mathbf{x}} \log P(\mathbf{x}_{t-1}) + \sqrt{\delta_t}\boldsymbol{\epsilon}_t, \tag{1}$$

where $\delta_t$ is the step size and $\boldsymbol{\epsilon}_t \sim \mathcal{N}(\mathbf{0}_d, \boldsymbol{I}_d)$ is Gaussian noise. It has been widely recognized that Langevin diffusion could take exponential time to mix without additional assumptions on the probability density (Bovier et al., 2002; 2004; Gayrard et al., 2005; Raginsky et al., 2017; Lee et al., 2018). To combat the slow mixing, Song & Ermon (2019) proposed annealed Langevin dynamics by perturbing the probability density with Gaussian noise of variance $\sigma^2$, i.e.,

$$P_\sigma(\mathbf{x}) := \int P(\mathbf{z})\mathcal{N}(\mathbf{x} \mid \mathbf{z}, \sigma^2 \boldsymbol{I}_d)\,\mathrm{d}\mathbf{z}, \tag{2}$$

and running Langevin dynamics on the perturbed data distribution $P_{\sigma_t}(\mathbf{x})$ with gradually decreasing noise levels $\{\sigma_t\}_{t\in[T]}$, i.e.,

$$\mathbf{x}_t = \mathbf{x}_{t-1} + \frac{\delta_t}{2}\nabla_{\mathbf{x}} \log P_{\sigma_t}(\mathbf{x}_{t-1}) + \sqrt{\delta_t}\boldsymbol{\epsilon}_t, \tag{3}$$

where $\delta_t$ is the step size and $\boldsymbol{\epsilon}_t \sim \mathcal{N}(\mathbf{0}_d, \boldsymbol{I}_d)$ is Gaussian noise. When the noise level $\sigma$ is vanishingly small, the perturbed distribution is close to the true distribution, i.e., $P_\sigma(\mathbf{x}) \approx P(\mathbf{x})$.

**Remark 1.** *In our theoretical analysis, we assume the sampler has access to the true score function* $\nabla_{\mathbf{x}} \log P_\sigma(\mathbf{x})$. *In some realistic scenarios such as image datasets, since we do not have direct access to the (perturbed) score function, Song & Ermon (2019) proposed the Noise Conditional Score Network (NCSN)* $\mathbf{s}_{\boldsymbol{\theta}}(\mathbf{x}, \sigma)$ *to jointly estimate the scores of all perturbed data distributions, i.e.,* $\forall \sigma \in \{\sigma_t\}_{t\in[T]}$, $\mathbf{s}_{\boldsymbol{\theta}}(\mathbf{x}, \sigma) \approx \nabla_{\mathbf{x}} \log P_\sigma(\mathbf{x})$.

### 3.2 MULTI-MODAL DISTRIBUTIONS

Our work focuses on multi-modal distributions. We use $P = \sum_{i\in[k]} w_i P^{(i)}$ to represent a mixture of $k$ modes, where each mode $P^{(i)}$ is a probability density with frequency $w_i$ such that $w_i > 0$ for all $i \in [k]$ and $\sum_{i\in[k]} w_i = 1$. In our theoretical analysis, we consider Gaussian mixtures and sub-Gaussian mixtures, i.e., every component $P^{(i)}$ is a Gaussian or sub-Gaussian distribution. A probability distribution $p(\mathbf{z})$ of dimension $d$ is defined as a sub-Gaussian distribution with parameter $\nu^2$ if, given the mean vector $\boldsymbol{\mu} := \mathbb{E}_{\mathbf{z}\sim p}[\mathbf{z}]$, the moment generating function (MGF) of $p$ satisfies the following inequality for every vector $\boldsymbol{\alpha} \in \mathbb{R}^d$:

$$\mathbb{E}_{\mathbf{z}\sim p}\left[\exp\left(\boldsymbol{\alpha}^T(\mathbf{z} - \boldsymbol{\mu})\right)\right] \le \exp\left(\frac{\nu^2 \|\boldsymbol{\alpha}\|_2^2}{2}\right). \tag{4}$$

We remark that sub-Gaussian distributions include a wide variety of distributions such as Gaussian distributions and any distribution within a bounded $\ell_2$-norm distance from the mean $\boldsymbol{\mu}$.

## 4 THEORETICAL ANALYSIS OF THE MODE-SEEKING PROPERTIES OF LANGEVIN DYNAMICS

In this section, we theoretically investigate the mode-seeking properties of vanilla Langevin dynamics. We begin with analyzing Langevin dynamics in Gaussian mixtures, and further generalize the results to sub-Gaussian mixtures. We again highlight that in our theoretical analysis, we assume the sampler has access to the score function $\nabla_{\mathbf{x}} \log P(\mathbf{x})$ of the underlying distribution $P$.

## 4.1 LANGEVIN DYNAMICS IN GAUSSIAN MIXTURES

**Assumption 1.** *Consider a data distribution $P := \sum_{i=0}^{k} w_i P^{(i)}$ as a mixture of Gaussian distributions, where $1 \leq k = o(d)$ and $w_i > 0$ is a positive constant such that $\sum_{i=0}^{k} w_i = 1$. Suppose that $P^{(i)} = \mathcal{N}(\boldsymbol{\mu}_i, \nu_i^2 \boldsymbol{I}_d)$ is a Gaussian distribution over $\mathbb{R}^d$ for all $i \in \{0\} \cup [k]$ such that for all $i \in [k]$, $\nu_i < \nu_0$ and $\|\boldsymbol{\mu}_i - \boldsymbol{\mu}_0\|^2 \leq \frac{\nu_0^2 - \nu_i^2}{2} \left( \log \left( \frac{\nu_i^2}{\nu_0^2} \right) - \frac{\nu_i^2}{2\nu_0^2} + \frac{\nu_0^2}{2\nu_i^2} \right) d$. Denote $\nu_{\max} := \max_{i \in [k]} \nu_i$.*

To intuitively understand Assumption 1, we first note that the probability density $p(\mathbf{z})$ of a Gaussian distribution $\mathcal{N}(\boldsymbol{\mu}, \nu^2 \boldsymbol{I}_d)$ decays exponentially in terms of $\frac{\|\mathbf{z} - \boldsymbol{\mu}\|^2}{\nu^2}$. When a state $\mathbf{z}$ is sufficiently far from one mode $P^{(i)}$, the probability density of $P^{(i)}$ is dominated by the high-variance component $P^{(0)}$, which implies that the gradient information from $P^{(i)}$ will be masked by $P^{(0)}$. Hence, the dynamics can only visit the universal mode unless the stochastic noise miraculously leads it to the region of another mode. In addition, it can be verified that $\log \left( \frac{\nu_i^2}{\nu_0^2} \right) - \frac{\nu_i^2}{2\nu_0^2} + \frac{\nu_0^2}{2\nu_i^2}$ is a positive constant for $\nu_i < \nu_0$, thus the last requirement of Assumption 1 essentially represents $\|\boldsymbol{\mu}_i - \boldsymbol{\mu}_0\|^2 \leq \mathcal{O}(d)$. We formalize the intuition in Theorem 1 and defer the proof to Appendix A.1.

**Theorem 1.** *Consider a data distribution $P$ satisfying Assumption 1. We follow Langevin dynamics for $T = \exp(\mathcal{O}(d))$ steps. Suppose the sample is initialized in $P^{(0)}$, then with probability at least $1 - T \cdot \exp(-\Omega(d))$, we have $\|\mathbf{x}_t - \boldsymbol{\mu}_i\|^2 > \frac{\nu_0^2 + \nu_{\max}^2}{2} d$ for all $t \in \{0\} \cup [T]$ and $i \in [k]$.*

The constants in the notation $\Omega(d)$ are specified in Equations 6 and 7 in the Appendix. We note that $\|\mathbf{x}_t - \boldsymbol{\mu}_i\|^2 > \frac{\nu_0^2 + \nu_{\max}^2}{2} d$ is a strong notion of mode-seekingness, since the density of mode $P^{(i)} = \mathcal{N}(\boldsymbol{\mu}_i, \nu_i^2 \boldsymbol{I}_d)$ concentrates around the $\ell_2$-norm ball $\left\{ \mathbf{z} : \|\mathbf{z} - \boldsymbol{\mu}_i\|^2 \leq \nu_i^2 d \right\}$. This notion can be translated into a lower bound in terms of other distance measures, e.g., total variation distance in the following corollary, whose proof is deferred to Appendix A.2.

**Corollary 1.** *Under the same assumptions as in Theorem 1, for all time steps $t \in \{0\} \cup [T]$, the distribution $\hat{P}_t$ of the generated sample $\mathbf{x}_t$ by Langevin dynamics at time $t$ satisfies*

$$TV(\hat{P}_t, P) \geq (1 - w_0) \left( 1 - \frac{T}{\exp(\Omega(d))} \right).$$

## 4.2 LANGEVIN DYNAMICS IN SUB-GAUSSIAN MIXTURES

We further generalize our results to sub-Gaussian mixtures. We impose the following assumptions on the mixture. It is worth noting that Assumptions 2.i.-iii. automatically hold for Gaussian mixtures such that $P^{(i)} = \mathcal{N}(\boldsymbol{\mu}_i, \nu_i^2 I)$, and Assumptions 2.iv. and v. are specific assumptions on the mean and variance of $P^{(i)}$, similar to Assumption 1.

**Assumption 2.** *Consider a data distribution $P := \sum_{i=0}^{k} w_i P^{(i)}$ as a mixture of sub-Gaussian distributions, where $1 \leq k = o(d)$ and $w_i > 0$ is a positive constant such that $\sum_{i=0}^{k} w_i = 1$. Suppose that $P^{(0)} = \mathcal{N}(\boldsymbol{\mu}_0, \nu_0^2 \boldsymbol{I}_d)$ is Gaussian and for all $i \in [k]$, $P^{(i)}$ satisfies*

 *i. $P^{(i)}$ is a sub-Gaussian distribution of mean $\boldsymbol{\mu}_i$ with parameter $\nu_i^2$,*

 *ii. $P^{(i)}$ is differentiable and $\nabla P^{(i)}(\boldsymbol{\mu}_i) = \mathbf{0}_d$,*

 *iii. the score function of $P^{(i)}$ is $L_i$-Lipschitz such that $L_i \leq \frac{c_L}{\nu_i^2}$ for some constant $c_L > 0$,*

 *iv. $\nu_0^2 > \max \left\{ 1, \frac{4(c_L^2 + c_\nu c_L)}{c_\nu (1 - c_\nu)} \right\} \frac{\nu_{\max}^2}{1 - c_\nu}$ for constant $c_\nu \in (0, 1)$, where $\nu_{\max} := \max_{i \in [k]} \nu_i$,*

 *v. $\|\boldsymbol{\mu}_i - \boldsymbol{\mu}_0\|^2 \leq \frac{(1 - c_\nu)\nu_0^2 - \nu_i^2}{2(1 - c_\nu)} \left( \log \frac{c_\nu \nu_i^2}{(c_L^2 + c_\nu c_L)\nu_0^2} - \frac{\nu_i^2}{2(1 - c_\nu)\nu_0^2} + \frac{(1 - c_\nu)\nu_0^2}{2\nu_i^2} \right) d$.*

The feasibility of Assumption 2.v. is validated by Lemma 9 in Appendix A.3. With Assumption 2, we show the mode-seeking tendency of Langevin dynamics under sub-Gaussian distributions in Theorem 2 and defer the proof to Appendix A.3.

---

**Algorithm 1** Chained Langevin Dynamics (Chained-LD)

---

**Require:** Patch size $Q$, dimension $d$, conditional score function $\nabla \log P_{\sigma_t}$, number of iterations $T$, noise levels $\{\sigma_t\}_{t \in [TQ/d]}$, step size $\{\delta_t\}_{t \in [TQ/d]}$.

1: Initialize $\mathbf{x}_0$, and divide $\mathbf{x}_0$ into $d/Q$ patches $\mathbf{x}_0^{(1)}, \cdots \mathbf{x}_0^{(d/Q)}$ of equal size $Q$

2: **for** $q \leftarrow 1$ to $d/Q$ **do**

3:     **for** $t \leftarrow 1$ to $TQ/d$ **do**

4:         $\mathbf{x}_t^{(q)} \leftarrow \mathbf{x}_{t-1}^{(q)} + \frac{\delta_t}{2} \nabla \log P_{\sigma_t} \left( \mathbf{x}_{t-1}^{(q)} \mid \mathbf{x}^{(1)}, \cdots, \mathbf{x}^{(q-1)} \right) + \sqrt{\delta_t} \boldsymbol{\epsilon}_t$, where $\boldsymbol{\epsilon}_t \sim \mathcal{N}(\mathbf{0}_Q, \boldsymbol{I}_Q)$

5:     **end for**

6:     $\mathbf{x}^{(q)} \leftarrow \mathbf{x}_{TQ/d}^{(q)}$

7: **end for**

8: **return x**

---

**Theorem 2.** *Consider a data distribution $P$ satisfying Assumption 2. We follow Langevin dynamics for $T = \exp(\mathcal{O}(d))$ steps. Suppose the sample is initialized in $P^{(0)}$, then with probability at least $1 - T \cdot \exp(-\mathcal{O}(d))$, we have $\|\mathbf{x}_t - \boldsymbol{\mu}_i\|^2 > \left( \frac{\nu_0^2}{2} + \frac{\nu_{\max}^2}{2(1-c_\nu)} \right) d$ for all $t \in \{0\} \cup [T]$ and $i \in [k]$.*

We remark that an implication of Theorem 2 is the potential difficulty of transition between low-variance modes for Langevin dynamics. For instance, suppose $P^{(1)}, \cdots, P^{(k)}$ have bounded support sets with small radius and sufficiently distant means. If the sample is initialized in a low-variance mode $P^{(m)}$ (for $m \in [k]$), either it stays in $P^{(m)}$ and cannot capture other modes, or it escapes $P^{(m)}$ (due to the random noise $\boldsymbol{\epsilon}$) and is expected to need to explore the whole space until finding the support sets of the other bounded modes.

Furthermore, in Appendix B we extend our theoretical analysis to annealed Langevin dynamics *with bounded noise levels*, indicating the effect of annealing noise levels on the mode-seeking tendencies of Langevin dynamics. Aligning with the empirical analysis in (Song & Ermon, 2020), we show that bounded noise levels will have a limited impact on Langevin dynamics since they exhibit similar mode-seeking tendencies. On the other hand, as suggested by Song & Ermon (2020), annealed Langevin dynamics with a significantly larger initial noise level could capture more modes, which is also confirmed by our numerical results in Section 6.

## 5   Chained Langevin Dynamics

To reduce the mode-seeking tendencies of vanilla Langevin dynamics, we propose Chained Langevin Dynamics (Chained-LD) in Algorithm 1. While vanilla Langevin dynamics apply gradient updates to all coordinates of the sample in every step, we decompose the sample into patches of constant size and generate each patch sequentially to alleviate the exponential dependency on the dimensionality. More precisely, we divide a sample $\mathbf{x}$ into $d/Q$ patches $\mathbf{x}^{(1)}, \cdots \mathbf{x}^{(d/Q)}$ of some constant size $Q$, and apply Langevin dynamics to sample each patch $\mathbf{x}^{(q)}$ (for $q \in [d/Q]$) from the conditional distribution $P(\mathbf{x}^{(q)} \mid \mathbf{x}^{(1)}, \cdots \mathbf{x}^{(q-1)})$. Intuitively, vanilla Langevin dynamics needs to explore the entire space (of volume exponentially large in $d$) to find the missing modes, while Chained-LD could significantly lower the volume by dimensionality reduction.

In practice, we can also apply annealed Langevin dynamics (Song & Ermon, 2019) to facilitate the sampling of each patch, by perturbing it with a series of noise levels $\{\sigma_t\}_{t \in [TQ/d]}$. Specifically, we refer *chained vanilla Langevin dynamics (Chained-VLD)* to Algorithm 1 without noise injection (i.e., $\sigma_t = 0$ for all $t \in [TQ/d]$), and *chained annealed Langevin dynamics (Chained-ALD)* otherwise. Ideally, if a sampler perfectly generates every patch, combining all patches gives a sample from the original distribution due to the chain rule $P(\mathbf{x}) = \prod_{q \in [d/Q]} P(\mathbf{x}^{(q)} \mid \mathbf{x}^{(1)}, \cdots \mathbf{x}^{(q-1)})$. In Proposition 1 we give a linear reduction from producing samples of dimension $d$ using Chained-LD to learning the distribution of a $Q$-dimensional variable for constant $Q$. The proof of Proposition 1 is deferred to Appendix C.

**Proposition 1.** *Consider a sampler algorithm taking the first $q - 1$ patches $\mathbf{x}^{(1)}, \cdots, \mathbf{x}^{(q-1)}$ as input and outputing a sample of the next patch $\mathbf{x}^{(q)}$ with probability $\hat{P}\left(\mathbf{x}^{(q)} \mid \mathbf{x}^{(1)}, \cdots, \mathbf{x}^{(q-1)}\right)$ for all $q \in [d/Q]$. Suppose that for every $q \in [d/Q]$ and any given previous patches $\mathbf{x}^{(1)}, \cdots, \mathbf{x}^{(q-1)}$, the sampler algorithm can achieve*

$$TV\left(\hat{P}\left(\mathbf{x}^{(q)} \mid \mathbf{x}^{(1)}, \cdots, \mathbf{x}^{(q-1)}\right), P\left(\mathbf{x}^{(q)} \mid \mathbf{x}^{(1)}, \cdots, \mathbf{x}^{(q-1)}\right)\right) \leq \varepsilon \cdot \frac{Q}{d}$$

*in $\tau(\varepsilon/d)$ iterations for some $\varepsilon > 0$. Then, equipped with the sampler algorithm, the Chained-LD algorithm in $\frac{d}{Q} \cdot \tau(\varepsilon/d)$ iterations can achieve*

$$TV\left(\hat{P}(\mathbf{x}), P(\mathbf{x})\right) \leq \varepsilon.$$

With additional assumptions on the target distribution $P$, we can obtain upper bounds on the iteration complexity of Chained-LD. If the conditional distribution $p := P\left(\mathbf{x}^{(q)} \mid \mathbf{x}^{(1)}, \cdots, \mathbf{x}^{(q-1)}\right)$ for all $q \in [d/Q]$ satisfies the assumptions specified in Appendix A of Ma et al. (2019), i.e., $\log p$ is $L$-Lipschitz smooth for all $\mathbf{x}^{(q)}$ and m-strongly concave for $\mathbf{x}^{(q)}$ outside an $\ell_2$-norm ball of radius $R$ (i.e., $\left\|\mathbf{x}^{(q)}\right\| > R$), by Theorem 1 of Ma et al. (2019) we obtain $\tau(\varepsilon/d) = \mathcal{O}\left(\exp(32LR^2)\frac{L^2}{m^2} \cdot \frac{R}{(\varepsilon/d)^2} \log \frac{R}{(\varepsilon/d)^2}\right)$. In common settings, the radius $R$ in $Q$–dimensional space scales as $R = \Theta(\sqrt{Q})$. Therefore, for constants $L$ and $m$, the iteration complexity of Chained-LD is a polynomial $\mathcal{O}\left(\exp(\mathcal{O}(LQ))\frac{d^3}{\varepsilon^2\sqrt{Q}} \log \frac{d^2\sqrt{Q}}{\varepsilon^2}\right)$ in $d$, which takes its minimum value $\mathcal{O}\left(\frac{d^3}{\varepsilon^2} \log \frac{d^2}{\varepsilon^2}\right)$ when the patch size $Q$ is a constant not growing with $d$. For comparison, the iteration complexity of vanilla Langevin dynamics could scale exponentially in $d$ when $R = \Theta(\sqrt{d})$ in common settings.

**Remark 2.** *We highlight that Chained-LD is a sampling algorithm. In Algorithm 1, we assume the sampler has direct access to the conditional score function. Also, the conditional densities used in Chained-LD do not require any extra information compared to the target distribution assumed in vanilla LD, since vanilla LD has access to the joint distribution and Chained-LD has access to conditional distributions that, based on chain rule, have the same information as the joint distribution.*

**Remark 3.** *We note that similar combinations of autoregressive models and denoising diffusion models have been studied in the generative modeling literature, in the context of text generation (Hoogeboom et al., 2022; Wu et al., 2023) and time series forecasting (Rasul et al., 2021). At a high level, the generative modeling literature focuses on the training and implementation of autoregressive diffusion models in time-dependent scenarios, while this work focuses on the theoretical guarantees of Chained Langevin dynamics and its comparison with vanilla Langevin dynamics, motivated by their mode-seeking properties under sub-Gaussian mixture distribution.*

# 6 NUMERICAL RESULTS

In this section, we empirically evaluated the mode-seeking tendencies of vanilla and chained Langevin dynamics. We performed numerical experiments on synthetic Gaussian mixture models and real image datasets including MNIST (LeCun, 1998) and Fashion-MNIST (Xiao et al., 2017). Details on the experiment setup are deferred to Appendix D.

**Synthetic Gaussian mixture model:** We consider the data distribution $P$ as a mixture of three Gaussian components in dimension $d = 100$, where mode 0 defined as $P^{(0)} = \mathcal{N}(\mathbf{0}_d, 3\boldsymbol{I}_d)$ is the in-between mode with high variance, and mode 1 and mode 2 are respectively defined as $P^{(1)} = \mathcal{N}(\mathbf{1}_d, \boldsymbol{I}_d)$ and $P^{(2)} = \mathcal{N}(-\mathbf{1}_d, \boldsymbol{I}_d)$. The frequencies of the three modes are 0.2, 0.4 and 0.4, i.e.,

$$P = 0.2P^{(0)} + 0.4P^{(1)} + 0.4P^{(2)} = 0.2\mathcal{N}(\mathbf{0}_d, 3\boldsymbol{I}_d) + 0.4\mathcal{N}(\mathbf{1}_d, \boldsymbol{I}_d) + 0.4\mathcal{N}(-\mathbf{1}_d, \boldsymbol{I}_d).$$

In the synthetic experiments, we give the samplers access to the true score function calculated from the target distribution. As shown in Figure 2, vanilla Langevin dynamics (VLD) cannot find mode 1 or 2 within $10^6$ iterations if the sample is initialized in mode 0, while chained vanilla Langevin dynamics (Chained-VLD) with patch size $Q = 10$ can find the other two modes in 1000 steps and correctly recover their frequencies as gradually increasing the number of iterations. When the sample is initialized in mode 1, as shown in Figure 5 in Appendix D.1, VLD is also likely to

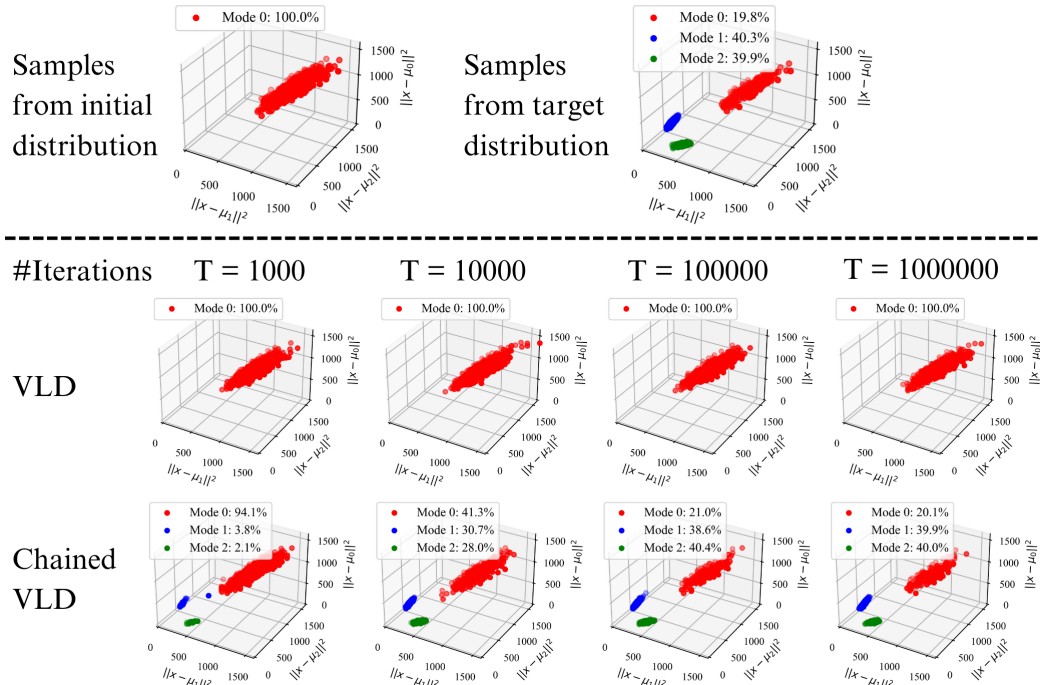

Figure 2: Samples from a mixture of three Gaussian modes generated by vanilla Langevin dynamics (VLD) and chained vanilla Langevin dynamics (Chained-VLD) with patch size $Q = 10$. Three axes are $\ell_2$ distance from samples to the mean of the three modes. The samples are initialized in mode 0.

be trapped by the high-variance mode 0 and cannot find mode 2, while Chained-VLD is capable of finding all modes. Additional experiments on samples initialized in mode 2 are presented in Appendix D.1, which also verify the mode-seeking tendencies of vanilla Langevin dynamics. We also investigated the effect of different choices of patch size $Q$ on the performance of Chained-LD. As shown in Figures 7, 8, and 9 in Appendix D.1, the convergence of Chained-LD are insensitive to moderate values of constant $Q \in \{1, 4, 10\}$; for large $Q = 20$, it takes more steps to find the other modes; while for overly large $Q = 50$, Chained-LD has mode-seeking tendencies similar to LD.

**Image datasets:** We also perform experiments on generating samples from image datasets by chained annealed Langevin dynamics (Chained-ALD). We construct the distribution as a mixture of two modes by using the original images from MNIST/Fashion-MNIST training dataset (black background and white digits/objects) as the first mode and constructing the second mode by i.i.d. randomly flipping an image (white background and black digits/objects) with probability 0.5.

Since the target distribution of image datasets is unknown, following from Song & Ermon (2019), we train an estimator to approximate the score function from training samples. More details are deferred to Appendix D.2. We use Recurrent Neural Network (RNN) architectures to estimate the perturbed conditional score function $\nabla_{\mathbf{x}^{(q)}} \log P_{\sigma_t}(\mathbf{x}^{(q)} \mid \mathbf{x}^{(1)}, \cdots \mathbf{x}^{(q-1)})$ for Chained-ALD. We note that for a sequence of inputs, the output of RNN from the previous step is fed as input to the current step. Therefore, in the scenario of chained Langevin dynamics, the hidden state of RNN contains information about the previous patches and allows the network to estimate the conditional score function of the next patch. More implementation details are deferred to Appendix D.3.

We numerically compare the performance of annealed Langevin dynamics (ALD) and Chained-ALD with different noise levels. The experimental results are shown in Figures 3 and 4. For ALD with bounded noise levels (i.e., the maximum noise $\sigma_{\max} = 1$), we observe that it tends to generate the samples from the same mode as initialization, aligning with our theoretical analysis in Theorem 4 in Appendix B. Then, if we apply significantly larger noise levels (i.e., the maximum noise $\sigma_{\max} = 50$ as suggested by Technique 1 in Song & Ermon (2020)), ALD could generate samples from both modes. On the other hand, Chained-ALD, even with bounded noise levels (i.e., $\sigma_{\max} = 1$), is capable of finding both modes. Further experiments are deferred to Appendix D.3.

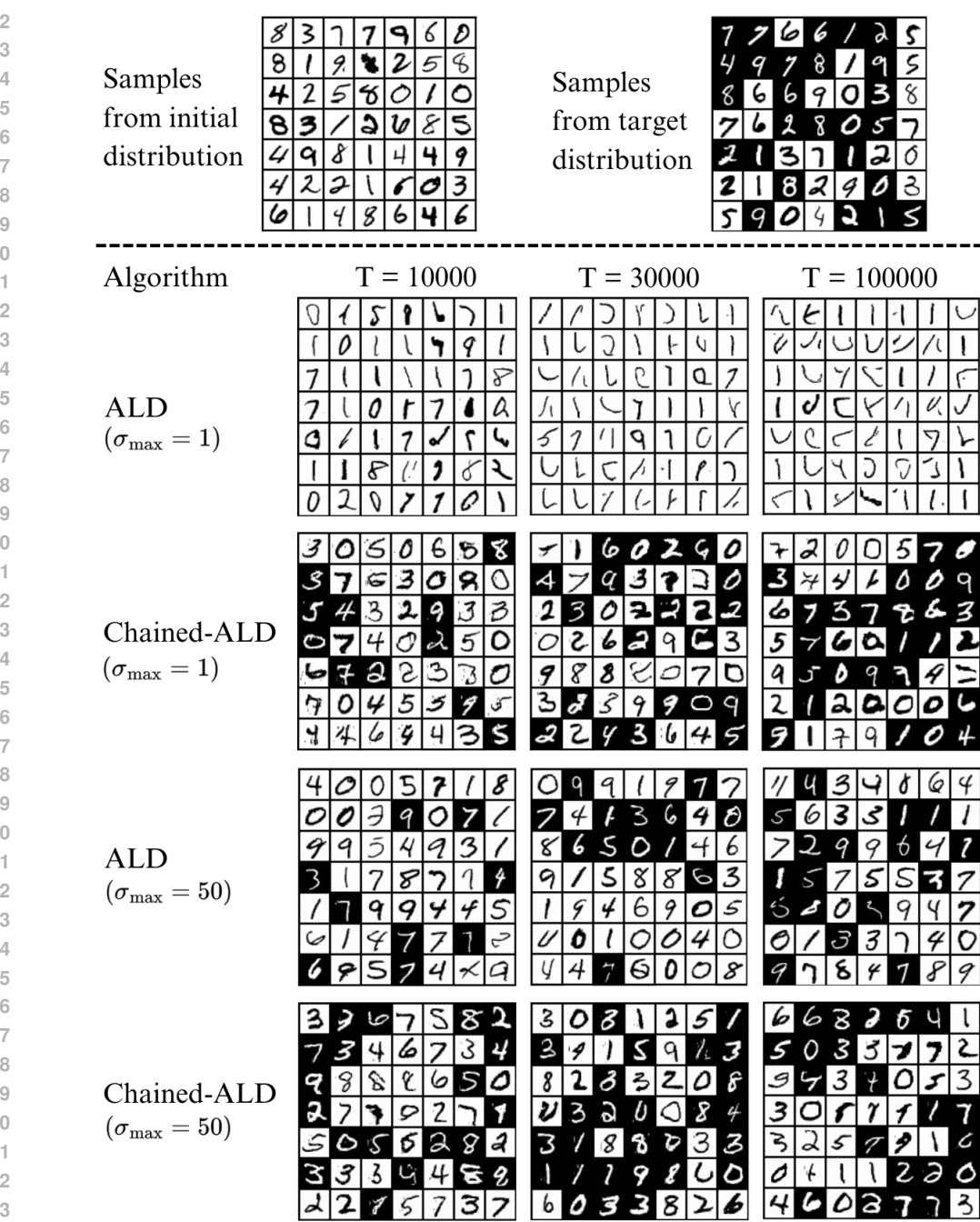

Figure 3: Samples from a mixture distribution of the original and flipped images from the MNIST dataset generated by annealed Langevin dynamics (ALD) and chained annealed Langevin dynamics (Chained-ALD) with patch size $Q = 14$ for different numbers of iterations. The maximum noise level $\sigma_{\max}$ is set to be 1 or 50. The samples are initialized as flipped images from MNIST.

## 7 CONCLUSION

In this work, we theoretically and numerically studied the mode-seeking properties of Langevin dynamics sampling methods under a multi-modal distribution. We characterized Gaussian and sub-Gaussian mixture models under which vanilla Langevin dynamics are unlikely to find all the components within a sub-exponential number of iterations. To reduce the mode-seeking tendency of vanilla

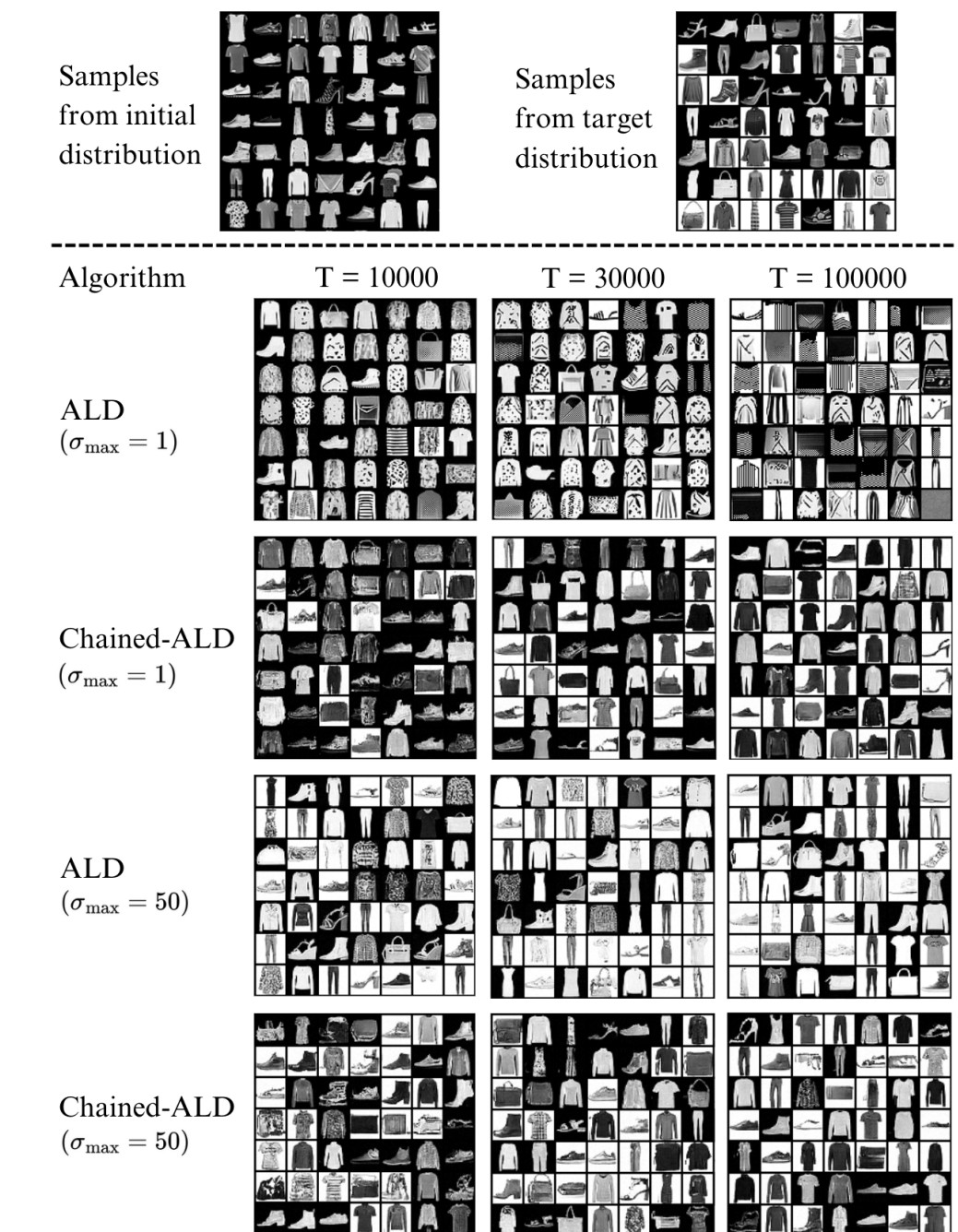

Figure 4: Samples from a mixture distribution of the original and flipped images from the Fashion-MNIST dataset generated by annealed Langevin dynamics (ALD) and chained annealed Langevin dynamics (Chained-ALD) with patch size $Q = 14$ for different numbers of iterations. The maximum noise level $\sigma_{\max}$ is set to be 1 or 50. The initialization is original images from Fashion-MNIST.

Langevin dynamics, we proposed Chained Langevin Dynamics (Chained-LD) and analyzed its convergence behavior. Studying the connections between Chained-LD and denoising diffusion models will be an interesting topic for future exploration. Our RNN-based implementation of Chained-LD is currently limited to image data generation tasks. An interesting future direction is to extend the application of Chained-LD to other domains such as audio and text data. Another future direction could be to study the convergence of Chained-LD under an imperfect score estimation.

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

## A   THEORETICAL ANALYSIS ON THE MODE-SEEKING TENDENCY OF LANGEVIN DYNAMICS

We begin by introducing some well-established lemmas used in our proof. We use the following lemma on the tail bound for multivariate Gaussian random variables.

**Lemma 1** (Lemma 1, Laurent & Massart (2000)). *Suppose that a random variable* $\mathbf{z} \sim \mathcal{N}(\mathbf{0}_d, \mathbf{I}_d)$. *Then for any* $\lambda > 0$,

$$\mathbb{P}\left(\|\mathbf{z}\|^2 \geq d + 2\sqrt{d\lambda} + 2\lambda\right) \leq \exp(-\lambda),$$

$$\mathbb{P}\left(\|\mathbf{z}\|^2 \leq d - 2\sqrt{d\lambda}\right) \leq \exp(-\lambda).$$

We also use a tail bound for one-dimensional Gaussian random variables and provide the proof here for completeness.

**Lemma 2.** *Suppose a random variable* $Z \sim \mathcal{N}(0,1)$. *Then for any* $t > 0$,

$$\mathbb{P}(Z \geq t) = \mathbb{P}(Z \leq -t) \leq \frac{\exp(-t^2/2)}{\sqrt{2\pi}t}.$$

*Proof of Lemma 2.* Since $\frac{z}{t} \geq 1$ for all $z \in [t, \infty)$, we have

$$\mathbb{P}(Z \geq t) = \frac{1}{\sqrt{2\pi}} \int_t^\infty \exp\left(-\frac{z^2}{2}\right) \mathrm{d}z \leq \frac{1}{\sqrt{2\pi}} \int_t^\infty \frac{z}{t} \exp\left(-\frac{z^2}{2}\right) \mathrm{d}z = \frac{\exp(-t^2/2)}{\sqrt{2\pi}t}.$$

Since the Gaussian distribution is symmetric, we have $\mathbb{P}(Z \geq t) = \mathbb{P}(Z \leq -t)$. Hence we obtain the desired bound. $\qquad\square$

### A.1   PROOF OF THEOREM 1: LANGEVIN DYNAMICS UNDER GAUSSIAN MIXTURES

Without loss of generality, we assume that $\boldsymbol{\mu}_0 = \mathbf{0}_d$ for simplicity. Let $r$ and $n$ respectively denote the rank and nullity of the vector space $\{\boldsymbol{\mu}_i\}_{i\in[k]}$, then we have $r + n = d$ and $0 \leq r \leq k = o(d)$. Denote $\mathbf{R} \in \mathbb{R}^{d\times r}$ an orthonormal basis of the vector space $\{\boldsymbol{\mu}_i\}_{i\in[k]}$, and denote $\mathbf{N} \in \mathbb{R}^{d\times n}$ an orthonormal basis of the null space of $\{\boldsymbol{\mu}_i\}_{i\in[k]}$. Now consider decomposing the sample $\mathbf{x}_t$ by

$$\mathbf{r}_t := \mathbf{R}^T\mathbf{x}_t, \text{ and } \mathbf{n}_t := \mathbf{N}^T\mathbf{x}_t,$$

where $\mathbf{r}_t \in \mathbb{R}^r$, $\mathbf{n}_t \in \mathbb{R}^n$. Then we have

$$\mathbf{x}_t = \mathbf{R}\mathbf{r}_t + \mathbf{N}\mathbf{n}_t.$$

Similarly, we decompose the noise $\boldsymbol{\epsilon}_t$ into

$$\boldsymbol{\epsilon}_t^{(\mathbf{r})} := \mathbf{R}^T\boldsymbol{\epsilon}_t, \text{ and } \boldsymbol{\epsilon}_t^{(\mathbf{n})} := \mathbf{N}^T\boldsymbol{\epsilon}_t,$$

where $\boldsymbol{\epsilon}_t^{(\mathbf{r})} \in \mathbb{R}^r$, $\boldsymbol{\epsilon}_t^{(\mathbf{n})} \in \mathbb{R}^n$. Then we have

$$\boldsymbol{\epsilon}_t = \mathbf{R}\boldsymbol{\epsilon}_t^{(\mathbf{r})} + \mathbf{N}\boldsymbol{\epsilon}_t^{(\mathbf{n})}.$$

Since a linear combination of a Gaussian random variable still follows Gaussian distribution, by $\boldsymbol{\epsilon}_t \sim \mathcal{N}(\mathbf{0}_d, \mathbf{I}_d)$, $\mathbf{R}^T\mathbf{R} = \mathbf{I}_r$, and $\mathbf{N}^T\mathbf{N} = \mathbf{I}_n$ we obtain

$$\boldsymbol{\epsilon}_t^{(\mathbf{r})} \sim \mathcal{N}(\mathbf{0}_r, \mathbf{I}_r), \text{ and } \boldsymbol{\epsilon}_t^{(\mathbf{n})} \sim \mathcal{N}(\mathbf{0}_n, \mathbf{I}_n).$$

By the definition of Langevin dynamics in equation 1, $\mathbf{n}_t$ follow from the update rule:

$$\mathbf{n}_t = \mathbf{n}_{t-1} + \frac{\delta_t}{2}\mathbf{N}^T\nabla_{\mathbf{x}}\log P(\mathbf{x}_{t-1}) + \sqrt{\delta_t}\boldsymbol{\epsilon}_t^{(\mathbf{n})}. \tag{5}$$

It is worth noting that since $\mathbf{N}^T\boldsymbol{\mu}_i = \mathbf{0}_n$. To show $\|\mathbf{x}_t - \boldsymbol{\mu}_i\|^2 > \frac{\nu_0^2 + \nu_{\max}^2}{2}d$, it suffices to prove

$$\|\mathbf{n}_t\|^2 > \frac{\nu_0^2 + \nu_{\max}^2}{2}d.$$

We start by proving that the initialization of the state $\mathbf{x}_0$ has a large norm on the null space with high probability in the following proposition. Throughout the proof, the notation $\Omega(d)$ refers to $\Omega(d) \geq cd$, for the constant $c$ defined as

$$c = \min \left\{ \frac{1}{2} \left( \frac{\nu_0^2 - \nu_{\max}^2}{8\nu_0^2} \right)^2, \frac{1}{8} \left( \log \left( \frac{\nu_{\max}^2}{\nu_0^2} \right) - \frac{\nu_{\max}^2}{2\nu_0^2} + \frac{\nu_0^2}{2\nu_{\max}^2} \right), \frac{1}{32}, \frac{(\nu_0^2 - \nu_{\max}^2)^2}{32\nu_0^2(\nu_0^2 + \nu_{\max}^2)} \right\}, \tag{6}$$

when $d$ satisfies

$$d \geq \max \left\{ 8 \left( \log \left( \frac{\nu_{\max}^2}{\nu_0^2} \right) - \frac{\nu_{\max}^2}{2\nu_0^2} + \frac{\nu_0^2}{2\nu_{\max}^2} \right)^{-1} \log \left( \frac{3\nu_0^3}{w_0 \min_{i \in [k]} \nu_i^2} \right), \frac{8\nu_0^2(3\nu_0^2 + \nu_{\max}^2)}{\pi(\nu_0^2 - \nu_{\max}^2)^2} \right\}. \tag{7}$$

**Proposition 2.** *Suppose that a sample $\mathbf{x}_0$ is initialized in the distribution $P^{(0)}$, i.e., $\mathbf{x}_0 \sim P^{(0)}$, then for any constant $\nu_{\max} < \nu_0$, with probability at least $1 - \exp(-\Omega(d))$, we have $\|\mathbf{n}_0\|^2 \geq \frac{3\nu_0^2 + \nu_{\max}^2}{4} d$.*

*Proof of Proposition 2.* Since $\mathbf{x}_0 \sim P^{(0)} = \mathcal{N}(\mathbf{0}_d, \nu_0^2 \mathbf{I}_d)$ and $\mathbf{N}^T \mathbf{N} = \mathbf{I}_n$, we know $\mathbf{n}_0 = \mathbf{N}^T \mathbf{x}_0 \sim \mathcal{N}(\mathbf{0}_n, \nu_0^2 \mathbf{I}_n)$. Therefore, by Lemma 1 we can bound

$$\mathbb{P} \left( \|\mathbf{n}_0\|^2 \leq \frac{3\nu_0^2 + \nu_{\max}^2}{4} d \right) = \mathbb{P} \left( \frac{\|\mathbf{n}_0\|^2}{\nu_0^2} \leq d - 2\sqrt{d \cdot \left( \frac{\nu_0^2 - \nu_{\max}^2}{8\nu_0^2} \right)^2 d} \right)$$

$$\leq \mathbb{P} \left( \frac{\|\mathbf{n}_0\|^2}{\nu_0^2} \leq n - 2\sqrt{n \left( \frac{\nu_0^2 - \nu_{\max}^2}{8\nu_0^2} \right)^2 \frac{d}{2}} \right)$$

$$\leq \exp \left( - \left( \frac{\nu_0^2 - \nu_{\max}^2}{8\nu_0^2} \right)^2 \frac{d}{2} \right),$$

where the second last step follows from the assumption $d - n = r = o(d)$. Hence we complete the proof of Proposition 2. $\qquad \square$

Then, with the assumption that the initialization satisfies $\|\mathbf{n}_0\|^2 \geq \frac{3\nu_0^2 + \nu_{\max}^2}{4} d$, the following proposition shows that $\|\mathbf{n}_t\|$ remains large with high probability.

**Proposition 3.** *Consider a data distribution $P$ satisfies the constraints specified in Theorem 1. We follow the Langevin dynamics for $T = \exp(\mathcal{O}(d))$ steps. Suppose that the initial sample satisfies $\|\mathbf{n}_0\|^2 \geq \frac{3\nu_0^2 + \nu_{\max}^2}{4} d$, then with probability at least $1 - T \cdot \exp(-\Omega(d))$, we have that $\|\mathbf{n}_t\|^2 > \frac{\nu_0^2 + \nu_{\max}^2}{2} d$ for all $t \in \{0\} \cup [T]$.*

*Proof of Proposition 3.* To establish a lower bound on $\|\mathbf{n}_t\|$, we consider different cases of the step size $\delta_t$. Intuitively, when $\delta_t$ is large enough, $\mathbf{n}_t$ will be too noisy due to the introduction of random noise $\sqrt{\delta_t} \boldsymbol{\epsilon}_t^{(\mathbf{n})}$ in equation 5. While for small $\delta_t$, the update of $\mathbf{n}_t$ is bounded and thus we can iteratively analyze $\mathbf{n}_t$. We first handle the case of large $\delta_t$ in the following lemma.

**Lemma 3.** *If $\delta_t > \nu_0^2$, with probability at least $1 - \exp(-\Omega(d))$, for $\mathbf{n}_t$ satisfying equation 5, we have $\|\mathbf{n}_t\|^2 \geq \frac{3\nu_0^2 + \nu_{\max}^2}{4} d$ regardless of the previous state $\mathbf{x}_{t-1}$.*

*Proof of Lemma 3.* Denote $\mathbf{v} := \mathbf{n}_{t-1} + \frac{\delta_t}{2} \mathbf{N}^T \nabla_{\mathbf{x}} \log P(\mathbf{x}_{t-1})$ for simplicity. Note that $\mathbf{v}$ is fixed for any given $\mathbf{x}_{t-1}$. We decompose $\boldsymbol{\epsilon}_t^{(\mathbf{n})}$ into a vector aligning with $\mathbf{v}$ and another vector orthogonal to $\mathbf{v}$. Consider an orthonormal matrix $\mathbf{M} \in \mathbb{R}^{n \times (n-1)}$ such that $\mathbf{M}^T \mathbf{v} = \mathbf{0}_{n-1}$ and $\mathbf{M}^T \mathbf{M} = \mathbf{I}_{n-1}$.

By denoting $\mathbf{u} := \boldsymbol{\epsilon}_t^{(\mathbf{n})} - \mathbf{M}\mathbf{M}^T \boldsymbol{\epsilon}_t^{(\mathbf{n})}$ we have $\mathbf{M}^T \mathbf{u} = \mathbf{0}_{n-1}$, thus we obtain

$$
\begin{aligned}
\|\mathbf{n}_t\|^2 &= \left\| \mathbf{v} + \sqrt{\delta_t} \boldsymbol{\epsilon}_t^{(\mathbf{n})} \right\|^2 \\
&= \left\| \mathbf{v} + \sqrt{\delta_t} \mathbf{u} + \sqrt{\delta_t} \mathbf{M}\mathbf{M}^T \boldsymbol{\epsilon}_t^{(\mathbf{n})} \right\|^2 \\
&= \left\| \mathbf{v} + \sqrt{\delta_t} \mathbf{u} \right\|^2 + \left\| \sqrt{\delta_t} \mathbf{M}\mathbf{M}^T \boldsymbol{\epsilon}_t^{(\mathbf{n})} \right\|^2 \\
&\geq \left\| \sqrt{\delta_t} \mathbf{M}\mathbf{M}^T \boldsymbol{\epsilon}_t^{(\mathbf{n})} \right\|^2 \\
&\geq \nu_0^2 \left\| \mathbf{M}^T \boldsymbol{\epsilon}_t^{(\mathbf{n})} \right\|^2 .
\end{aligned}
$$

Since $\boldsymbol{\epsilon}_t^{(\mathbf{n})} \sim \mathcal{N}(\mathbf{0}_n, \boldsymbol{I}_n)$ and $\mathbf{M}^T \mathbf{M} = \boldsymbol{I}_{n-1}$, we obtain $\mathbf{M}^T \boldsymbol{\epsilon}_t^{(\mathbf{n})} \sim \mathcal{N}(\mathbf{0}_{n-1}, \boldsymbol{I}_{n-1})$. Therefore, by Lemma 1 we can bound

$$
\begin{aligned}
\mathbb{P}\left( \|\mathbf{n}_t\|^2 \leq \frac{3\nu_0^2 + \nu_{\max}^2}{4} d \right) &\leq \mathbb{P}\left( \left\| \mathbf{M}^T \boldsymbol{\epsilon}_t^{(\mathbf{n})} \right\|^2 \leq \frac{3\nu_0^2 + \nu_{\max}^2}{4\nu_0^2} d \right) \\
&= \mathbb{P}\left( \left\| \mathbf{M}^T \boldsymbol{\epsilon}_t^{(\mathbf{n})} \right\|^2 \leq d - 2\sqrt{d \cdot \left( \frac{\nu_0^2 - \nu_{\max}^2}{8\nu_0^2} \right)^2 d} \right) \\
&\leq \mathbb{P}\left( \left\| \mathbf{M}^T \boldsymbol{\epsilon}_t^{(\mathbf{n})} \right\|^2 \leq (n-1) - 2\sqrt{(n-1)\left( \frac{\nu_0^2 - \nu_{\max}^2}{8\nu_0^2} \right)^2 \frac{d}{2}} \right) \\
&\leq \exp\left( -\left( \frac{\nu_0^2 - \nu_{\max}^2}{8\nu_0^2} \right)^2 \frac{d}{2} \right),
\end{aligned}
$$

where the second last step follows from the assumption $d - n = r = o(d)$. Hence we complete the proof of Lemma 3. $\qquad\square$

We then consider the case when $\delta_t \leq \nu_0^2$. Let $\mathbf{r} := \mathbf{R}^T \mathbf{x}$ and $\mathbf{n} := \mathbf{N}^T \mathbf{x}$, then $\mathbf{x} = \mathbf{R}\mathbf{r} + \mathbf{N}\mathbf{n}$. We first show that when $\|\mathbf{n}\|^2 \geq \frac{\nu_0^2 + \nu_{\max}^2}{2} d$, $P^{(i)}(\mathbf{x})$ is exponentially smaller than $P^{(0)}(\mathbf{x})$ for all $i \in [k]$ in the following lemma.

**Lemma 4.** *Given that* $\|\mathbf{n}\|^2 \geq \frac{\nu_0^2 + \nu_{\max}^2}{2} d$ *and* $\|\boldsymbol{\mu}_i\|^2 \leq \frac{\nu_0^2 - \nu_i^2}{2}\left( \log\left( \frac{\nu_i^2}{\nu_0^2} \right) - \frac{\nu_i^2}{2\nu_0^2} + \frac{\nu_0^2}{2\nu_i^2} \right) d$ *for all* $i \in [k]$, *we have* $\frac{P^{(i)}(\mathbf{x})}{P^{(0)}(\mathbf{x})} \leq \exp(-\Omega(d))$ *for all* $i \in [k]$.

*Proof of Lemma 4.* For all $i \in [k]$, define $\rho_i(\mathbf{x}) := \frac{P^{(i)}(\mathbf{x})}{P^{(0)}(\mathbf{x})}$, then

$$
\begin{aligned}
\rho_i(\mathbf{x}) = \frac{P^{(i)}(\mathbf{x})}{P^{(0)}(\mathbf{x})} &= \frac{(2\pi\nu_i^2)^{-d/2} \exp\left( -\frac{1}{2\nu_i^2} \|\mathbf{x} - \boldsymbol{\mu}_i\|^2 \right)}{(2\pi\nu_0^2)^{-d/2} \exp\left( -\frac{1}{2\nu_0^2} \|\mathbf{x}\|^2 \right)} \\
&= \left( \frac{\nu_0^2}{\nu_i^2} \right)^{d/2} \exp\left( \frac{1}{2\nu_0^2} \|\mathbf{x}\|^2 - \frac{1}{2\nu_i^2} \|\mathbf{x} - \boldsymbol{\mu}_i\|^2 \right) \\
&= \left( \frac{\nu_0^2}{\nu_i^2} \right)^{d/2} \exp\left( \left( \frac{1}{2\nu_0^2} - \frac{1}{2\nu_i^2} \right) \|\mathbf{N}\mathbf{n}\|^2 + \left( \frac{\|\mathbf{R}\mathbf{r}\|^2}{2\nu_0^2} - \frac{\|\mathbf{R}\mathbf{r} - \boldsymbol{\mu}_i\|^2}{2\nu_i^2} \right) \right) \\
&= \left( \frac{\nu_0^2}{\nu_i^2} \right)^{d/2} \exp\left( \left( \frac{1}{2\nu_0^2} - \frac{1}{2\nu_i^2} \right) \|\mathbf{n}\|^2 + \left( \frac{\|\mathbf{r}\|^2}{2\nu_0^2} - \frac{\|\mathbf{r} - \mathbf{R}^T \boldsymbol{\mu}_i\|^2}{2\nu_i^2} \right) \right),
\end{aligned}
$$

where the last step follows from the definition that $\mathbf{R} \in \mathbb{R}^{d \times r}$ an orthonormal basis of the vector space $\{\boldsymbol{\mu}_i\}_{i \in [k]}$ and $\mathbf{N}^T \mathbf{N} = \boldsymbol{I}_n$. Since $\nu_0^2 > \nu_i^2$, the quadratic term $\frac{\|\mathbf{r}\|^2}{2\nu_0^2} - \frac{\|\mathbf{r} - \mathbf{R}^T \boldsymbol{\mu}_i\|^2}{2\nu_i^2}$ is

maximized at $\mathbf{r} = \frac{\nu_0^2 \mathbf{R}^T \boldsymbol{\mu}_i}{\nu_0^2 - \nu_i^2}$. Therefore,

$$
\frac{\|\mathbf{r}\|^2}{2\nu_0^2} - \frac{\|\mathbf{r} - \mathbf{R}^T \boldsymbol{\mu}_i\|^2}{2\nu_i^2} \leq \frac{\nu_0^4 \|\mathbf{R}^T \boldsymbol{\mu}_i\|^2}{2\nu_0^2(\nu_0^2 - \nu_i^2)^2} - \frac{1}{2\nu_i^2} \left( \frac{\nu_0^2}{\nu_0^2 - \nu_i^2} - 1 \right)^2 \|\mathbf{R}^T \boldsymbol{\mu}_i\|^2 = \frac{\|\boldsymbol{\mu}_i\|^2}{2(\nu_0^2 - \nu_i^2)}.
$$

Hence, for $\|\mathbf{n}\|^2 \geq \frac{\nu_0^2 + \nu_{\max}^2}{2} d$ and $\|\boldsymbol{\mu}_i\|^2 \leq \frac{\nu_0^2 - \nu_i^2}{2} \left( \log\left(\frac{\nu_i^2}{\nu_0^2}\right) - \frac{\nu_i^2}{2\nu_0^2} + \frac{\nu_0^2}{2\nu_i^2} \right) d$, we have

$$
\rho_i(\mathbf{x}) = \left( \frac{\nu_0^2}{\nu_i^2} \right)^{d/2} \exp\left( \left( \frac{1}{2\nu_0^2} - \frac{1}{2\nu_i^2} \right) \|\mathbf{n}\|^2 + \left( \frac{\|\mathbf{r}\|^2}{2\nu_0^2} - \frac{\|\mathbf{r} - \mathbf{R}^T \boldsymbol{\mu}_i\|^2}{2\nu_i^2} \right) \right)
$$

$$
\leq \left( \frac{\nu_0^2}{\nu_i^2} \right)^{d/2} \exp\left( \left( \frac{1}{2\nu_0^2} - \frac{1}{2\nu_i^2} \right) \frac{\nu_0^2 + \nu_i^2}{2} d + \frac{\|\boldsymbol{\mu}_i\|^2}{2(\nu_0^2 - \nu_i^2)} \right)
$$

$$
= \exp\left( -\left( \log\left( \frac{\nu_i^2}{\nu_0^2} \right) - \frac{\nu_i^2}{2\nu_0^2} + \frac{\nu_0^2}{2\nu_i^2} \right) \frac{d}{2} + \frac{\|\boldsymbol{\mu}_i\|^2}{2(\nu_0^2 - \nu_i^2)} \right)
$$

$$
\leq \exp\left( -\left( \log\left( \frac{\nu_i^2}{\nu_0^2} \right) - \frac{\nu_i^2}{2\nu_0^2} + \frac{\nu_0^2}{2\nu_i^2} \right) \frac{d}{4} \right).
$$

Notice that for function $f(z) = \log z - \frac{z}{2} + \frac{1}{2z}$, we have $f(1) = 0$ and $\frac{\mathrm{d}}{\mathrm{d}z} f(z) = \frac{1}{z} - \frac{1}{2} - \frac{1}{2z^2} = -\frac{1}{2}\left( \frac{1}{z} - 1 \right)^2 < 0$ when $z \in (0, 1)$. Thus, $\log\left( \frac{\nu_i^2}{\nu_0^2} \right) - \frac{\nu_i^2}{2\nu_0^2} + \frac{\nu_0^2}{2\nu_i^2}$ is a positive constant for $\nu_i < \nu_0$, i.e., $\rho_i(\mathbf{x}) = \exp(-\Omega(d))$. Therefore we finish the proof of Lemma 4. $\qquad\square$

Lemma 4 implies that when $\|\mathbf{n}\|$ is large, the Gaussian mode $P^{(0)}$ dominates other modes $P^{(i)}$. To bound $\|\mathbf{n}_t\|$, we first consider a simpler case that $\|\mathbf{n}_{t-1}\|$ is large. Intuitively, the following lemma proves that when the previous state $\mathbf{n}_{t-1}$ is far from a mode, a single step of Langevin dynamics with bounded step size is not enough to find the mode.

**Lemma 5.** *Suppose $\delta_t \leq \nu_0^2$ and $\|\mathbf{n}_{t-1}\|^2 > 36\nu_0^2 d$, then for $\mathbf{n}_t$ following from equation 5, we have $\|\mathbf{n}_t\|^2 \geq \nu_0^2 d$ with probability at least $1 - \exp(-\Omega(d))$.*

*Proof of Lemma 5.* From the recursion of $\mathbf{n}_t$ in equation 5 we have

$$
\mathbf{n}_t = \mathbf{n}_{t-1} + \frac{\delta_t}{2} \mathbf{N}^T \nabla_{\mathbf{x}} \log P(\mathbf{x}_{t-1}) + \sqrt{\delta_t} \boldsymbol{\epsilon}_t^{(\mathbf{n})}
$$

$$
= \mathbf{n}_{t-1} - \frac{\delta_t}{2} \sum_{i=0}^{k} \frac{P^{(i)}(\mathbf{x}_{t-1})}{P(\mathbf{x}_{t-1})} \cdot \frac{\mathbf{N}^T (\mathbf{x}_{t-1} - \boldsymbol{\mu}_i)}{\nu_i^2} + \sqrt{\delta_t} \boldsymbol{\epsilon}_t^{(\mathbf{n})}
$$

$$
= \left( 1 - \frac{\delta_t}{2} \sum_{i=0}^{k} \frac{P^{(i)}(\mathbf{x}_{t-1})}{P(\mathbf{x}_{t-1})} \cdot \frac{1}{\nu_i^2} \right) \mathbf{n}_{t-1} + \sqrt{\delta_t} \boldsymbol{\epsilon}_t^{(\mathbf{n})}. \tag{8}
$$

By Lemma 4, we have $\frac{P^{(i)}(\mathbf{x}_{j-1})}{P^{(0)}(\mathbf{x}_{j-1})} \leq \exp(-\Omega(d))$ for all $i \in [k]$, therefore

$$
1 - \frac{\delta_t}{2} \sum_{i=0}^{k} \frac{P^{(i)}(\mathbf{x}_{t-1})}{P(\mathbf{x}_{t-1})} \cdot \frac{1}{\nu_i^2} \geq 1 - \frac{\delta_t}{2} \cdot \frac{1}{\nu_0^2} - \frac{\delta_t}{2} \sum_{i \in [k]} \frac{w_i P^{(i)}(\mathbf{x}_{t-1})}{w_0 P^{(0)}(\mathbf{x}_{t-1})} \cdot \frac{1}{\nu_i^2} \geq 1 - \frac{1}{2} - \exp(-\Omega(d)) > \frac{1}{3}. \tag{9}
$$

On the other hand, from $\boldsymbol{\epsilon}_t^{(\mathbf{n})} \sim \mathcal{N}(\mathbf{0}_n, \boldsymbol{I}_n)$ we know $\frac{\langle \mathbf{n}_{t-1}, \boldsymbol{\epsilon}_t^{(\mathbf{n})} \rangle}{\|\mathbf{n}_{t-1}\|} \sim \mathcal{N}(0, 1)$ for any fixed $\mathbf{n}_{t-1} \neq \mathbf{0}_n$, hence by Lemma 2 we have

$$
\mathbb{P}\left( \frac{\langle \mathbf{n}_{t-1}, \boldsymbol{\epsilon}_t^{(\mathbf{n})} \rangle}{\|\mathbf{n}_{t-1}\|} \geq \frac{\sqrt{d}}{4} \right) = \mathbb{P}\left( \frac{\langle \mathbf{n}_{t-1}, \boldsymbol{\epsilon}_t^{(\mathbf{n})} \rangle}{\|\mathbf{n}_{t-1}\|} \leq -\frac{\sqrt{d}}{4} \right) \leq \frac{4}{\sqrt{2\pi d}} \exp\left( -\frac{d}{32} \right) \tag{10}
$$

Combining equation 8, equation 9 and equation 10 gives that

$$\|\mathbf{n}_t\|^2 \geq \left(\frac{1}{3}\right)^2 \|\mathbf{n}_{t-1}\|^2 - 2\nu_0 |\langle \mathbf{n}_{t-1}, \boldsymbol{\epsilon}_t^{(\mathbf{n})}\rangle|$$

$$\geq \frac{1}{9}\|\mathbf{n}_{t-1}\|^2 - \frac{\nu_0\sqrt{d}}{2}\|\mathbf{n}_{t-1}\|$$

$$\geq \frac{1}{9}\cdot 36\nu_0^2 d - \frac{\nu_0\sqrt{d}}{2}\cdot 6\nu_0\sqrt{d}$$

$$= \nu_0^2 d$$

with probability at least $1 - \frac{8}{\sqrt{2\pi d}}\exp\left(-\frac{d}{32}\right) = 1 - \exp(-\Omega(d))$. This proves Lemma 5. $\square$

We then proceed to bound $\|\mathbf{n}_t\|$ iteratively for $\|\mathbf{n}_{t-1}\|^2 \leq 36\nu_0^2 d$. Recall that equation 5 gives

$$\mathbf{n}_t = \mathbf{n}_{t-1} + \frac{\delta_t}{2}\mathbf{N}^T \nabla_{\mathbf{x}} \log P(\mathbf{x}_{t-1}) + \sqrt{\delta_t}\boldsymbol{\epsilon}_t^{(\mathbf{n})}.$$

We notice that the difficulty of solving $\mathbf{n}_t$ exhibits in the dependence of $\log P(\mathbf{x}_{t-1})$ on $\mathbf{r}_{t-1}$. Since $P = \sum_{i=0}^k w_i P^{(i)} = \sum_{i=0}^k w_i \mathcal{N}(\boldsymbol{\mu}_i, \nu_i^2 \mathbf{I}_d)$, we can rewrite the score function as

$$\nabla_{\mathbf{x}} \log P(\mathbf{x}) = \frac{\nabla_{\mathbf{x}} P(\mathbf{x})}{P(\mathbf{x})} = -\sum_{i=0}^k \frac{P^{(i)}(\mathbf{x})}{P(\mathbf{x})}\cdot\frac{\mathbf{x} - \boldsymbol{\mu}_i}{\nu_i^2} = -\frac{\mathbf{x}}{\nu_0^2} + \sum_{i\in[k]} \frac{P^{(i)}(\mathbf{x})}{P(\mathbf{x})}\left(\frac{\mathbf{x}}{\nu_0^2} - \frac{\mathbf{x} - \boldsymbol{\mu}_i}{\nu_i^2}\right). \tag{11}$$

Now, instead of directly working with $\mathbf{n}_t$, we consider a surrogate recursion $\hat{\mathbf{n}}_t$ such that $\hat{\mathbf{n}}_0 = \mathbf{n}_0$ and for all $t \geq 1$,

$$\hat{\mathbf{n}}_t = \hat{\mathbf{n}}_{t-1} - \frac{\delta_t}{2\nu_0^2}\hat{\mathbf{n}}_{t-1} + \sqrt{\delta_t}\boldsymbol{\epsilon}_t^{(\mathbf{n})}. \tag{12}$$

The advantage of the surrogate recursion is that $\hat{\mathbf{n}}_t$ is independent of $\mathbf{r}$, thus we can obtain the closed-form solution to $\hat{\mathbf{n}}_t$. Before we proceed to bound $\hat{\mathbf{n}}_t$, we first show that $\hat{\mathbf{n}}_t$ is sufficiently close to the original recursion $\mathbf{n}_t$ in the following lemma.

**Lemma 6.** *For any* $t \geq 1$, *given that* $\delta_j \leq \nu_0^2$ *and* $\frac{\nu_0^2 + \nu_{\max}^2}{2}d \leq \|\mathbf{n}_{j-1}\|^2 \leq 36\nu_0^2 d$ *for all* $j \in [t]$ *and* $\|\boldsymbol{\mu}_i\|^2 \leq \frac{\nu_0^2 - \nu_i^2}{2}\left(\log\left(\frac{\nu_i^2}{\nu_0^2}\right) - \frac{\nu_i^2}{2\nu_0^2} + \frac{\nu_0^2}{2\nu_i^2}\right)d$ *for all* $i \in [k]$, *we have* $\|\hat{\mathbf{n}}_t - \mathbf{n}_t\| \leq \frac{t}{\exp(\Omega(d))}\sqrt{d}$.

*Proof of Lemma 6.* Upon comparing equation 5 and equation 12, by equation 11 we have that for all $j \in [t]$,

$$\|\hat{\mathbf{n}}_j - \mathbf{n}_j\| = \left\|\hat{\mathbf{n}}_{j-1} - \frac{\delta_j}{2\nu_0^2}\hat{\mathbf{n}}_{j-1} - \mathbf{n}_{j-1} - \frac{\delta_j}{2}\mathbf{N}^T\nabla_{\mathbf{x}}\log P(\mathbf{x}_{j-1})\right\|$$

$$= \left\|\left(1 - \frac{\delta_j}{2\nu_0^2}\right)(\hat{\mathbf{n}}_{j-1} - \mathbf{n}_{j-1}) + \frac{\delta_j}{2}\sum_{i\in[k]}\frac{P^{(i)}(\mathbf{x}_{j-1})}{P(\mathbf{x}_{j-1})}\left(\frac{1}{\nu_i^2} - \frac{1}{\nu_0^2}\right)\mathbf{n}_{j-1}\right\|$$

$$\leq \left(1 - \frac{\delta_j}{2\nu_0^2}\right)\|\hat{\mathbf{n}}_{j-1} - \mathbf{n}_{j-1}\| + \sum_{i\in[k]}\frac{\delta_j}{2}\frac{P^{(i)}(\mathbf{x}_{j-1})}{P(\mathbf{x}_{j-1})}\left(\frac{1}{\nu_i^2} - \frac{1}{\nu_0^2}\right)\|\mathbf{n}_{j-1}\|$$

$$\leq \|\hat{\mathbf{n}}_{j-1} - \mathbf{n}_{j-1}\| + \sum_{i\in[k]}\frac{\delta_j}{2}\frac{P^{(i)}(\mathbf{x}_{j-1})}{P^{(0)}(\mathbf{x}_{j-1})}\left(\frac{1}{\nu_i^2} - \frac{1}{\nu_0^2}\right)6\nu_0\sqrt{d}.$$

By Lemma 4, we have $\frac{P^{(i)}(\mathbf{x}_{j-1})}{P^{(0)}(\mathbf{x}_{j-1})} \leq \exp(-\Omega(d))$ for all $i \in [k]$, hence we obtain a recursive bound

$$\|\hat{\mathbf{n}}_j - \mathbf{n}_j\| \leq \|\hat{\mathbf{n}}_{j-1} - \mathbf{n}_{j-1}\| + \frac{1}{\exp(\Omega(d))}\sqrt{d}.$$

Finally, by $\hat{\mathbf{n}}_0 = \mathbf{n}_0$, we have

$$\|\hat{\mathbf{n}}_t - \mathbf{n}_t\| = \sum_{j \in [t]} (\|\hat{\mathbf{n}}_j - \mathbf{n}_j\| - \|\hat{\mathbf{n}}_{j-1} - \mathbf{n}_{j-1}\|) \leq \frac{t}{\exp(\Omega(d))}\sqrt{d}.$$

Hence we obtain Lemma 6. $\qquad\square$

We then proceed to analyze $\hat{\mathbf{n}}_t$, The following lemma gives us the closed-form solution of $\hat{\mathbf{n}}_t$. We slightly abuse the notations here, e.g., $\prod_{i=c_1}^{c_2} \left(1 - \frac{\delta_i}{2\nu_0^2}\right) = 1$ and $\sum_{j=c_1}^{c_2} \delta_j = 0$ for $c_1 > c_2$.

**Lemma 7.** *For all $t \geq 0$, $\hat{\mathbf{n}}_t \sim \mathcal{N}\left(\prod_{i=1}^{t}\left(1 - \frac{\delta_i}{2\nu_0^2}\right)\mathbf{n}_0, \ \sum_{j=1}^{t}\prod_{i=j+1}^{t}\left(1 - \frac{\delta_i}{2\nu_0^2}\right)^2 \delta_j \boldsymbol{I}_n\right)$, where*

*the mean and covariance satisfy $\prod_{i=1}^{t}\left(1 - \frac{\delta_i}{2\nu_0^2}\right)^2 + \frac{1}{\nu_0^2}\sum_{j=1}^{t}\prod_{i=j+1}^{t}\left(1 - \frac{\delta_i}{2\nu_0^2}\right)^2 \delta_j \geq 1$.*

*Proof of Lemma 7.* We prove the two properties by induction. When $t = 0$, they are trivial. Suppose they hold for $t - 1$, then for the distribution of $\hat{\mathbf{n}}_t$, we have

$$\hat{\mathbf{n}}_t = \hat{\mathbf{n}}_{t-1} - \frac{\delta_t}{2\nu_0^2}\hat{\mathbf{n}}_{t-1} + \sqrt{\delta_t}\boldsymbol{\epsilon}_t^{(\mathbf{n})}$$

$$\sim \mathcal{N}\left(\left(1 - \frac{\delta_t}{2\nu_0^2}\right)\prod_{i=1}^{t-1}\left(1 - \frac{\delta_i}{2\nu_0^2}\right)\mathbf{n}_0, \ \left(1 - \frac{\delta_t}{2\nu_0^2}\right)^2 \sum_{j=1}^{t-1}\prod_{i=j+1}^{t-1}\left(1 - \frac{\delta_i}{2\nu_0^2}\right)^2 \delta_j \boldsymbol{I}_n + \delta_t \boldsymbol{I}_n\right)$$

$$= \mathcal{N}\left(\prod_{i=1}^{t}\left(1 - \frac{\delta_i}{2\nu_0^2}\right)\mathbf{n}_0, \ \sum_{j=1}^{t}\prod_{i=j+1}^{t}\left(1 - \frac{\delta_i}{2\nu_0^2}\right)^2 \delta_j \boldsymbol{I}_n\right).$$

For the second property,

$$\prod_{i=1}^{t}\left(1 - \frac{\delta_i}{2\nu_0^2}\right)^2 + \frac{1}{\nu_0^2}\sum_{j=1}^{t}\prod_{i=j+1}^{t}\left(1 - \frac{\delta_i}{2\nu_0^2}\right)^2 \delta_j$$

$$= \left(1 - \frac{\delta_t}{2\nu_0^2}\right)^2\left(\prod_{i=1}^{t-1}\left(1 - \frac{\delta_i}{2\nu_0^2}\right)^2 + \frac{1}{\nu_0^2}\sum_{j=1}^{t-1}\prod_{i=j+1}^{t-1}\left(1 - \frac{\delta_i}{2\nu_0^2}\right)^2 \delta_j\right) + \frac{1}{\nu_0^2}\delta_t$$

$$\geq \left(1 - \frac{\delta_t}{2\nu_0^2}\right)^2 + \frac{1}{\nu_0^2}\delta_t = 1 + \frac{\delta_t^2}{4\nu_0^4} \geq 1.$$

Hence we finish the proof of Lemma 7. $\qquad\square$

Armed with Lemma 7, we are now ready to establish the lower bound on $\|\hat{\mathbf{n}}_t\|$. For simplicity, denote $\alpha := \prod_{i=1}^{t}\left(1 - \frac{\delta_i}{2\nu_0^2}\right)^2$ and $\beta := \frac{1}{\nu_0^2}\sum_{j=1}^{t}\prod_{i=j+1}^{t}\left(1 - \frac{\delta_i}{2\nu_0^2}\right)^2 \delta_j$. By Lemma 7 we know $\hat{\mathbf{n}}_t \sim \mathcal{N}(\alpha\mathbf{n}_0, \beta\nu_0^2\boldsymbol{I}_n)$, so we can write $\hat{\mathbf{n}}_t = \alpha\mathbf{n}_0 + \sqrt{\beta}\nu_0\boldsymbol{\epsilon}$, where $\boldsymbol{\epsilon} \sim \mathcal{N}(\mathbf{0}_n, \boldsymbol{I}_n)$.

**Lemma 8.** *Given that $\|\hat{\mathbf{n}}_0\|^2 \geq \frac{3\nu_0^2 + \nu_{\max}^2}{4}d$, we have $\|\hat{\mathbf{n}}_t\|^2 \geq \frac{5\nu_0^2 + 3\nu_{\max}^2}{8}d$ with probability at least $1 - \exp(-\Omega(d))$.*

*Proof of Lemma 8.* By $\hat{\mathbf{n}}_t = \alpha\mathbf{n}_0 + \sqrt{\beta}\nu_0\boldsymbol{\epsilon}$ we have

$$\|\hat{\mathbf{n}}_t\|^2 = \alpha^2\|\mathbf{n}_0\|^2 + \beta\nu_0^2\|\boldsymbol{\epsilon}\|^2 + 2\alpha\sqrt{\beta}\nu_0\langle\mathbf{n}_0, \boldsymbol{\epsilon}\rangle$$

By Lemma 1 we can bound

$$
\mathbb{P}\left(\|\boldsymbol{\epsilon}\|^2 \leq \frac{3\nu_0^2 + \nu_{\max}^2}{4\nu_0^2} d\right) = \mathbb{P}\left(\|\boldsymbol{\epsilon}\|^2 \leq d - 2\sqrt{d \cdot \left(\frac{\nu_0^2 - \nu_{\max}^2}{8\nu_0^2}\right)^2 d}\right)
$$

$$
\leq \mathbb{P}\left(\|\boldsymbol{\epsilon}\|^2 \leq (n-1) - 2\sqrt{(n-1)\left(\frac{\nu_0^2 - \nu_{\max}^2}{8\nu_0^2}\right)^2 \frac{d}{2}}\right)
$$

$$
\leq \exp\left(-\left(\frac{\nu_0^2 - \nu_{\max}^2}{8\nu_0^2}\right)^2 \frac{d}{2}\right),
$$

where the second last step follows from the assumption $d - n = r = o(d)$. Since $\boldsymbol{\epsilon} \sim \mathcal{N}(\mathbf{0}_n, \boldsymbol{I}_n)$, we know $\frac{\langle \mathbf{n}_0, \boldsymbol{\epsilon}\rangle}{\|\mathbf{n}_0\|} \sim \mathcal{N}(0, 1)$. Therefore by Lemma 2,

$$
\mathbb{P}\left(\frac{\langle \mathbf{n}_0, \boldsymbol{\epsilon}\rangle}{\|\mathbf{n}_0\|} \leq -\frac{\nu_0^2 - \nu_{\max}^2}{4\nu_0\sqrt{3\nu_0^2 + \nu_{\max}^2}}\sqrt{d}\right) \leq \frac{4\nu_0\sqrt{3\nu_0^2 + \nu_{\max}^2}}{\sqrt{2\pi}(\nu_0^2 - \nu_{\max}^2)\sqrt{d}} \exp\left(-\frac{(\nu_0^2 - \nu_{\max}^2)^2 d}{32\nu_0^2(3\nu_0^2 + \nu_{\max}^2)}\right)
$$

Conditioned on $\|\hat{\mathbf{n}}_0\|^2 \geq \frac{3\nu_0^2 + \nu_{\max}^2}{4}d$, $\|\boldsymbol{\epsilon}\|^2 > \frac{3\nu_0^2 + \nu_{\max}^2}{4\nu_0^2}d$ and $\frac{1}{\|\mathbf{n}_0\|}\langle \mathbf{n}_0, \boldsymbol{\epsilon}\rangle > -\frac{\nu_0^2 - \nu_{\max}^2}{4\nu_0\sqrt{3\nu_0^2 + \nu_{\max}^2}}\sqrt{d}$, since Lemma 7 gives $\alpha^2 + \beta \geq 1$ we have

$$
\|\hat{\mathbf{n}}_t\|^2 = \alpha^2 \|\mathbf{n}_0\|^2 + \beta\nu_0^2 \|\boldsymbol{\epsilon}\|^2 + 2\alpha\sqrt{\beta}\nu_0\langle \mathbf{n}_0, \boldsymbol{\epsilon}\rangle
$$

$$
\geq \alpha^2 \|\mathbf{n}_0\|^2 + \beta\nu_0^2 \|\boldsymbol{\epsilon}\|^2 - 2\alpha\sqrt{\beta}\nu_0 \|\mathbf{n}_0\| \frac{\nu_0^2 - \nu_{\max}^2}{4\nu_0\sqrt{3\nu_0^2 + \nu_{\max}^2}}\sqrt{d}
$$

$$
\geq \alpha^2 \|\mathbf{n}_0\|^2 + \beta\nu_0^2 \|\boldsymbol{\epsilon}\|^2 - 2\alpha\sqrt{\beta}\nu_0 \|\mathbf{n}_0\| \|\boldsymbol{\epsilon}\| \cdot \frac{\nu_0^2 - \nu_{\max}^2}{6\nu_0^2 + 2\nu_{\max}^2}
$$

$$
\geq \left(1 - \frac{\nu_0^2 - \nu_{\max}^2}{6\nu_0^2 + 2\nu_{\max}^2}\right)\left(\alpha^2 \|\mathbf{n}_0\|^2 + \beta\nu_0^2 \|\boldsymbol{\epsilon}\|^2\right)
$$

$$
\geq \frac{5\nu_0^2 + 3\nu_{\max}^2}{6\nu_0^2 + 2\nu_{\max}^2}\left(\alpha^2 + \beta\right) \cdot \frac{3\nu_0^2 + \nu_{\max}^2}{4}d
$$

$$
\geq \frac{5\nu_0^2 + 3\nu_{\max}^2}{8}d.
$$

Hence by union bound, we complete the proof of Lemma 8. $\qquad\square$

Upon having all the above lemmas, we are now ready to establish Proposition 3 by induction. Suppose the theorem holds for all $T$ values of $1, \cdots, T-1$. We consider the following 3 cases:

- If there exists some $t \in [T]$ such that $\delta_t > \nu_0^2$, by Lemma 3 we know that with probability at least $1 - \exp(-\Omega(d))$, we have $\|\mathbf{n}_t\|^2 \geq \frac{3\nu_0^2 + \nu_{\max}^2}{4}d$, thus the problem reduces to the two sub-arrays $\mathbf{n}_0, \cdots, \mathbf{n}_{t-1}$ and $\mathbf{n}_t, \cdots, \mathbf{n}_T$, which can be solved by induction.

- Suppose $\delta_t \leq \nu_0^2$ for all $t \in [T]$. If there exists some $t \in [T]$ such that $\|\mathbf{n}_{t-1}\|^2 > 36\nu_0^2 d$, by Lemma 5 we know that with probability at least $1 - \exp(-\Omega(d))$, we have $\|\mathbf{n}_t\|^2 \geq \nu_0^2 d > \frac{3\nu_0^2 + \nu_{\max}^2}{4}d$, thus the problem similarly reduces to the two sub-arrays $\mathbf{n}_0, \cdots, \mathbf{n}_{t-1}$ and $\mathbf{n}_t, \cdots, \mathbf{n}_T$, which can be solved by induction.

- Suppose $\delta_t \leq \nu_0^2$ and $\|\mathbf{n}_{t-1}\|^2 \leq 36\nu_0^2 d$ for all $t \in [T]$. Conditioned on $\|\mathbf{n}_{t-1}\|^2 > \frac{\nu_0^2 + \nu_{\max}^2}{2}d$ for all $t \in [T]$, by Lemma 6 we have that for $T = \exp(\mathcal{O}(d))$,

$$
\|\hat{\mathbf{n}}_T - \mathbf{n}_T\| < \left(\sqrt{\frac{5\nu_0^2 + 3\nu_{\max}^2}{8}} - \sqrt{\frac{\nu_0^2 + \nu_{\max}^2}{2}}\right)\sqrt{d}.
$$

By Lemma 8 we have that with probability at least $1 - \exp(-\Omega(d))$,

$$\|\hat{\mathbf{n}}_T\|^2 \geq \frac{5\nu_0^2 + 3\nu_{\max}^2}{8} d.$$

Combining the two inequalities implies the desired bound

$$\|\mathbf{n}_T\| \geq \|\hat{\mathbf{n}}_T\| - \|\hat{\mathbf{n}}_T - \mathbf{n}_T\| > \sqrt{\frac{\nu_0^2 + \nu_{\max}^2}{2} d}.$$

Hence by induction we obtain $\|\mathbf{n}_t\|^2 > \frac{\nu_0^2 + \nu_{\max}^2}{2} d$ for all $t \in [T]$ with probability at least

$$(1 - (T-1)\exp(-\Omega(d))) \cdot (1 - \exp(-\Omega(d))) \geq 1 - T\exp(-\Omega(d)).$$

Therefore we complete the proof of Proposition 3. $\qquad\square$

Finally, combining Propositions 2 and 3 finishes the proof of Theorem 1.

## A.2 PROOF OF COROLLARY 1

By the definition of total variation distance, we have

$$\mathrm{TV}(\hat{P}_t, P) = \sup_A |\hat{P}_t(A) - P(A)|.$$

Specifically, by choosing the event $A$ as $\left\{ \mathbf{x} : \forall i \in [k], \|\mathbf{x} - \boldsymbol{\mu}_i\|^2 \geq \frac{\nu_0^2 + \nu_{\max}^2}{2} d \right\}$, from Theorem 1 we know $\hat{P}_t(A) \geq 1 - T \cdot \exp(-\Omega(d))$. On the other hand, by Lemma 1 we have

$$P(A) = \sum_{i=0}^{k} w_i P^{(i)}(A)$$

$$\leq w_0 + \sum_{i=0}^{k} w_i \exp\left( -\left( \frac{\nu_0^2 - \nu_{\max}^2}{8\nu_{\max}^2} \right) d \right)$$

$$= w_0 + (1 - w_0) \exp\left( -\left( \frac{\nu_0^2 - \nu_{\max}^2}{8\nu_{\max}^2} \right) d \right).$$

Combining the two bounds, we obtain a lower bound on the total variation distance

$$\mathrm{TV}(\hat{P}_t, P) \geq \hat{P}_t(A) - P(A) \geq (1 - w_0)\left( 1 - \frac{T}{\exp(\Omega(d))} \right).$$

## A.3 PROOF OF THEOREM 2: LANGEVIN DYNAMICS UNDER SUB-GAUSSIAN MIXTURES

The proof framework is similar to the proof of Theorem 1. To begin with, we validate Assumption 2.v. in the following lemma:

**Lemma 9.** *For constants $\nu_0, \nu_i, c_\nu, c_L$ satisfying Assumptions 2.iii. and 2.iv., we have $\frac{(1-c_\nu)\nu_0^2 - \nu_i^2}{2(1-c_\nu)} > 0$ and $\log \frac{c_\nu \nu_i^2}{(c_L^2 + c_\nu c_L)\nu_0^2} - \frac{\nu_i^2}{2(1-c_\nu)\nu_0^2} + \frac{(1-c_\nu)\nu_0^2}{2\nu_i^2} > 0$ are both positive constants.*

*Proof of Lemma 9.* From Assumption 2.iv. that $\nu_0^2 > \frac{\nu_{\max}^2}{1-c_\nu} \geq \frac{\nu_i^2}{1-c_\nu}$, we easily obtain $\frac{(1-c_\nu)\nu_0^2 - \nu_i^2}{2(1-c_\nu)} > 0$ is a positive constant. For the second property, let $f(z) := \log \frac{c_\nu \nu_i^2}{(c_L^2 + c_\nu c_L)z} - \frac{\nu_i^2}{2(1-c_\nu)z} + \frac{(1-c_\nu)z}{2\nu_i^2}$. For any $z > \frac{\nu_i^2}{1-c_\nu}$, the derivative of $f(z)$ satisfies

$$\frac{\mathrm{d}}{\mathrm{d}z} f(z) = -\frac{1}{z} + \frac{\nu_i^2}{2(1-c_\nu)z^2} + \frac{1-c_\nu}{2\nu_i^2} = \frac{\nu_i^2}{2(1-c_\nu)}\left( \frac{1-c_\nu}{\nu_i^2} - \frac{1}{z} \right)^2 > 0.$$

Therefore, when $\frac{4(c_L^2 + c_\nu c_L)}{c_\nu(1-c_\nu)} \leq 1$, we have

$$f(\nu_0^2) > f\left( \frac{\nu_i^2}{1-c_\nu} \right) = \log \frac{c_\nu(1-c_\nu)}{c_L^2 + c_\nu c_L} \geq \log 4 > 0.$$

When $\frac{4(c_L^2 + c_\nu c_L)}{c_\nu(1 - c_\nu)} > 1$, we have

$$f(\nu_0^2) > f\left(\frac{4(c_L^2 + c_\nu c_L)}{c_\nu(1 - c_\nu)} \frac{\nu_i^2}{1 - c_\nu}\right) = 2\log \frac{c_\nu(1 - c_\nu)}{2(c_L^2 + c_\nu c_L)} - \frac{c_\nu(1 - c_\nu)}{8(c_L^2 + c_\nu c_L)} + \frac{2(c_L^2 + c_\nu c_L)}{c_\nu(1 - c_\nu)}$$

$$\geq 2 - 2\log 2 - \frac{2(c_L^2 + c_\nu c_L)}{c_\nu(1 - c_\nu)} - \frac{c_\nu(1 - c_\nu)}{8(c_L^2 + c_\nu c_L)} + \frac{2(c_L^2 + c_\nu c_L)}{c_\nu(1 - c_\nu)} > 2 - 2\log 2 - \frac{1}{2} > 0.$$

Thus we obtain Lemma 9. $\qquad\square$

Without loss of generality, we assume $\boldsymbol{\mu}_0 = \mathbf{0}_d$. Similar to the proof of Theorem 1, we decompose

$$\mathbf{x}_t = \mathbf{R}\mathbf{r}_t + \mathbf{N}\mathbf{n}_t, \text{ and } \boldsymbol{\epsilon}_t = \mathbf{R}\boldsymbol{\epsilon}_t^{(\mathbf{r})} + \mathbf{N}\boldsymbol{\epsilon}_t^{(\mathbf{n})},$$

where $\mathbf{R} \in \mathbb{R}^{d \times r}$ an orthonormal basis of the vector space $\{\boldsymbol{\mu}_i\}_{i \in [k]}$ and $\mathbf{N} \in \mathbb{R}^{d \times n}$ an orthonormal basis of the null space of $\{\boldsymbol{\mu}_i\}_{i \in [k]}$. To show $\|\mathbf{x}_t - \boldsymbol{\mu}_i\|^2 > \left(\frac{\nu_0^2}{2} + \frac{\nu_{\max}^2}{2(1 - c_\nu)}\right) d$, it suffices to prove $\|\mathbf{n}_t\|^2 > \left(\frac{\nu_0^2}{2} + \frac{\nu_{\max}^2}{2(1 - c_\nu)}\right) d$. By Proposition 2, if $\mathbf{x}_0$ is initialized in the distribution $P^{(0)}$, i.e., $\mathbf{x}_0 \sim P^{(0)}$, since $\nu_0^2 > \frac{1}{1 - c_\nu}\nu_{\max}^2$, with probability at least $1 - \exp(-\Omega(d))$ we have

$$\|\mathbf{n}_0\|^2 \geq \left(\frac{3\nu_0^2}{4} + \frac{\nu_{\max}^2}{4(1 - c_\nu)}\right) d. \tag{13}$$

Then, conditioned on $\|\mathbf{n}_0\|^2 \geq \left(\frac{3\nu_0^2}{4} + \frac{\nu_{\max}^2}{4(1 - c_\nu)}\right) d$, the following proposition shows that $\|\mathbf{n}_t\|$ remains large with high probability.

**Proposition 4.** *Consider a distribution $P$ satisfying Assumption 2. We follow the Langevin dynamics for $T = \exp(\mathcal{O}(d))$ steps. Suppose that the initial sample satisfies $\|\mathbf{n}_0\|^2 \geq \left(\frac{3\nu_0^2}{4} + \frac{\nu_{\max}^2}{4(1 - c_\nu)}\right) d$, then with probability at least $1 - T \cdot \exp(-\Omega(d))$, we have that $\|\mathbf{n}_t\|^2 > \left(\frac{\nu_0^2}{2} + \frac{\nu_{\max}^2}{2(1 - c_\nu)}\right) d$ for all $t \in \{0\} \cup [T]$.*

*Proof of Proposition 4.* Firstly, by Lemma 3, if $\delta_t > \nu_0^2$, since $\nu_0^2 > \frac{\nu_{\max}^2}{1 - c_\nu}$, we similarly have that $\|\mathbf{n}_t\|^2 \geq \left(\frac{3\nu_0^2}{4} + \frac{\nu_{\max}^2}{4(1 - c_\nu)}\right) d$ with probability at least $1 - \exp(-\Omega(d))$ regardless of the previous state $\mathbf{x}_{t-1}$. We then consider the case when $\delta_t \leq \nu_0^2$. Intuitively, we aim to prove that the score function is close to $-\frac{\mathbf{x}}{\nu_0^2}$ when $\|\mathbf{n}\|^2 \geq \left(\frac{\nu_0^2}{2} + \frac{\nu_{\max}^2}{2(1 - c_\nu)}\right) d$. Towards this goal, we first show that $P^{(0)}(\mathbf{x})$ is exponentially larger than $P^{(i)}(\mathbf{x})$ for all $i \in [k]$ in the following lemma:

**Lemma 10.** *Suppose $P$ satisfies Assumption 2. Then for any $\|\mathbf{n}\|^2 \geq \left(\frac{\nu_0^2}{2} + \frac{\nu_{\max}^2}{2(1 - c_\nu)}\right) d$, we have $\frac{P^{(i)}(\mathbf{x})}{P^{(0)}(\mathbf{x})} \leq \exp(-\Omega(d))$ and $\frac{\|\nabla_\mathbf{x} P^{(i)}(\mathbf{x})\|}{P(\mathbf{x})} \leq \exp(-\Omega(d))$ for all $i \in [k]$.*

*Proof of Lemma 10.* We first give an upper bound on the sub-Gaussian probability density. For any vector $\mathbf{v} \in \mathbb{R}^d$, by considering some vector $\mathbf{m} \in \mathbb{R}^d$, from Markov's inequality and the definition in equation 4 we can bound

$$\mathbb{P}_{\mathbf{z} \sim P^{(i)}}\left(\mathbf{m}^T(\mathbf{z} - \boldsymbol{\mu}_i) \geq \mathbf{m}^T(\mathbf{v} - \boldsymbol{\mu}_i)\right) \leq \frac{\mathbb{E}_{\mathbf{z} \sim P^{(i)}}\left[\exp\left(\mathbf{m}^T(\mathbf{z} - \boldsymbol{\mu}_i)\right)\right]}{\exp\left(\mathbf{m}^T(\mathbf{v} - \boldsymbol{\mu}_i)\right)}$$

$$\leq \exp\left(\frac{\nu_i^2 \|\mathbf{m}\|^2}{2} - \mathbf{m}^T(\mathbf{v} - \boldsymbol{\mu}_i)\right).$$

Upon optimizing the last term at $\mathbf{m} = \frac{\mathbf{v} - \boldsymbol{\mu}_i}{\nu_i^2}$, we obtain

$$\mathbb{P}_{\mathbf{z} \sim P^{(i)}}\left((\mathbf{v} - \boldsymbol{\mu}_i)^T(\mathbf{v} - \mathbf{z}) \leq 0\right) \leq \exp\left(-\frac{\|\mathbf{v} - \boldsymbol{\mu}_i\|^2}{2\nu_i^2}\right). \tag{14}$$

Denote $\mathbb{B} := \left\{ \mathbf{z} : (\mathbf{v} - \boldsymbol{\mu}_i)^T(\mathbf{v} - \mathbf{z}) \le 0 \right\}$. To bound $\mathbb{P}_{\mathbf{z} \sim P^{(i)}}(\mathbf{z} \in \mathbb{B})$, we first note that

$$
\log P^{(i)}(\mathbf{v}) - \log P^{(i)}(\mathbf{z})
$$

$$
= \int_0^1 \langle \mathbf{v} - \mathbf{z}, \nabla \log P^{(i)}(\mathbf{v} + \lambda(\mathbf{z} - \mathbf{v})) \rangle \, \mathrm{d}\lambda
$$

$$
= \langle \mathbf{v} - \mathbf{z}, \nabla \log P^{(i)}(\mathbf{v}) \rangle + \int_0^1 \langle \mathbf{v} - \mathbf{z}, \nabla \log P^{(i)}(\mathbf{v} + \lambda(\mathbf{z} - \mathbf{v})) - \nabla \log P^{(i)}(\mathbf{v}) \rangle \, \mathrm{d}\lambda
$$

$$
\le \|\mathbf{v} - \mathbf{z}\| \left\| \nabla \log P^{(i)}(\mathbf{v}) \right\| + \int_0^1 \|\mathbf{v} - \mathbf{z}\| \left\| \nabla \log P^{(i)}(\mathbf{v} + \lambda(\mathbf{z} - \mathbf{v})) - \nabla \log P^{(i)}(\mathbf{v}) \right\| \, \mathrm{d}\lambda
$$

$$
\le \|\mathbf{v} - \mathbf{z}\| \cdot L_i \|\mathbf{v} - \boldsymbol{\mu}_i\| + \int_0^1 \|\mathbf{v} - \mathbf{z}\| \cdot L_i \|\lambda(\mathbf{z} - \mathbf{v})\| \, \mathrm{d}\lambda
$$

$$
\le \frac{L_i c_\nu}{2c_L} \|\mathbf{v} - \boldsymbol{\mu}_i\|^2 + \left( \frac{c_L + c_\nu}{2c_\nu} \right) L_i \|\mathbf{v} - \mathbf{z}\|^2 ,
$$

where the second last inequality follows from Assumption 2.ii. that $\nabla \log P^{(i)}(\boldsymbol{\mu}_i) = \mathbf{0}_d$ and Assumption 2.iii. that the score function $\nabla \log P^{(i)}$ is $L_i$-Lipschitz. Therefore we obtain

$$
\mathbb{P}_{\mathbf{z} \sim P^{(i)}}(\mathbf{z} \in \mathbb{B}) = \int_{\mathbf{z} \in \mathbb{B}} P^{(i)}(\mathbf{z}) \, \mathrm{d}\mathbf{z}
$$

$$
\ge \int_{\mathbf{z} \in \mathbb{B}} P^{(i)}(\mathbf{v}) \exp\left( -\frac{L_i c_\nu}{2c_L} \|\mathbf{v} - \boldsymbol{\mu}_i\|^2 - \frac{c_L + c_\nu}{2c_\nu} L_i \|\mathbf{v} - \mathbf{z}\|^2 \right) \, \mathrm{d}\mathbf{z}
$$

$$
= P^{(i)}(\mathbf{v}) \exp\left( -\frac{L_i c_\nu}{2c_L} \|\mathbf{v} - \boldsymbol{\mu}_i\|^2 \right) \int_{\mathbf{z} \in \mathbb{B}} \exp\left( -\frac{c_L + c_\nu}{2c_\nu} L_i \|\mathbf{v} - \mathbf{z}\|^2 \right) \, \mathrm{d}\mathbf{z}. \tag{15}
$$

By observing that $g : \mathbb{B} \to \left\{ \mathbf{z} : (\mathbf{v} - \boldsymbol{\mu}_i)^T(\mathbf{v} - \mathbf{z}) \ge 0 \right\}$ with $g(\mathbf{z}) = 2\mathbf{v} - \mathbf{z}$ is a bijection such that $\|\mathbf{v} - \mathbf{z}\| = \|\mathbf{v} - g(\mathbf{z})\|$ for any $\mathbf{z} \in \mathbb{B}$, we have

$$
\int_{\mathbf{z} \in \mathbb{B}} \exp\left( -\frac{c_L + c_\nu}{2c_\nu} L_i \|\mathbf{v} - \mathbf{z}\|^2 \right) \, \mathrm{d}\mathbf{z} = \frac{1}{2} \int_{\mathbf{z} \in \mathbb{R}^d} \exp\left( -\frac{c_L + c_\nu}{2c_\nu} L_i \|\mathbf{v} - \mathbf{z}\|^2 \right) \, \mathrm{d}\mathbf{z}
$$

$$
= \frac{1}{2} \left( \frac{2\pi c_\nu}{(c_L + c_\nu) L_i} \right)^{\frac{d}{2}}. \tag{16}
$$

Hence, by combining equation 14, equation 15, and equation 16, we obtain

$$
\exp\left( -\frac{\|\mathbf{v} - \boldsymbol{\mu}_i\|^2}{2\nu_i^2} \right) \ge \mathbb{P}_{\mathbf{z} \sim P^{(i)}}\left( (\mathbf{v} - \boldsymbol{\mu}_i)^T(\mathbf{v} - \mathbf{z}) \le 0 \right)
$$

$$
\ge P^{(i)}(\mathbf{v}) \exp\left( -\frac{L_i c_\nu}{2c_L} \|\mathbf{v} - \boldsymbol{\mu}_i\|^2 \right) \cdot \frac{1}{2} \left( \frac{2\pi c_\nu}{(c_L + c_\nu) L_i} \right)^{\frac{d}{2}}.
$$

By Assumption 2.iii. that $L_i \le \frac{c_L}{\nu_i^2}$ we obtain the following bound on the probability density:

$$
P^{(i)}(\mathbf{v}) \le 2 \left( \frac{2\pi c_\nu \nu_i^2}{(c_L + c_\nu) c_L} \right)^{-\frac{d}{2}} \exp\left( -\frac{1 - c_\nu}{2\nu_i^2} \|\mathbf{v} - \boldsymbol{\mu}_i\|^2 \right). \tag{17}
$$

Then we can bound the ratio of $P^{(i)}$ and $P^{(0)}$. For all $i \in [k]$, define $\rho_i(\mathbf{x}) := \frac{P^{(i)}(\mathbf{x})}{P^{(0)}(\mathbf{x})}$, then we have

$$\rho_i(\mathbf{x}) = \frac{P^{(i)}(\mathbf{x})}{P^{(0)}(\mathbf{x})} \leq \frac{2(2\pi c_\nu \nu_i^2/(c_L^2 + c_\nu c_L))^{-d/2} \exp\left(-(1-c_\nu)\|\mathbf{x} - \boldsymbol{\mu}_i\|^2 / 2\nu_i^2\right)}{(2\pi\nu_0^2)^{-d/2} \exp\left(-\|\mathbf{x}\|^2 / 2\nu_0^2\right)}$$

$$= 2\left(\frac{(c_L^2 + c_\nu c_L)\nu_0^2}{c_\nu \nu_i^2}\right)^{\frac{d}{2}} \exp\left(\frac{\|\mathbf{x}\|^2}{2\nu_0^2} - \frac{(1-c_\nu)\|\mathbf{x} - \boldsymbol{\mu}_i\|^2}{2\nu_i^2}\right)$$

$$= 2\left(\frac{(c_L^2 + c_\nu c_L)\nu_0^2}{c_\nu \nu_i^2}\right)^{\frac{d}{2}} \exp\left(\left(\frac{1}{2\nu_0^2} - \frac{1-c_\nu}{2\nu_i^2}\right)\|\mathbf{N}\mathbf{n}\|^2 + \left(\frac{\|\mathbf{R}\mathbf{r}\|^2}{2\nu_0^2} - \frac{(1-c_\nu)\|\mathbf{R}\mathbf{r} - \boldsymbol{\mu}_i\|^2}{2\nu_i^2}\right)\right)$$

$$= 2\left(\frac{(c_L^2 + c_\nu c_L)\nu_0^2}{c_\nu \nu_i^2}\right)^{\frac{d}{2}} \exp\left(\left(\frac{1}{2\nu_0^2} - \frac{1-c_\nu}{2\nu_i^2}\right)\|\mathbf{n}\|^2 + \left(\frac{\|\mathbf{r}\|^2}{2\nu_0^2} - \frac{(1-c_\nu)\|\mathbf{r} - \mathbf{R}^T\boldsymbol{\mu}_i\|^2}{2\nu_i^2}\right)\right),$$

where the last step follows from the definition that $\mathbf{R} \in \mathbb{R}^{d \times r}$ an orthogonal basis of the vector space $\{\boldsymbol{\mu}_i\}_{i \in [k]}$ and $\mathbf{N}^T\mathbf{N} = \boldsymbol{I}_n$. Since $\nu_i^2 < (1-c_\nu)\nu_0^2$, the quadratic term $\frac{\|\mathbf{r}\|^2}{2\nu_0^2} - \frac{(1-c_\nu)\|\mathbf{r} - \mathbf{R}^T\boldsymbol{\mu}_i\|^2}{2\nu_i^2}$ is maximized at $\mathbf{r} = \frac{(1-c_\nu)\nu_0^2 \mathbf{R}^T\boldsymbol{\mu}_i}{(1-c_\nu)\nu_0^2 - \nu_i^2}$. Therefore, we obtain

$$\frac{\|\mathbf{r}\|^2}{2\nu_0^2} - \frac{(1-c_\nu)\|\mathbf{r} - \mathbf{R}^T\boldsymbol{\mu}_i\|^2}{2\nu_i^2} \leq \frac{(1-c_\nu)\|\boldsymbol{\mu}_i\|^2}{2((1-c_\nu)\nu_0^2 - \nu_i^2)}.$$

Hence, for $\|\boldsymbol{\mu}_i - \boldsymbol{\mu}_0\|^2 \leq \frac{(1-c_\nu)\nu_0^2 - \nu_i^2}{2(1-c_\nu)}\left(\log \frac{c_\nu \nu_i^2}{(c_L^2 + c_\nu c_L)\nu_0^2} - \frac{\nu_i^2}{2(1-c_\nu)\nu_0^2} + \frac{(1-c_\nu)\nu_0^2}{2\nu_i^2}\right)d$ and $\|\mathbf{n}\|^2 \geq \left(\frac{\nu_0^2}{2} + \frac{\nu_{\max}^2}{2(1-c_\nu)}\right)d$, we have

$$\rho_i(\mathbf{x}) \leq 2\left(\frac{(c_L^2 + c_\nu c_L)\nu_0^2}{c_\nu \nu_i^2}\right)^{\frac{d}{2}} \exp\left(\left(\frac{1}{2\nu_0^2} - \frac{1-c_\nu}{2\nu_i^2}\right)\|\mathbf{n}\|^2 + \frac{(1-c_\nu)\|\boldsymbol{\mu}_i\|^2}{2((1-c_\nu)\nu_0^2 - \nu_i^2)}\right)$$

$$\leq 2\left(\frac{(c_L^2 + c_\nu c_L)\nu_0^2}{c_\nu \nu_i^2}\right)^{\frac{d}{2}} \exp\left(\left(\frac{1}{2\nu_0^2} - \frac{1-c_\nu}{2\nu_i^2}\right)\left(\frac{\nu_0^2}{2} + \frac{\nu_i^2}{2(1-c_\nu)}\right)d + \frac{(1-c_\nu)\|\boldsymbol{\mu}_i\|^2}{2((1-c_\nu)\nu_0^2 - \nu_i^2)}\right)$$

$$= 2\exp\left(-\left(\log \frac{c_\nu \nu_i^2}{(c_L^2 + c_\nu c_L)\nu_0^2} - \frac{\nu_i^2}{2(1-c_\nu)\nu_0^2} + \frac{(1-c_\nu)\nu_0^2}{2\nu_i^2}\right)\frac{d}{2} + \frac{(1-c_\nu)\|\boldsymbol{\mu}_i\|^2}{2((1-c_\nu)\nu_0^2 - \nu_i^2)}\right)$$

$$\leq 2\exp\left(-\left(\log \frac{c_\nu \nu_i^2}{(c_L^2 + c_\nu c_L)\nu_0^2} - \frac{\nu_i^2}{2(1-c_\nu)\nu_0^2} + \frac{(1-c_\nu)\nu_0^2}{2\nu_i^2}\right)\frac{d}{4}\right).$$

From Lemma 9, we obtain $\rho_i(\mathbf{x}) \leq \exp(-\Omega(d))$.

To show $\frac{\|\nabla_\mathbf{x} P^{(i)}(\mathbf{x})\|}{P(\mathbf{x})} \leq \exp(-\Omega(d))$, from Assumptions 2.ii. and 2.iii. we have

$$\left\|\frac{\nabla_\mathbf{x} P^{(i)}(\mathbf{x})}{P^{(i)}(\mathbf{x})}\right\| = \left\|\frac{\nabla_\mathbf{x} P^{(i)}(\mathbf{x})}{P^{(i)}(\mathbf{x})} - \frac{\nabla_\mathbf{x} P^{(i)}(\boldsymbol{\mu}_i)}{P^{(i)}(\boldsymbol{\mu}_i)}\right\| = \left\|\nabla_\mathbf{x} \log P^{(i)}(\mathbf{x}) - \nabla_\mathbf{x} \log P^{(i)}(\boldsymbol{\mu}_i)\right\|$$

$$\leq L_i \|\mathbf{x} - \boldsymbol{\mu}_i\| \leq \frac{c_L}{\nu_i^2}\|\mathbf{x} - \boldsymbol{\mu}_i\|.$$

Therefore, we can bound $\frac{\|\nabla_\mathbf{x} P^{(i)}(\mathbf{x})\|}{P(\mathbf{x})} \leq \frac{c_L}{\nu_i^2}\rho_i(\mathbf{x})\|\mathbf{x} - \boldsymbol{\mu}_i\|$. When $\|\mathbf{x} - \boldsymbol{\mu}_i\| = \exp(o(d))$ is small, by $\rho_i(\mathbf{x}) \leq \exp(-\Omega(d))$ we directly have $\frac{\|\nabla_\mathbf{x} P^{(i)}(\mathbf{x})\|}{P(\mathbf{x})} \leq \exp(-\Omega(d))$. When $\|\mathbf{x} - \boldsymbol{\mu}_i\| = \exp(\Omega(d))$ is exceedingly large, from equation 17 we have

$$\frac{\|\nabla_\mathbf{x} P^{(i)}(\mathbf{x})\|}{P(\mathbf{x})} \leq \frac{2c_L}{\nu_i^2}\left(\frac{(c_L^2 + c_\nu c_L)\nu_0^2}{c_\nu \nu_i^2}\right)^{\frac{d}{2}} \exp\left(\frac{\|\mathbf{x}\|^2}{2\nu_0^2} - \frac{(1-c_\nu)\|\mathbf{x} - \boldsymbol{\mu}_i\|^2}{2\nu_i^2}\right)\|\mathbf{x} - \boldsymbol{\mu}_i\|.$$

Since $\nu_0^2 > \frac{\nu_i^2}{1-c_\nu}$, when $\|\mathbf{x} - \boldsymbol{\mu}_i\| = \exp(\Omega(d)) \gg \|\boldsymbol{\mu}_i\|$ we have

$$\exp\left(\frac{\|\mathbf{x}\|^2}{2\nu_0^2} - \frac{(1-c_\nu)\|\mathbf{x}-\boldsymbol{\mu}_i\|^2}{2\nu_i^2}\right) = \exp(-\Omega(\|\mathbf{x}-\boldsymbol{\mu}_i\|^2)).$$

Therefore $\frac{\|\nabla_{\mathbf{x}}P^{(i)}(\mathbf{x})\|}{P(\mathbf{x})} \leq \exp(-\Omega(d))$. Thus we complete the proof of Lemma 10. $\qquad\square$

Similar to Lemma 5, the following lemma proves that when the previous state $\mathbf{n}_{t-1}$ is far from a mode, a single step of Langevin dynamics with bounded step size is not enough to find the mode.

**Lemma 11.** *Suppose $\delta_t \leq \nu_0^2$ and $\|\mathbf{n}_{t-1}\|^2 > 36\nu_0^2 d$, then we have $\|\mathbf{n}_t\|^2 \geq \nu_0^2 d$ with probability at least $1 - \exp(-\Omega(d))$.*

*Proof of Lemma 11.* For simplicity, denote $\mathbf{v} := \mathbf{n}_{t-1} + \frac{\delta_t}{2}\mathbf{N}^T\nabla_{\mathbf{x}}\log P(\mathbf{x}_{t-1})$. Since $P = \sum_{i=0}^k w_i P^{(i)}$ and $P^{(0)} = \mathcal{N}(\boldsymbol{\mu}_0, \nu_0^2 \boldsymbol{I}_d)$, the score function can be written as

$$\begin{aligned}
\nabla_{\mathbf{x}}\log P(\mathbf{x}) = \frac{\nabla_{\mathbf{x}}P(\mathbf{x})}{P(\mathbf{x})} &= \frac{\nabla_{\mathbf{x}}w_0 P^{(0)}(\mathbf{x})}{P(\mathbf{x})} + \sum_{i\in[k]}\frac{\nabla_{\mathbf{x}}w_i P^{(i)}(\mathbf{x})}{P(\mathbf{x})} \\
&= -\frac{w_0 P^{(0)}(\mathbf{x})}{P(\mathbf{x})}\cdot\frac{\mathbf{x}}{\nu_0^2} + \sum_{i\in[k]}\frac{w_i\nabla_{\mathbf{x}}P^{(i)}(\mathbf{x})}{P(\mathbf{x})} \\
&= -\frac{\mathbf{x}}{\nu_0^2} + \sum_{i\in[k]}\frac{w_i P^{(i)}(\mathbf{x})}{P(\mathbf{x})}\cdot\frac{\mathbf{x}}{\nu_0^2} + \sum_{i\in[k]}\frac{w_i\nabla_{\mathbf{x}}P^{(i)}(\mathbf{x})}{P(\mathbf{x})}. \quad (18)
\end{aligned}$$

For $\|\mathbf{n}_{t-1}\|^2 > 36\nu_0^2 d$ by Lemma 10 we have $\frac{\|\nabla_{\mathbf{x}}P^{(i)}(\mathbf{x}_{t-1})\|}{P(\mathbf{x}_{t-1})} \leq \exp(-\Omega(d))$. Since $\delta_t \leq \nu_0^2$, we can bound the norm of $\mathbf{v}$ by

$$\begin{aligned}
\|\mathbf{v}\| &= \left\|\mathbf{n}_{t-1} + \frac{\delta_t}{2}\mathbf{N}^T\nabla_{\mathbf{x}}\log P(\mathbf{x}_{t-1})\right\| \\
&= \left\|\mathbf{n}_{t-1} - \frac{\delta_t}{2\nu_0^2}\mathbf{n}_{t-1} + \sum_{i\in[k]}\frac{w_i\delta_t}{2\nu_0^2}\frac{P^{(i)}(\mathbf{x}_{t-1})}{P(\mathbf{x}_{t-1})}\mathbf{n}_{t-1} + \sum_{i\in[k]}\frac{w_i\delta_t}{2}\frac{\mathbf{N}^T\nabla_{\mathbf{x}}P^{(i)}(\mathbf{x}_{t-1})}{P(\mathbf{x}_{t-1})}\right\| \\
&\geq \left\|\left(1 - \frac{\delta_t}{2\nu_0^2} + \sum_{i\in[k]}\frac{w_i\delta_t}{2\nu_0^2}\frac{P^{(i)}(\mathbf{x}_{t-1})}{P(\mathbf{x}_{t-1})}\right)\mathbf{n}_{t-1}\right\| - \sum_{i\in[k]}\frac{w_i\delta_t}{2}\frac{\|\nabla_{\mathbf{x}}P^{(i)}(\mathbf{x}_{t-1})\|}{P(\mathbf{x}_{t-1})} \\
&\geq \frac{1}{2}\|\mathbf{n}_{t-1}\| - \sum_{i\in[k]}\frac{w_i\delta_t}{2}\exp(-\Omega(d)) \\
&> 2\nu_0\sqrt{d}.
\end{aligned}$$

On the other hand, from $\boldsymbol{\epsilon}_t^{(\mathbf{n})} \sim \mathcal{N}(\mathbf{0}_n, \boldsymbol{I}_n)$ we know $\frac{\langle\mathbf{v},\boldsymbol{\epsilon}_t^{(\mathbf{n})}\rangle}{\|\mathbf{v}\|} \sim \mathcal{N}(0,1)$ for any fixed $\mathbf{v} \neq \mathbf{0}_n$, hence by Lemma 2 we have

$$\mathbb{P}\left(\frac{\langle\mathbf{v},\boldsymbol{\epsilon}_t^{(\mathbf{n})}\rangle}{\|\mathbf{v}\|} \geq \frac{\sqrt{d}}{4}\right) = \mathbb{P}\left(\frac{\langle\mathbf{v},\boldsymbol{\epsilon}_t^{(\mathbf{n})}\rangle}{\|\mathbf{v}\|} \leq -\frac{\sqrt{d}}{4}\right) \leq \frac{4}{\sqrt{2\pi d}}\exp\left(-\frac{d}{32}\right)$$

Combining the above inequalities gives

$$\|\mathbf{n}_t\|^2 = \left\|\mathbf{v} + \sqrt{\delta_t}\boldsymbol{\epsilon}_t^{(\mathbf{n})}\right\|^2 \geq \|\mathbf{v}\|^2 - 2\nu_0|\langle\mathbf{v},\boldsymbol{\epsilon}_t^{(\mathbf{n})}\rangle| \geq \|\mathbf{v}\|^2 - \frac{\nu_0\sqrt{d}}{2}\|\mathbf{v}\| > \nu_0^2 d$$

with probability at least $1 - \frac{8}{\sqrt{2\pi d}}\exp\left(-\frac{d}{32}\right) = 1 - \exp(-\Omega(d))$. This proves Lemma 11. $\qquad\square$

When $\|\mathbf{n}_{t-1}\|^2 \leq 36\nu_0^2 d$, similar to Theorem 1, we consider a surrogate recursion $\hat{\mathbf{n}}_t$ such that $\hat{\mathbf{n}}_0 = \mathbf{n}_0$ and for all $t \geq 1$,

$$\hat{\mathbf{n}}_t = \hat{\mathbf{n}}_{t-1} - \frac{\delta_t}{2\nu_0^2}\hat{\mathbf{n}}_{t-1} + \sqrt{\delta_t}\boldsymbol{\epsilon}_t^{(\mathbf{n})}. \tag{19}$$

The following Lemma shows that $\hat{\mathbf{n}}_t$ is sufficiently close to the original recursion $\mathbf{n}_t$.

**Lemma 12.** *For any $t \geq 1$, given that for all $j \in [t]$, $\delta_j \leq \nu_0^2$ and $\left(\frac{\nu_0^2}{2} + \frac{\nu_{\max}^2}{2(1-c_\nu)}\right)d \leq \|\mathbf{n}_{j-1}\|^2 \leq 36\nu_0^2 d$, if $\boldsymbol{\mu}_i$ satisfies Assumption 2.v. for all $i \in [k]$, we have $\|\hat{\mathbf{n}}_t - \mathbf{n}_t\| \leq \frac{t}{\exp(\Omega(d))}\sqrt{d}$.*

*Proof of Lemma 12.* By equation 18 we have that for all $j \in [t]$,

$$\|\hat{\mathbf{n}}_j - \mathbf{n}_j\| = \left\|\hat{\mathbf{n}}_{j-1} - \mathbf{n}_{j-1} - \frac{\delta_j}{2\nu_0^2}\hat{\mathbf{n}}_{j-1} - \frac{\delta_j}{2}\mathbf{N}^T \nabla_{\mathbf{x}} \log P(\mathbf{x}_{j-1})\right\|$$

$$= \left\|\hat{\mathbf{n}}_{j-1} - \mathbf{n}_{j-1} - \sum_{i \in [k]} \frac{w_i P^{(i)}(\mathbf{x}_{j-1})}{\nu_0^2 P(\mathbf{x}_{j-1})}\mathbf{n}_{j-1} - \sum_{i \in [k]} \frac{w_i \mathbf{N}^T \nabla_{\mathbf{x}} P^{(i)}(\mathbf{x}_{j-1})}{P(\mathbf{x}_{j-1})}\right\|$$

$$\leq \|\hat{\mathbf{n}}_{j-1} - \mathbf{n}_{j-1}\| + \sum_{i \in [k]} \frac{w_i P^{(i)}(\mathbf{x}_{j-1})}{\nu_0^2 P(\mathbf{x}_{j-1})}\|\mathbf{n}_{j-1}\| + \sum_{i \in [k]} \frac{w_i \left\|\nabla_{\mathbf{x}} P^{(i)}(\mathbf{x}_{j-1})\right\|}{P(\mathbf{x}_{j-1})}.$$

By Lemma 10, we have $\frac{P^{(i)}(\mathbf{x}_{j-1})}{P^{(0)}(\mathbf{x}_{j-1})} \leq \exp(-\Omega(d))$ and $\frac{\left\|\nabla_{\mathbf{x}} P^{(i)}(\mathbf{x}_{j-1})\right\|}{P(\mathbf{x}_{j-1})} \leq \exp(-\Omega(d))$ for all $i \in [k]$, hence from $\|\mathbf{n}_{j-1}\| \leq 6\nu_0\sqrt{d}$ we obtain a recursive bound

$$\|\hat{\mathbf{n}}_j - \mathbf{n}_j\| \leq \|\hat{\mathbf{n}}_{j-1} - \mathbf{n}_{j-1}\| + \frac{1}{\exp(\Omega(d))}\sqrt{d}.$$

Finally, by $\hat{\mathbf{n}}_0 = \mathbf{n}_0$, we have

$$\|\hat{\mathbf{n}}_t - \mathbf{n}_t\| = \sum_{j \in [t]} \left(\|\hat{\mathbf{n}}_j - \mathbf{n}_j\| - \|\hat{\mathbf{n}}_{j-1} - \mathbf{n}_{j-1}\|\right) \leq \frac{t}{\exp(\Omega(d))}\sqrt{d}.$$

Hence we obtain Lemma 12. $\qquad\square$

Armed with the above lemmas, we are now ready to establish Proposition 4 by induction. Please note that we also apply some lemmas from the proof of Theorem 1 by substituting $\nu_{\max}^2$ with $\frac{\nu_{\max}^2}{1-c_\nu}$. Suppose the theorem holds for all $T$ values of $1, \cdots, T-1$. We consider the following 3 cases:

- If there exists some $t \in [T]$ such that $\delta_t > \nu_0^2$, by Lemma 3 we know that with probability at least $1 - \exp(-\Omega(d))$, we have $\|\mathbf{n}_t\|^2 \geq \left(\frac{3\nu_0^2}{4} + \frac{\nu_{\max}^2}{4(1-c_\nu)}\right)d$, thus the problem reduces to the two sub-arrays $\mathbf{n}_0, \cdots, \mathbf{n}_{t-1}$ and $\mathbf{n}_t, \cdots, \mathbf{n}_T$, which can be solved by induction.

- Suppose $\delta_t \leq \nu_0^2$ for all $t \in [T]$. If there exists some $t \in [T]$ such that $\|\mathbf{n}_{t-1}\|^2 > 36\nu_0^2 d$, by Lemma 11 we know that with probability at least $1 - \exp(-\Omega(d))$, we have $\|\mathbf{n}_t\|^2 \geq \nu_0^2 d > \left(\frac{3\nu_0^2}{4} + \frac{\nu_{\max}^2}{4(1-c_\nu)}\right)d$, thus the problem similarly reduces to the two sub-arrays $\mathbf{n}_0, \cdots, \mathbf{n}_{t-1}$ and $\mathbf{n}_t, \cdots, \mathbf{n}_T$, which can be solved by induction.

- Suppose $\delta_t \leq \nu_0^2$ and $\|\mathbf{n}_{t-1}\|^2 \leq 36\nu_0^2 d$ for all $t \in [T]$. Conditioned on $\|\mathbf{n}_{t-1}\|^2 > \left(\frac{\nu_0^2}{2} + \frac{\nu_{\max}^2}{2(1-c_\nu)}\right)d$ for all $t \in [T]$, by Lemma 12 we have that for $T = \exp(\mathcal{O}(d))$,

$$\|\hat{\mathbf{n}}_T - \mathbf{n}_T\| < \left(\sqrt{\frac{5\nu_0^2}{8} + \frac{3\nu_{\max}^2}{8(1-c_\nu)}} - \sqrt{\frac{\nu_0^2}{2} + \frac{\nu_{\max}^2}{2(1-c_\nu)}}\right)\sqrt{d}.$$

By Lemma 8 we have that with probability at least $1 - \exp(-\Omega(d))$,

$$\|\hat{\mathbf{n}}_T\|^2 \geq \left(\frac{5\nu_0^2}{8} + \frac{3\nu_{\max}^2}{8(1-c_\nu)}\right)d.$$

Combining the two inequalities implies the desired bound

$$\|\mathbf{n}_T\| \geq \|\hat{\mathbf{n}}_T\| - \|\hat{\mathbf{n}}_T - \mathbf{n}_T\| > \sqrt{\left(\frac{\nu_0^2}{2} + \frac{\nu_{\max}^2}{2(1 - c_\nu)}\right) d}.$$

Hence by induction we obtain $\|\mathbf{n}_t\|^2 > \left(\frac{\nu_0^2}{2} + \frac{\nu_{\max}^2}{2(1-c_\nu)}\right) d$ for all $t \in [T]$ with probability at least

$$(1 - (T - 1) \exp(-\Omega(d))) \cdot (1 - \exp(-\Omega(d))) \geq 1 - T \exp(-\Omega(d)).$$

Therefore we complete the proof of Proposition 4. $\qquad\square$

Finally, combining equation 13 and Proposition 4 finishes the proof of Theorem 2.

## B    THEORETICAL ANALYSIS ON ANNEALED LANGEVIN DYNAMICS

### B.1    ANNEALED LANGEVIN DYNAMICS IN GAUSSIAN MIXTURES

In the following Theorem 3 we extend the result in Theorem 1 to annealed Langevin dynamics *with bounded noise levels*.

**Theorem 3.** *Consider a data distribution $P$ satisfying Assumption 1. We follow annealed Langevin dynamics for $T = \exp(\mathcal{O}(d))$ steps with noise levels $c_\sigma \geq \sigma_0 \geq \cdots \geq \sigma_T \geq 0$ for constant $c_\sigma > 0$. In addition, assume for all $i \in [k]$, $\|\boldsymbol{\mu}_i - \boldsymbol{\mu}_0\|^2 \leq \frac{\nu_0^2 - \nu_i^2}{2} \left(\log\left(\frac{\nu_i^2 + c_\sigma^2}{\nu_0^2 + c_\sigma^2}\right) - \frac{\nu_i^2 + c_\sigma^2}{2\nu_0^2 + c_\sigma^2} + \frac{\nu_0^2 + c_\sigma^2}{2\nu_i^2 + c_\sigma^2}\right) d$. Suppose that the sample is initialized in $P_{\sigma_0}^{(0)}$, then with probability at least $1 - T \cdot \exp(-\Omega(d))$, we have $\|\mathbf{x}_t - \boldsymbol{\mu}_i\|^2 > \frac{\nu_0^2 + \nu_{\max}^2 + 2\sigma_t^2}{2} d$ for all $t \in \{0\} \cup [T]$ and $i \in [k]$.*

*Proof of Theorem 3.* From equation 2 we note that the perturbed distribution is the convolution of the original distribution and a Gaussian random variable, i.e., for random variables $\mathbf{z} \sim p$ and $\mathbf{t} \sim \mathcal{N}(\mathbf{0}_d, \boldsymbol{I}_d)$, their sum $\mathbf{z} + \mathbf{t} \sim p_\sigma$ follows the perturbed distribution with noise level $\sigma$. Therefore, a perturbed (sub)Gaussian distribution remains (sub)Gaussian. We formalize this property in Proposition 5.

**Proposition 5.** *Suppose the perturbed distribution of a $d$-dimensional probability distribution $p$ with noise level $\sigma$ is $p_\sigma$, then the mean of the perturbed distribution is the same as the original distribution, i.e., $\mathbb{E}_{\mathbf{z} \sim p_\sigma}[\mathbf{z}] = \mathbb{E}_{\mathbf{z} \sim p}[\mathbf{z}]$. If $p = \mathcal{N}(\boldsymbol{\mu}, \boldsymbol{\Sigma})$ is a Gaussian distribution, $p_\sigma = \mathcal{N}(\boldsymbol{\mu}, \boldsymbol{\Sigma} + \sigma^2 \boldsymbol{I}_d)$ is also a Gaussian distribution. If $p$ is a sub-Gaussian distribution with parameter $\nu^2$, $p_\sigma$ is a sub-Gaussian distribution with parameter $(\nu^2 + \sigma^2)$.*

*Proof of Proposition 5.* By the definition in equation 2, we have

$$p_\sigma(\mathbf{z}) = \int p(\mathbf{t}) \mathcal{N}(\mathbf{z} \mid \mathbf{t}, \sigma^2 \boldsymbol{I}_d) \, \mathrm{d}\mathbf{t} = \int p(\mathbf{t}) \mathcal{N}(\mathbf{z} - \mathbf{t} \mid \mathbf{0}_d, \sigma^2 \boldsymbol{I}_d) \, \mathrm{d}\mathbf{t}.$$

For random variables $\mathbf{t} \sim p$ and $\mathbf{y} \sim \mathcal{N}(\mathbf{0}_d, \boldsymbol{I}_d)$, their sum $\mathbf{z} = \mathbf{t} + \mathbf{y} \sim p_\sigma$ follows the perturbed distribution with noise level $\sigma$. Therefore,

$$\mathbb{E}_{\mathbf{z} \sim p_\sigma}[\mathbf{z}] = \mathbb{E}_{(\mathbf{t}+\mathbf{y}) \sim p_\sigma}[\mathbf{t} + \mathbf{y}] = \mathbb{E}_{\mathbf{t} \sim p}[\mathbf{t}] + \mathbb{E}_{\mathbf{y} \sim \mathcal{N}(\mathbf{0}_d, \boldsymbol{I}_d)}[\mathbf{y}] = \mathbb{E}_{\mathbf{t} \sim p}[\mathbf{t}].$$

If $\mathbf{t} \sim p = \mathcal{N}(\boldsymbol{\mu}, \boldsymbol{\Sigma})$ follows a Gaussian distribution, we have $\mathbf{z} = \mathbf{t} + \mathbf{y} \sim p_\sigma = \mathcal{N}(\boldsymbol{\mu}, \boldsymbol{\Sigma} + \sigma^2 \boldsymbol{I}_d)$. If $p$ is a sub-Gaussian distribution with parameter $\nu^2$, we have $\mathbf{z} = \mathbf{t} + \mathbf{y} \sim p_\sigma$ is a sub-Gaussian distribution with parameter $(\nu^2 + \sigma^2)$. Hence we obtain Proposition 5. $\qquad\square$

To establish Theorem 3, we first note from Proposition 5 that perturbing a Gaussian distribution $\mathcal{N}(\boldsymbol{\mu}, \nu^2 \boldsymbol{I}_d)$ with noise level $\sigma$ results in a Gaussian distribution $\mathcal{N}(\boldsymbol{\mu}, (\nu^2 + \sigma^2) \boldsymbol{I}_d)$. Therefore, for a Gaussian mixture $P = \sum_{i=0}^k w_i P^{(i)} = \sum_{i=0}^k w_i \mathcal{N}(\boldsymbol{\mu}_i, \nu_i^2 \boldsymbol{I}_d)$, the perturbed distribution of noise level $\sigma$ is

$$P_\sigma = \sum_{i=0}^k w_i \mathcal{N}(\boldsymbol{\mu}_i, (\nu_i^2 + \sigma^2) \boldsymbol{I}_d).$$

Similar to the proof of Theorem 1, we decompose

$$\mathbf{x}_t = \mathbf{R}\mathbf{r}_t + \mathbf{N}\mathbf{n}_t, \text{ and } \boldsymbol{\epsilon}_t = \mathbf{R}\boldsymbol{\epsilon}_t^{(\mathbf{r})} + \mathbf{N}\boldsymbol{\epsilon}_t^{(\mathbf{n})},$$

where $\mathbf{R} \in \mathbb{R}^{d \times r}$ an orthonormal basis of the vector space $\{\boldsymbol{\mu}_i\}_{i \in [k]}$ and $\mathbf{N} \in \mathbb{R}^{d \times n}$ an orthonormal basis of the null space of $\{\boldsymbol{\mu}_i\}_{i \in [k]}$. Now, we prove Theorem 3 by applying the techniques developed in Appendix A.1 via substituting $\nu^2$ with $\nu^2 + \sigma_t^2$ at time step $t$.

First, by Proposition 2, suppose that the sample is initialized in the distribution $P_{\sigma_0}^{(0)}$, then with probability at least $1 - \exp(-\Omega(d))$, we have

$$\|\mathbf{n}_0\|^2 \geq \frac{3(\nu_0^2 + \sigma_0^2) + (\nu_{\max}^2 + \sigma_0^2)}{4}d = \frac{3\nu_0^2 + \nu_{\max}^2 + 4\sigma_0^2}{4}d. \tag{20}$$

Then, with the assumption that the initialization satisfies $\|\mathbf{n}_0\|^2 \geq \frac{3\nu_0^2 + \nu_{\max}^2 + 4\sigma_0^2}{4}d$, the following proposition similar to Proposition 3 shows that $\|\mathbf{n}_t\|$ remains large with high probability.

**Proposition 6.** *Consider a data distribution $P$ satisfies the constraints specified in Theorem 3. We follow annealed Langevin dynamics for $T = \exp(\mathcal{O}(d))$ steps with noise level $c_\sigma \geq \sigma_0 \geq \sigma_1 \geq \sigma_2 \geq \cdots \geq \sigma_T \geq 0$ for some constant $c_\sigma > 0$. Suppose that the initial sample satisfies $\|\mathbf{n}_0\|^2 \geq \frac{3\nu_0^2 + \nu_{\max}^2 + 4\sigma_0^2}{4}d$, then with probability at least $1 - T \cdot \exp(-\Omega(d))$, we have that $\|\mathbf{n}_t\|^2 > \frac{\nu_0^2 + \nu_{\max}^2 + 2\sigma_t^2}{2}d$ for all $t \in \{0\} \cup [T]$.*

*Proof of Proposition 6.* We prove Proposition 6 by induction. Suppose the theorem holds for all $T$ values of $1, \cdots, T - 1$. We consider the following 3 cases:

- If there exists some $t \in [T]$ such that $\delta_t > \nu_0^2 + \sigma_t^2$, by Lemma 3 we know that with probability at least $1 - \exp(-\Omega(d))$, we have $\|\mathbf{n}_t\|^2 \geq \frac{3(\nu_0^2 + \sigma_t^2) + (\nu_{\max}^2 + \sigma_t^2)}{4}d = \frac{3\nu_0^2 + \nu_{\max}^2 + 4\sigma_t^2}{4}d$, thus the problem reduces to the two sub-arrays $\mathbf{n}_0, \cdots, \mathbf{n}_{t-1}$ and $\mathbf{n}_t, \cdots, \mathbf{n}_T$, which can be solved by induction.

- Suppose $\delta_t \leq \nu_0^2 + \sigma_t^2$ for all $t \in [T]$. If there exists some $t \in [T]$ such that $\|\mathbf{n}_{t-1}\|^2 > 36(\nu_0^2 + \sigma_{t-1}^2)d \geq 36(\nu_0^2 + \sigma_t^2)d$, by Lemma 5 we know that with probability at least $1 - \exp(-\Omega(d))$, we have $\|\mathbf{n}_t\|^2 \geq (\nu_0^2 + \sigma_t^2)d > \frac{3\nu_0^2 + \nu_{\max}^2 + 4\sigma_t^2}{4}d$, thus the problem similarly reduces to the two sub-arrays $\mathbf{n}_0, \cdots, \mathbf{n}_{t-1}$ and $\mathbf{n}_t, \cdots, \mathbf{n}_T$, which can be solved by induction.

- Suppose $\delta_t \leq \nu_0^2 + \sigma_t^2$ and $\|\mathbf{n}_{t-1}\|^2 \leq 36(\nu_0^2 + \sigma_{t-1}^2)d$ for all $t \in [T]$. Consider a surrogate sequence $\hat{\mathbf{n}}_t$ such that $\hat{\mathbf{n}}_0 = \mathbf{n}_0$ and for all $t \geq 1$,

$$\hat{\mathbf{n}}_t = \hat{\mathbf{n}}_{t-1} - \frac{\delta_t}{2\nu_0^2 + 2\sigma_t^2}\hat{\mathbf{n}}_{t-1} + \sqrt{\delta_t}\boldsymbol{\epsilon}_t^{(\mathbf{n})}.$$

Since $\nu_0 > \nu_i$ and $c_\sigma \geq \sigma_t$ for all $t \in \{0\} \cup [T]$, we have $\frac{\nu_i^2 + c_\sigma^2}{\nu_0^2 + c_\sigma^2} \geq \frac{\nu_i^2 + \sigma_t^2}{\nu_0^2 + \sigma_t^2}$. Notice that for function $f(z) = \log z - \frac{z}{2} + \frac{1}{2z}$, we have $\frac{\mathrm{d}}{\mathrm{d}z}f(z) = \frac{1}{z} - \frac{1}{2} - \frac{1}{2z^2} = -\frac{1}{2}\left(\frac{1}{z} - 1\right)^2 \leq 0$. Thus, by the assumption

$$\|\boldsymbol{\mu}_i - \boldsymbol{\mu}_0\|^2 \leq \frac{\nu_0^2 - \nu_i^2}{2}\left(\log\left(\frac{\nu_i^2 + c_\sigma^2}{\nu_0^2 + c_\sigma^2}\right) - \frac{\nu_i^2 + c_\sigma^2}{2\nu_0^2 + c_\sigma^2} + \frac{\nu_0^2 + c_\sigma^2}{2\nu_i^2 + c_\sigma^2}\right)d,$$

we have that for all $t \in [T]$,

$$\|\boldsymbol{\mu}_i - \boldsymbol{\mu}_0\|^2 \leq \frac{\nu_0^2 - \nu_i^2}{2}\left(\log\left(\frac{\nu_i^2 + \sigma_t^2}{\nu_0^2 + \sigma_t^2}\right) - \frac{\nu_i^2 + \sigma_t^2}{2\nu_0^2 + \sigma_t^2} + \frac{\nu_0^2 + \sigma_t^2}{2\nu_i^2 + \sigma_t^2}\right)d.$$

Conditioned on $\|\mathbf{n}_{t-1}\|^2 > \frac{\nu_0^2 + \nu_{\max}^2 + 2\sigma_{t-1}^2}{2}d$ for all $t \in [T]$, by Lemma 6 we have that for $T = \exp(\mathcal{O}(d))$,

$$\|\hat{\mathbf{n}}_T - \mathbf{n}_T\| < \left(\sqrt{\frac{5\nu_0^2 + 3\nu_{\max}^2 + 8\sigma_T^2}{8}} - \sqrt{\frac{\nu_0^2 + \nu_{\max}^2 + 2\sigma_T^2}{2}}\right)\sqrt{d}.$$

By Lemma 8 we have that with probability at least $1 - \exp(-\Omega(d))$,

$$\|\hat{\mathbf{n}}_T\|^2 \geq \frac{5\nu_0^2 + 3\nu_{\max}^2 + 8\sigma_T^2}{8} d.$$

Combining the two inequalities implies the desired bound

$$\|\mathbf{n}_T\| \geq \|\hat{\mathbf{n}}_T\| - \|\hat{\mathbf{n}}_T - \mathbf{n}_T\| > \sqrt{\frac{\nu_0^2 + \nu_{\max}^2 + 2\sigma_T^2}{2}} d.$$

Hence by induction we obtain $\|\mathbf{n}_t\|^2 > \frac{\nu_0^2 + \nu_{\max}^2 + 2\sigma_t^2}{2} d$ for all $t \in \{0\} \cup [T]$ with probability at least

$$(1 - (T - 1)\exp(-\Omega(d))) \cdot (1 - \exp(-\Omega(d))) \geq 1 - T\exp(-\Omega(d)).$$

Therefore we complete the proof of Proposition 6. $\qquad\square$

Finally, combining equation 20 and Proposition 6 finishes the proof of Theorem 3. $\qquad\square$

### B.2 ANNEALED LANGEVIN DYNAMICS IN SUB-GAUSSIAN MIXTURES

Finally, we slightly modify Assumption 2 and extend our results to annealed Langevin dynamics (with bounded noise levels) under sub-Gaussian mixtures in Theorem 4.

**Assumption 3.** *Consider a data distribution $P := \sum_{i=0}^k w_i P^{(i)}$ as a mixture of sub-Gaussian distributions, where $1 \leq k = o(d)$ and $w_i > 0$ is a positive constant such that $\sum_{i=0}^k w_i = 1$. Suppose that $P^{(0)} = \mathcal{N}(\boldsymbol{\mu}_0, \nu_0^2 \mathbf{I}_d)$ is Gaussian and for all $i \in [k]$, $P^{(i)}$ satisfies*

*i. $P^{(i)}$ is a sub-Gaussian distribution of mean $\boldsymbol{\mu}_i$ with parameter $\nu_i^2$,*

*ii. $P^{(i)}$ is differentiable and $\nabla P_{\sigma_t}^{(i)}(\boldsymbol{\mu}_i) = \mathbf{0}_d$ for all $t \in \{0\} \cup [T]$,*

*iii. for all $t \in \{0\} \cup [T]$, the score function of $P_{\sigma_t}^{(i)}$ is $L_{i,t}$-Lipschitz such that $L_{i,t} \leq \frac{c_L}{\nu_i^2 + \sigma_t^2}$ for some constant $c_L > 0$,*

*iv. $\nu_0^2 > \max\left\{1, \frac{4(c_L^2 + c_\nu c_L)}{c_\nu(1-c_\nu)}\right\} \frac{\nu_{\max}^2 + c_\sigma^2}{1-c_\nu} - c_\sigma^2$ for constant $c_\nu \in (0,1)$, where $\nu_{\max} := \max_{i \in [k]} \nu_i$,*

*v. $\|\boldsymbol{\mu}_i - \boldsymbol{\mu}_0\|^2 \leq \frac{(1-c_\nu)\nu_0^2 - \nu_i^2 - c_\nu c_\sigma^2}{2(1-c_\nu)} \left(\log \frac{c_\nu(\nu_i^2 + c_\sigma^2)}{(c_L^2 + c_\nu c_L)(\nu_0^2 + c_\sigma^2)} - \frac{(\nu_i^2 + c_\sigma^2)}{2(1-c_\nu)(\nu_0^2 + c_\sigma^2)} + \frac{(1-c_\nu)(\nu_0^2 + c_\sigma^2)}{2(\nu_i^2 + c_\sigma^2)}\right) d.$*

**Theorem 4.** *Consider a data distribution $P$ satisfying Assumption 3. We follow annealed Langevin dynamics for $T = \exp(\mathcal{O}(d))$ steps with noise levels $c_\sigma \geq \sigma_0 \geq \cdots \geq \sigma_T \geq 0$. Suppose the sample is initialized in $P_{\sigma_0}^{(0)}$, then with probability at least $1 - T \cdot \exp(-\mathcal{O}(d))$, we have $\|\mathbf{x}_t - \boldsymbol{\mu}_i\|^2 > \left(\frac{\nu_0^2 + \sigma_t^2}{2} + \frac{\nu_{\max}^2 + \sigma_t^2}{2(1-c_\nu)}\right) d$ for all $t \in \{0\} \cup [T]$ and $i \in [k]$.*

*Proof of Theorem 4.* The feasibility of Assumption 3.v. can be validated by substituting $\nu^2$ in Lemma 9 with $\nu^2 + c_\sigma^2$. To establish Theorem 4, we first note from Proposition 5 that for a sub-Gaussian mixture $P = \sum_{i=0}^k w_i P^{(i)}$, the perturbed distribution of noise level $\sigma$ is $P_\sigma = \sum_{i=0}^k w_i P_\sigma^{(i)}$, where $P^{(0)} = \mathcal{N}(\boldsymbol{\mu}_0, (\nu_i^2 + \sigma^2)\mathbf{I}_d)$ and $P^{(i)}$ is a sub-Gaussian distribution with mean $\boldsymbol{\mu}_i$ and sub-Gaussian parameter $(\nu_i^2 + \sigma^2)$. Similar to the proof of Theorem 1, we decompose

$$\mathbf{x}_t = \mathbf{R}\mathbf{r}_t + \mathbf{N}\mathbf{n}_t, \text{ and } \boldsymbol{\epsilon}_t = \mathbf{R}\boldsymbol{\epsilon}_t^{(\mathbf{r})} + \mathbf{N}\boldsymbol{\epsilon}_t^{(\mathbf{n})},$$

where $\mathbf{R} \in \mathbb{R}^{d \times r}$ an orthonormal basis of the vector space $\{\boldsymbol{\mu}_i\}_{i \in [k]}$ and $\mathbf{N} \in \mathbb{R}^{d \times n}$ an orthonormal basis of the null space of $\{\boldsymbol{\mu}_i\}_{i \in [k]}$. Now, we prove Theorem 4 by applying the techniques developed in Appendix A.1 and A.3 via substituting $\nu^2$ and $\frac{\nu^2}{1-c_\nu}$ with $\frac{\nu^2 + \sigma_t^2}{1-c_\nu}$ at time step $t$. Note that for all $t \in \{0\} \cup [T]$, Assumption 3.iv. implies $\nu_0^2 + \sigma_t^2 > \max\left\{1, \frac{4(c_L^2 + c_\nu c_L)}{c_\nu(1-c_\nu)}\right\} \frac{\nu_{\max}^2 + \sigma_t^2}{1-c_\nu}$ because $c_\sigma \geq \sigma_t$.

First, by Proposition 2, suppose that the sample is initialized in the distribution $P_{\sigma_0}^{(0)}$, then with probability at least $1 - \exp(-\Omega(d))$, we have

$$\|\mathbf{n}_0\|^2 \geq \left(\frac{3(\nu_0^2 + \sigma_0^2)}{4} + \frac{\nu_{\max}^2 + \sigma_0^2}{4(1 - c_\nu)}\right) d. \tag{21}$$

Then, with the assumption that the initialization satisfies $\|\mathbf{n}_0\|^2 \geq \left(\frac{3(\nu_0^2 + \sigma_0^2)}{4} + \frac{\nu_{\max}^2 + \sigma_0^2}{4(1 - c_\nu)}\right) d$, the following proposition similar to Proposition 4 shows that $\|\mathbf{n}_t\|$ remains large with high probability.

**Proposition 7.** *Consider a distribution $P$ satisfying Assumption 3. We follow annealed Langevin dynamics for $T = \exp(\mathcal{O}(d))$ steps with noise level $c_\sigma \geq \sigma_0 \geq \sigma_1 \geq \cdots \geq \sigma_T \geq 0$ for some constant $c_\sigma > 0$. Suppose that the initial sample satisfies $\|\mathbf{n}_0\|^2 \geq \left(\frac{3(\nu_0^2 + \sigma_0^2)}{4} + \frac{\nu_{\max}^2 + \sigma_0^2}{4(1 - c_\nu)}\right) d$, then with probability at least $1 - T \cdot \exp(-\Omega(d))$, we have that $\|\mathbf{n}_t\|^2 > \left(\frac{\nu_0^2 + \sigma_t^2}{2} + \frac{\nu_{\max}^2 + \sigma_t^2}{2(1 - c_\nu)}\right) d$ for all $t \in \{0\} \cup [T]$.*

*Proof of Proposition 7.* We prove Proposition 7 by induction. Suppose the theorem holds for all $T$ values of $1, \cdots, T - 1$. We consider the following 3 cases:

- If there exists some $t \in [T]$ such that $\delta_t > \nu_0^2 + \sigma_t^2$, by Lemma 3 we know that with probability at least $1 - \exp(-\Omega(d))$, we have $\|\mathbf{n}_t\|^2 \geq \left(\frac{3(\nu_0^2 + \sigma_t^2)}{4} + \frac{\nu_{\max}^2 + \sigma_t^2}{4(1 - c_\nu)}\right) d$, thus the problem reduces to the two sub-arrays $\mathbf{n}_0, \cdots, \mathbf{n}_{t-1}$ and $\mathbf{n}_t, \cdots, \mathbf{n}_T$, which can be solved by induction.

- Suppose $\delta_t \leq \nu_0^2 + \sigma_t^2$ for all $t \in [T]$. If there exists some $t \in [T]$ such that $\|\mathbf{n}_{t-1}\|^2 > 36(\nu_0^2 + \sigma_{t-1}^2)d \geq 36(\nu_0^2 + \sigma_t^2)d$, by Lemma 11 we know that with probability at least $1 - \exp(-\Omega(d))$, we have $\|\mathbf{n}_t\|^2 \geq (\nu_0^2 + \sigma_t^2)d > \left(\frac{3(\nu_0^2 + \sigma_t^2)}{4} + \frac{\nu_{\max}^2 + \sigma_t^2}{4(1 - c_\nu)}\right) d$, thus the problem similarly reduces to the two sub-arrays $\mathbf{n}_0, \cdots, \mathbf{n}_{t-1}$ and $\mathbf{n}_t, \cdots, \mathbf{n}_T$, which can be solved by induction.

- Suppose $\delta_t \leq \nu_0^2 + \sigma_t^2$ and $\|\mathbf{n}_{t-1}\|^2 \leq 36(\nu_0^2 + \sigma_{t-1}^2)d$ for all $t \in [T]$. Consider a surrogate sequence $\hat{\mathbf{n}}_t$ such that $\hat{\mathbf{n}}_0 = \mathbf{n}_0$ and for all $t \geq 1$,

$$\hat{\mathbf{n}}_t = \hat{\mathbf{n}}_{t-1} - \frac{\delta_t}{2\nu_0^2 + 2\sigma_t^2}\hat{\mathbf{n}}_{t-1} + \sqrt{\delta_t}\boldsymbol{\epsilon}_t^{(\mathbf{n})}.$$

  Since $\nu_0 > \nu_i$ and $c_\sigma \geq \sigma_t$ for all $t \in \{0\} \cup [T]$, we have $\frac{\nu_i^2 + c_\sigma^2}{\nu_0^2 + c_\sigma^2} > \frac{\nu_i^2 + \sigma_t^2}{\nu_0^2 + \sigma_t^2}$. Notice that for function $f(z) = \log z - \frac{z}{2} + \frac{1}{2z}$, we have $\frac{\mathrm{d}}{\mathrm{d}z}f(z) = \frac{1}{z} - \frac{1}{2} - \frac{1}{2z^2} = -\frac{1}{2}\left(\frac{1}{z} - 1\right)^2 \leq 0$.

  Thus, by Assumption 3.v. we have that for all $t \in [T]$,

$$\|\boldsymbol{\mu}_i - \boldsymbol{\mu}_0\|^2 \leq \frac{(1 - c_\nu)\nu_0^2 - \nu_i^2 - c_\nu c_\sigma^2}{2(1 - c_\nu)}\left(\log\frac{c_\nu(\nu_i^2 + c_\sigma^2)}{(c_L^2 + c_\nu c_L)(\nu_0^2 + c_\sigma^2)}\right.$$
$$\left. - \frac{(\nu_i^2 + c_\sigma^2)}{2(1 - c_\nu)(\nu_0^2 + c_\sigma^2)} + \frac{(1 - c_\nu)(\nu_0^2 + c_\sigma^2)}{2(\nu_i^2 + c_\sigma^2)}\right) d$$
$$\leq \frac{(1 - c_\nu)\nu_0^2 - \nu_i^2 - c_\nu\sigma_t^2}{2(1 - c_\nu)}\left(\log\frac{c_\nu(\nu_i^2 + \sigma_t^2)}{(c_L^2 + c_\nu c_L)(\nu_0^2 + \sigma_t^2)}\right.$$
$$\left. - \frac{(\nu_i^2 + \sigma_t^2)}{2(1 - c_\nu)(\nu_0^2 + \sigma_t^2)} + \frac{(1 - c_\nu)(\nu_0^2 + \sigma_t^2)}{2(\nu_i^2 + \sigma_t^2)}\right) d$$

  Conditioned on $\|\mathbf{n}_{t-1}\|^2 > \left(\frac{\nu_0^2 + \sigma_{t-1}^2}{2} + \frac{\nu_{\max}^2 + \sigma_{t-1}^2}{2(1 - c_\nu)}\right) d$ for all $t \in [T]$, by Lemma 12 we have that for $T = \exp(\mathcal{O}(d))$,

$$\|\hat{\mathbf{n}}_T - \mathbf{n}_T\| < \left(\sqrt{\frac{5(\nu_0^2 + \sigma_T^2)}{8} + \frac{3(\nu_{\max}^2 + \sigma_T^2)}{8(1 - c_\nu)}} - \sqrt{\frac{\nu_0^2 + \sigma_T^2}{2} + \frac{\nu_{\max}^2 + \sigma_T^2}{2(1 - c_\nu)}}\right)\sqrt{d}.$$

By Lemma 8 we have that with probability at least $1 - \exp(-\Omega(d))$,

$$\|\hat{\mathbf{n}}_T\|^2 \geq \left( \frac{5(\nu_0^2 + \sigma_T^2)}{8} + \frac{3(\nu_{\max}^2 + \sigma_T^2)}{8(1 - c_\nu)} \right) d.$$

Combining the two inequalities implies the desired bound

$$\|\mathbf{n}_T\| \geq \|\hat{\mathbf{n}}_T\| - \|\hat{\mathbf{n}}_T - \mathbf{n}_T\| > \sqrt{\left( \frac{\nu_0^2 + \sigma_T^2}{2} + \frac{\nu_{\max}^2 + \sigma_T^2}{2(1 - c_\nu)} \right) d}.$$

Hence by induction we obtain $\|\mathbf{n}_t\|^2 > \left( \frac{\nu_0^2 + \sigma_T^2}{2} + \frac{\nu_{\max}^2 + \sigma_T^2}{2(1 - c_\nu)} \right) d$ for all $t \in [T]$ with probability at least

$$(1 - (T - 1) \exp(-\Omega(d))) \cdot (1 - \exp(-\Omega(d))) \geq 1 - T \exp(-\Omega(d)).$$

Therefore we complete the proof of Proposition 7. □

Finally, combining equation 21 and Proposition 7 finishes the proof of Theorem 4. □

## C CONVERGENCE ANALYSIS OF CHAINED LANGEVIN DYNAMICS

*Proof of Proposition 1.* For simplicity, denote $\mathbf{x}^{[q]} = \{\mathbf{x}^{(1)}, \cdots, \mathbf{x}^{(q)}\}$. By the definition of total variation distance, for all $q \in [d/Q]$ we have

$$\mathrm{TV}\left( \hat{P}\left( \mathbf{x}^{[q]} \right), P\left( \mathbf{x}^{[q]} \right) \right)$$

$$= \frac{1}{2} \int \left| \hat{P}\left( \mathbf{x}^{[q]} \right) - P\left( \mathbf{x}^{[q]} \right) \right| \mathrm{d}\mathbf{x}^{[q]}$$

$$= \frac{1}{2} \int \left| \hat{P}\left( \mathbf{x}^{(q)} \mid \mathbf{x}^{[q-1]} \right) \hat{P}\left( \mathbf{x}^{[q-1]} \right) - P\left( \mathbf{x}^{(q)} \mid \mathbf{x}^{[q-1]} \right) P\left( \mathbf{x}^{[q-1]} \right) \right| \mathrm{d}\mathbf{x}^{[q]}$$

$$\leq \frac{1}{2} \int \left| \hat{P}\left( \mathbf{x}^{(q)} \mid \mathbf{x}^{[q-1]} \right) \hat{P}\left( \mathbf{x}^{[q-1]} \right) - \hat{P}\left( \mathbf{x}^{(q)} \mid \mathbf{x}^{[q-1]} \right) P\left( \mathbf{x}^{[q-1]} \right) \right| \mathrm{d}\mathbf{x}^{[q]}$$

$$\quad + \frac{1}{2} \int \left| \hat{P}\left( \mathbf{x}^{(q)} \mid \mathbf{x}^{[q-1]} \right) P\left( \mathbf{x}^{[q-1]} \right) - P\left( \mathbf{x}^{(q)} \mid \mathbf{x}^{[q-1]} \right) P\left( \mathbf{x}^{[q-1]} \right) \right| \mathrm{d}\mathbf{x}^{[q]}$$

$$= \frac{1}{2} \int \hat{P}\left( \mathbf{x}^{(q)} \mid \mathbf{x}^{[q-1]} \right) \mathrm{d}\mathbf{x}^{(q)} \int \left| \hat{P}\left( \mathbf{x}^{[q-1]} \right) - P\left( \mathbf{x}^{[q-1]} \right) \right| \mathrm{d}\mathbf{x}^{[q-1]}$$

$$\quad + \frac{1}{2} \int \left| \hat{P}\left( \mathbf{x}^{(q)} \mid \mathbf{x}^{[q-1]} \right) - P\left( \mathbf{x}^{(q)} \mid \mathbf{x}^{[q-1]} \right) \right| \mathrm{d}\mathbf{x}^{(q)} \int P\left( \mathbf{x}^{[q-1]} \right) \mathrm{d}\mathbf{x}^{[q-1]}$$

$$= \mathrm{TV}\left( \hat{P}\left( \mathbf{x}^{[q-1]} \right), P\left( \mathbf{x}^{[q-1]} \right) \right) + \mathrm{TV}\left( \hat{P}\left( \mathbf{x}^{(q)} \mid \mathbf{x}^{[q-1]} \right), P\left( \mathbf{x}^{(q)} \mid \mathbf{x}^{[q-1]} \right) \right)$$

$$\leq \mathrm{TV}\left( \hat{P}\left( \mathbf{x}^{[q-1]} \right), P\left( \mathbf{x}^{[q-1]} \right) \right) + \varepsilon \cdot \frac{Q}{d}.$$

Upon summing up the above inequality for all $q \in [d/Q]$, we obtain

$$\mathrm{TV}\left( \hat{P}(\mathbf{x}), P(\mathbf{x}) \right) = \sum_{q=1}^{d/Q} \left( \mathrm{TV}\left( \hat{P}\left( \mathbf{x}^{[q]} \right), P\left( \mathbf{x}^{[q]} \right) \right) - \mathrm{TV}\left( \hat{P}\left( \mathbf{x}^{[q-1]} \right), P\left( \mathbf{x}^{[q-1]} \right) \right) \right)$$

$$\leq \sum_{q=1}^{d/Q} \varepsilon \cdot \frac{Q}{d} = \varepsilon$$

Thus we finish the proof of Proposition 1. □

# D    EXPERIMENTAL DETAILS AND ADDITIONAL EXPERIMENTS

**Algorithm Setup:** Our choices of algorithm hyperparameters are based on (Song & Ermon, 2019) and (Song & Ermon, 2020). For $\sigma_{\max} = 1$, following from Song & Ermon (2019), we consider $L = 10$ different standard deviations such that $\{\lambda_i\}_{i \in [L]}$ is a geometric sequence with $\lambda_1 = 1$ and $\lambda_{10} = 0.01$. For annealed Langevin dynamics with $T$ iterations, we choose the noise levels $\{\sigma_t\}_{t \in [T]}$ by repeating every element of $\{\lambda_i\}_{i \in [L]}$ for $T/L$ times and we set the step size as $\delta_t = 2 \times 10^{-5} \cdot \sigma_t^2/\sigma_T^2$ for every $t \in [T]$. For vanilla Langevin dynamics with $T$ iterations, we use the same step size as annealed Langevin dynamics. For Chained-VLD and Chained-ALD, the patch size $Q$ is chosen depending on different tasks. For every patch of Chained-ALD, we choose the noise levels $\{\sigma_t\}_{t \in [TQ/d]}$ by repeating every element of $\{\lambda_i\}_{i \in [L]}$ for $TQ/dL$ times and we set the step size as $\delta_t = 2 \times 10^{-5} \cdot \sigma_t^2/\sigma_{TQ/d}^2$ for every $t \in [TQ/d]$. The step size of Chained-VLD is the same as Chained-ALD.

We would like to highlight that the inference time of Chained-LD is significantly lower than vanilla LD in practice. Our theoretical comparison between Chained-LD and vanilla LD is based on iteration complexity, i.e., the number of queries to the score function $\nabla \log P(x^{(q)}|x^{(1)}, \cdots, x^{(q-1)})$ or $\nabla \log P(x)$. Since Chained-LD only updates one patch at every iteration while vanilla LD updates the whole image, Chained-LD will be significantly faster than vanilla LD.

## D.1    SYNTHETIC GAUSSIAN MIXTURE MODEL

We choose the data distribution $P$ as a mixture of three Gaussian components in dimension $d = 100$:

$$P = 0.2P^{(0)} + 0.4P^{(1)} + 0.4P^{(2)} = 0.2\mathcal{N}(\mathbf{0}_d, 3\boldsymbol{I}_d) + 0.4\mathcal{N}(\mathbf{1}_d, \boldsymbol{I}_d) + 0.4\mathcal{N}(-\mathbf{1}_d, \boldsymbol{I}_d).$$

Since the distribution is given, we assume that the sampling algorithms have access to the ground-truth score function. We set the batch size as 1000 and patch size $Q = 10$ for chained Langevin dynamics. We use $T \in \{10^3, 10^4, 10^5, 10^6\}$ iterations for vanilla and chained Langevin dynamics. A sample $\mathbf{x}$ is clustered in mode 1 if it satisfies $\|\mathbf{x} - \boldsymbol{\mu}_1\|^2 \leq 5d$ and $\|\mathbf{x} - \boldsymbol{\mu}_1\|^2 \leq \|\mathbf{x} - \boldsymbol{\mu}_2\|^2$; in mode 2 if $\|\mathbf{x} - \boldsymbol{\mu}_2\|^2 \leq 5d$ and $\|\mathbf{x} - \boldsymbol{\mu}_1\|^2 > \|\mathbf{x} - \boldsymbol{\mu}_2\|^2$; and in mode 0 otherwise. The initial samples are i.i.d. chosen from $P^{(0)}$, $P^{(1)}$, or $P^{(2)}$, and the results are presented in Figures 2, 5, and 6 respectively. The two subfigures above the dashed line illustrate the samples from the initial distribution and target distribution, and the subfigures below the dashed line are the samples generated by different algorithms. Furthermore, in Figures 7, 8 and 9 we demonstrate the effect of different values of $Q \in \{1, 4, 10, 20, 50\}$ on the mode-seeking tendencies of Chained-LD. We can observe that for dimension $d = 100$, a moderate patch size $Q \in \{1, 4, 10\}$ has similar performance, a large patch size $Q = 20$ needs more steps to find the other two modes, while an overly-large patch size $Q = 50$ almost cannot find other modes.

## D.2    SCORE FUNCTION ESTIMATOR

In realistic scenarios, since we do not have direct access to the (perturbed) score function, Song & Ermon (2019) proposed the Noise Conditional Score Network (NCSN) $\mathbf{s}_{\boldsymbol{\theta}}(\mathbf{x}, \sigma)$ to jointly estimate the scores of all perturbed data distributions, i.e.,

$$\forall \sigma \in \{\sigma_t\}_{t \in [T]}, \ \mathbf{s}_{\boldsymbol{\theta}}(\mathbf{x}, \sigma) \approx \nabla_{\mathbf{x}} \log P_\sigma(\mathbf{x}).$$

To train the NCSN, Song & Ermon (2019) adopted denoising score matching, which minimizes the following loss

$$\mathcal{L}\left(\boldsymbol{\theta}; \{\sigma_t\}_{t \in [T]}\right) := \frac{1}{2T} \sum_{t \in [T]} \sigma_t^2 \mathbb{E}_{\mathbf{x} \sim P} \mathbb{E}_{\tilde{\mathbf{x}} \sim \mathcal{N}(\mathbf{x}, \sigma_t^2 \boldsymbol{I}_d)} \left[\left\|\mathbf{s}_{\boldsymbol{\theta}}(\tilde{\mathbf{x}}, \sigma_t) - \frac{\tilde{\mathbf{x}} - \mathbf{x}}{\sigma_t^2}\right\|^2\right].$$

Assuming the NCSN has enough capacity and sufficient training samples, $\mathbf{s}_{\boldsymbol{\theta}^*}(\mathbf{x}, \sigma)$ minimizes the loss $\mathcal{L}\left(\boldsymbol{\theta}; \{\sigma_t\}_{t \in [T]}\right)$ if and only if $\mathbf{s}_{\boldsymbol{\theta}^*}(\mathbf{x}, \sigma_t) = \nabla_{\mathbf{x}} \log P_{\sigma_t}(\mathbf{x})$ almost surely for all $t \in [T]$.

In Chained Langevin dynamics, an ideal conditional score function estimator $\mathbf{s}_{\boldsymbol{\theta}}$ could jointly estimate the scores of all perturbed conditional patch distribution, i.e., $\forall \sigma \in \{\sigma_t\}_{t \in [TQ/d]}, q \in [d/Q]$,

$$\mathbf{s}_{\boldsymbol{\theta}}\left(\mathbf{x}^{(q)} \mid \sigma, \mathbf{x}^{(1)}, \cdots, \mathbf{x}^{(q-1)}\right) \approx \nabla_{\mathbf{x}^{(q)}} \log P_{\sigma}(\mathbf{x}^{(q)} \mid \mathbf{x}^{(1)}, \cdots \mathbf{x}^{(q-1)}).$$

Following from Song & Ermon (2019), we use the denoising score matching to train the estimator. For a given $\sigma$, the denoising score matching objective is

$$\ell(\boldsymbol{\theta}; \sigma) := \frac{1}{2} \mathbb{E}_{\mathbf{x} \sim P} \mathbb{E}_{\tilde{\mathbf{x}} \sim \mathcal{N}(\mathbf{x}, \sigma^2 \boldsymbol{I}_d)} \sum_{q \in [d/Q]} \left[ \left\| \mathbf{s}_{\boldsymbol{\theta}}\left(\mathbf{x}^{(q)} \mid \sigma, \mathbf{x}^{(1)}, \cdots, \mathbf{x}^{(q-1)}\right) - \frac{\tilde{\mathbf{x}}^{(q)} - \mathbf{x}^{(q)}}{\sigma^2} \right\|^2 \right].$$

Then, combining the objectives gives the following loss

$$\mathcal{L}\left(\boldsymbol{\theta}; \{\sigma_t\}_{t \in [TQ/d]}\right) := \frac{d}{TQ} \sum_{t \in [TQ/d]} \sigma_t^2 \ell(\boldsymbol{\theta}; \sigma_t).$$

As shown in Vincent (2011), an estimator $\mathbf{s}_{\boldsymbol{\theta}}$ with enough capacity and sufficient training samples minimizes the loss $\mathcal{L}$ if and only if $\mathbf{s}_{\boldsymbol{\theta}}$ outputs the scores of all perturbed conditional patch distribution almost surely.

### D.3 IMAGE DATASETS

Our implementation and hyperparameter selection are based on (Song & Ermon, 2019) and (Song & Ermon, 2020). During training, we i.i.d. randomly flip an image with probability 0.5 to construct the two modes (i.e., original and flipped images). All models are optimized by Adam with learning rate 0.001 and batch size 128 for a total of 200000 training steps, and we use the model at the last iteration to generate the samples. We perform experiments on MNIST (LeCun, 1998) (CC BY-SA 3.0 License) and Fashion-MNIST (Xiao et al., 2017) (MIT License) datasets and we set the patch size as $Q = 14$.

For the score networks of chained annealed Langevin dynamics (Chained-ALD), we use the official PyTorch implementation of an LSTM network (Sak et al., 2014) followed by a linear layer. For MNIST and Fashion-MNIST datasets, we set the input size of the LSTM as $Q = 14$, the number of features in the hidden state as 1024, and the number of recurrent layers as 2. The inputs of LSTM include inputting tensor, hidden state, and cell state, and the outputs of LSTM include the next hidden state and cell state, which can be fed to the next input. To estimate the noisy score function, we first input the noise level $\sigma$ (repeated for $Q$ times to match the input size of LSTM) and all-0 hidden and cell states to obtain an initialization of the hidden and cell states. Then, we divide a sample into $d/Q$ patches and input the sequence of patches to the LSTM. For every output hidden state corresponding to one patch, we apply a linear layer of size $1024 \times Q$ to estimate the noisy score function of the patch.

To generate samples, we use $T \in \{10000, 30000, 100000\}$ iterations for annealed Langevin dynamics (ALD) and Chained-ALD. The initial samples are chosen as either original or flipped images from the dataset, and the results for MNIST and Fashion-MNIST datasets are presented in Figures 11, 3, 4, and 12 respectively. The two subfigures above the dashed line illustrate the samples from the initial distribution and target distribution, and the subfigures below the dashed line are the samples generated by different algorithms.

All experiments were run with one RTX3090 GPU. It is worth noting that the training and inference time of chained Langevin dynamics using LSTM is considerably faster than vanilla/annealed Langevin dynamics using RefineNet. For a course of 200000 training steps on MNIST/Fashion-MNIST, due to the different network architectures, LSTM takes around 2.3 hours while RefineNet takes around 9.2 hours. Concerning image generation, chained Langevin dynamics is significantly faster than vanilla/annealed Langevin dynamics since every iteration of chained Langevin dynamics only updates a patch of constant size, while every iteration of vanilla/annealed Langevin dynamics requires computing all coordinates of the sample. One iteration of chained Langevin dynamics using LSTM takes around 1.97 ms, while one iteration of vanilla/annealed Langevin dynamics using RefineNet takes around 43.7 ms.

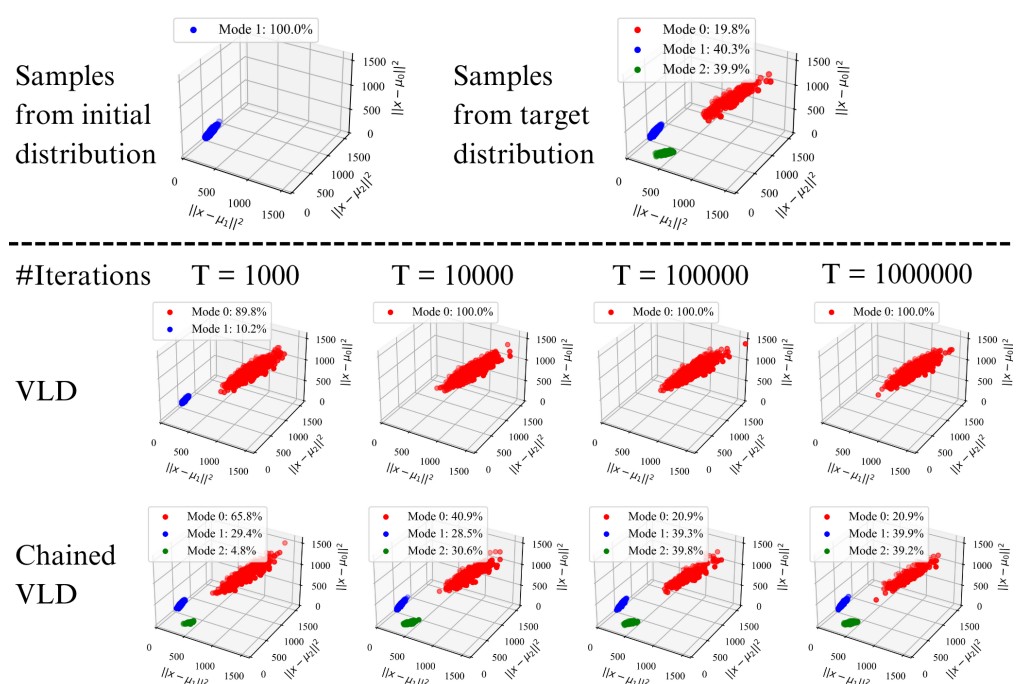

Figure 5: Samples from a mixture of three Gaussian modes generated by vanilla Langevin dynamics (VLD) and chained vanilla Langevin dynamics (Chained-VLD). Three axes are $\ell_2$ distance from samples to the mean of the three modes. The samples are initialized in mode 1.

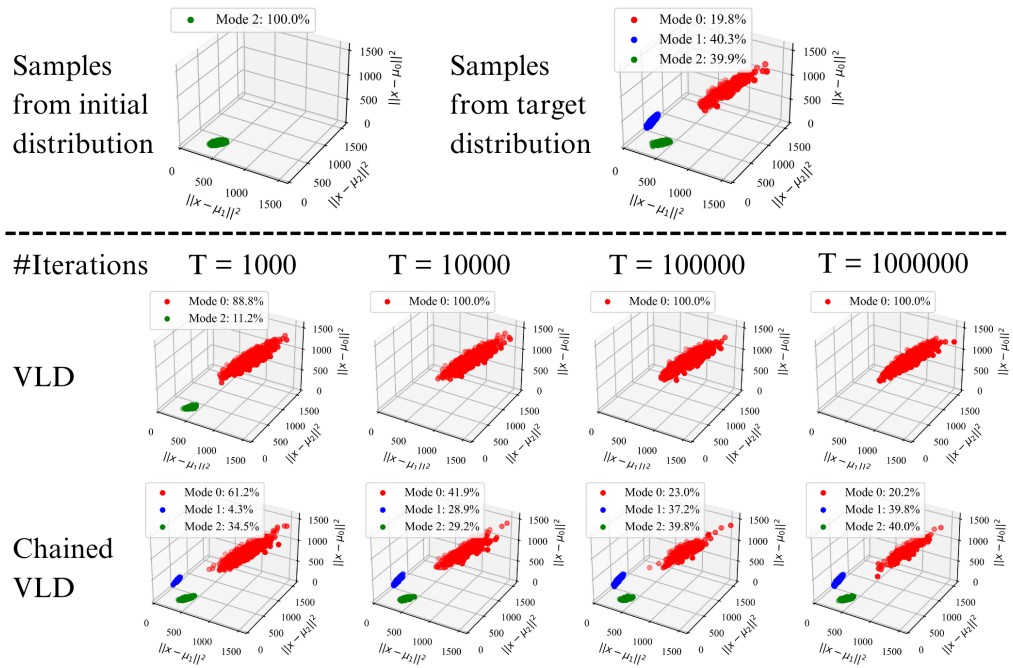

Figure 6: Samples from a mixture of three Gaussian modes generated by vanilla Langevin dynamics (VLD) and chained vanilla Langevin dynamics (Chained-VLD). Three axes are $\ell_2$ distance from samples to the mean of the three modes. The samples are initialized in mode 2.

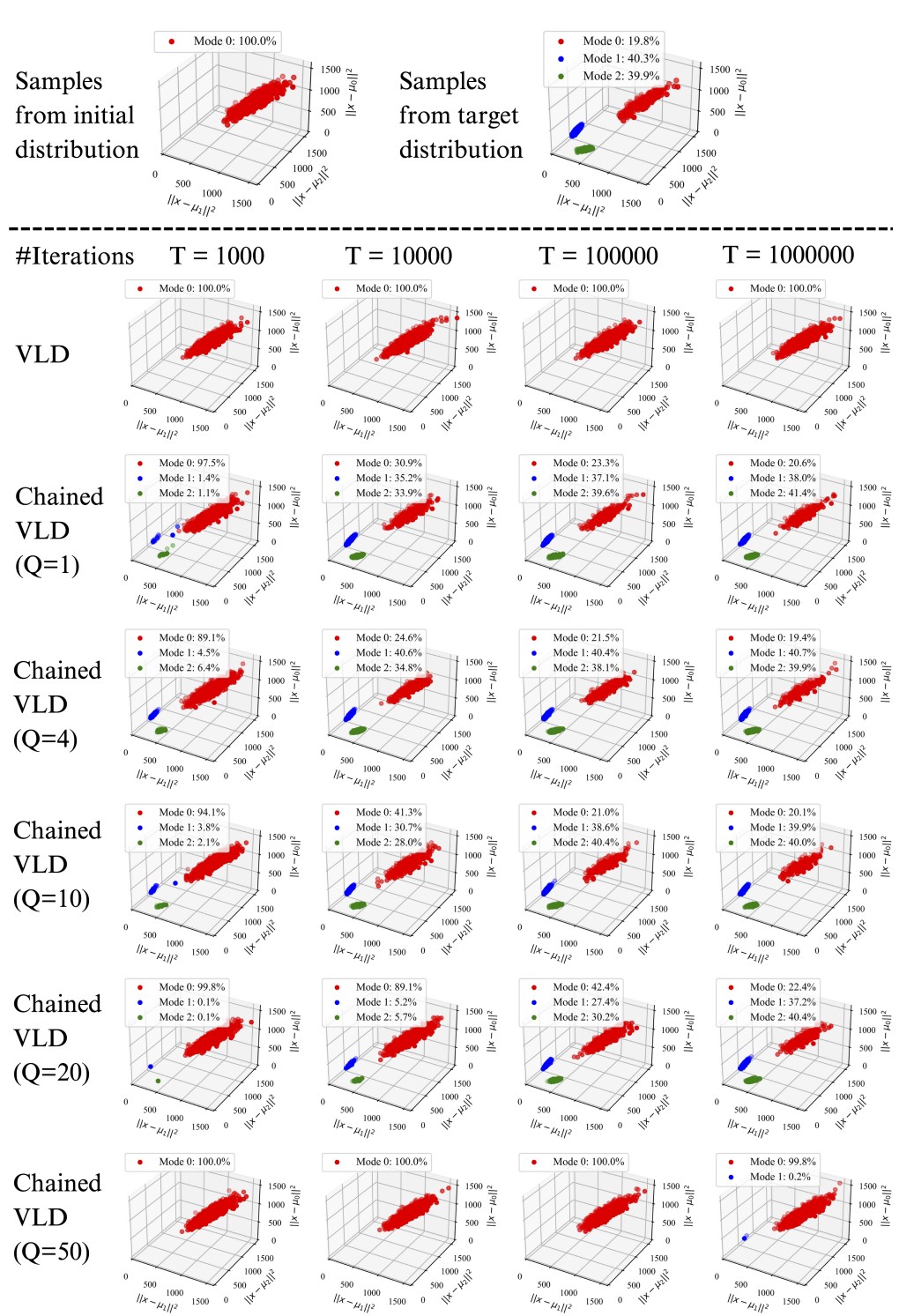

Figure 7: Samples from a mixture of three Gaussian modes generated by vanilla Langevin dynamics (VLD) and chained vanilla Langevin dynamics (Chained-VLD) with patch size $Q \in \{1, 4, 10, 20, 50\}$. Three axes are $\ell_2$ distance from samples to the mean of the three modes. The samples are initialized in mode 0.

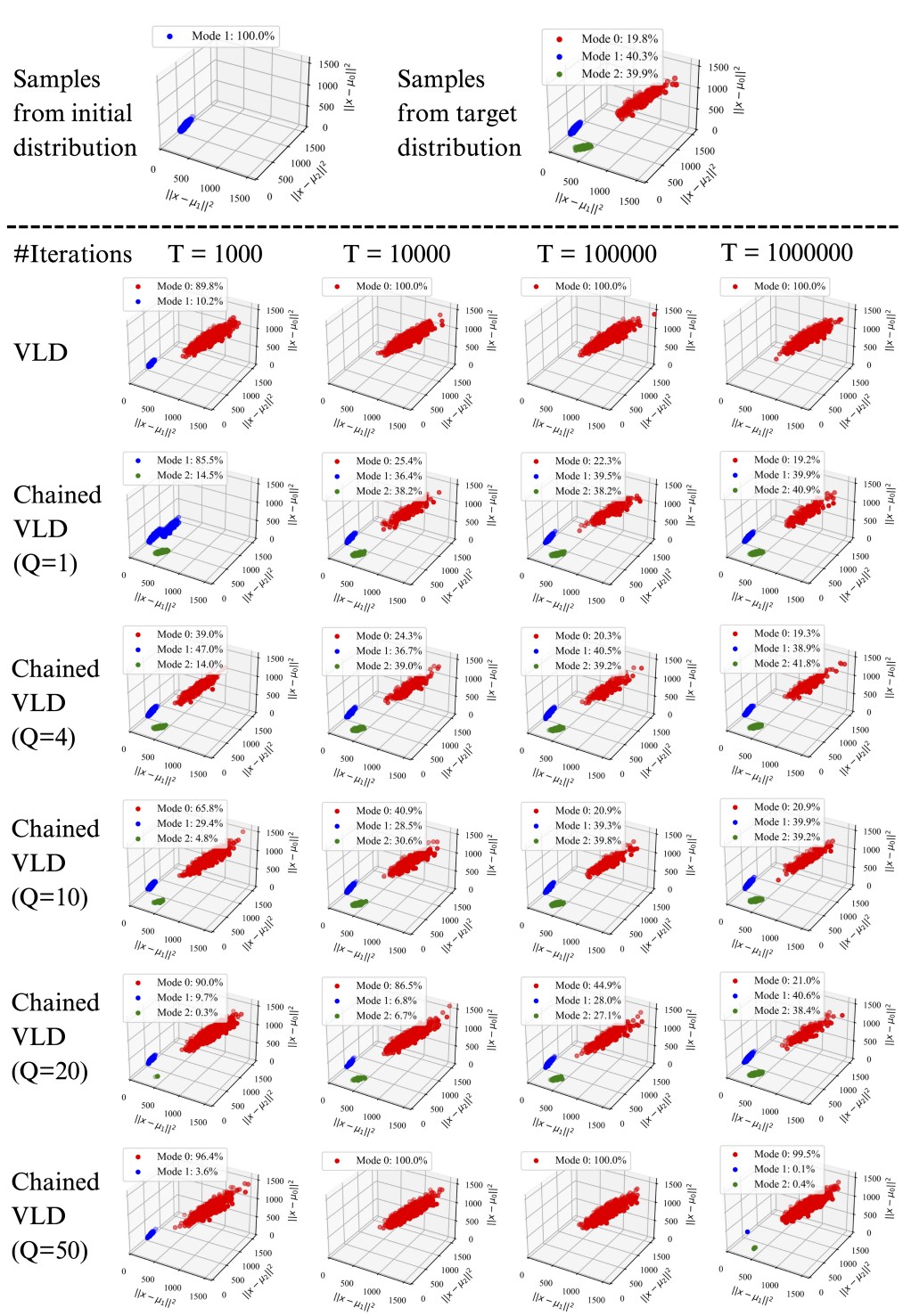

Figure 8: Samples from a mixture of three Gaussian modes generated by vanilla Langevin dynamics (VLD) and chained vanilla Langevin dynamics (Chained-VLD) with patch size $Q \in \{1, 4, 10, 20, 50\}$. Three axes are $\ell_2$ distance from samples to the mean of the three modes. The samples are initialized in mode 1.

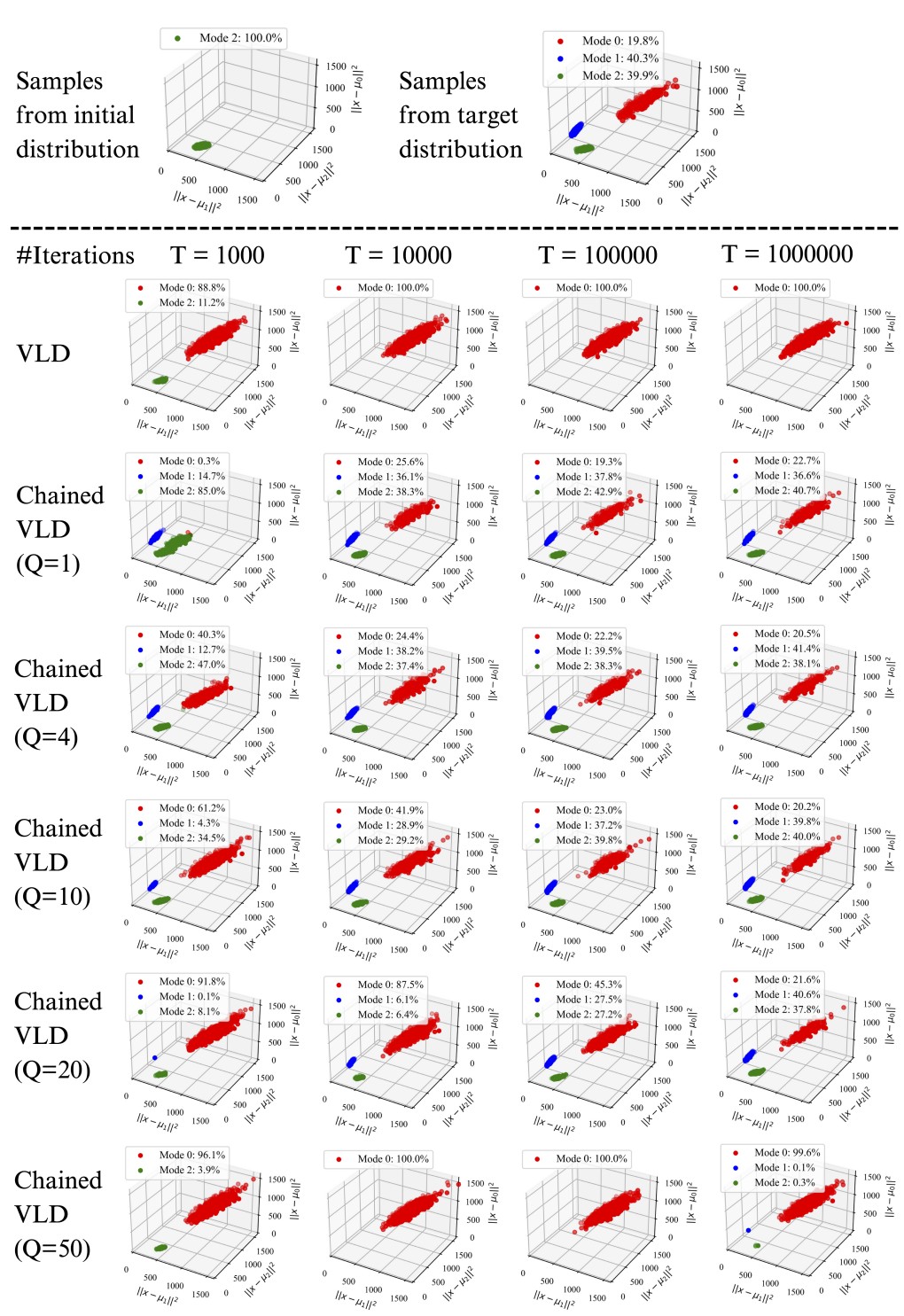

Figure 9: Samples from a mixture of three Gaussian modes generated by vanilla Langevin dynamics (VLD) and chained vanilla Langevin dynamics (Chained-VLD) with patch size $Q \in \{1, 4, 10, 20, 50\}$. Three axes are $\ell_2$ distance from samples to the mean of the three modes. The samples are initialized in mode 2.

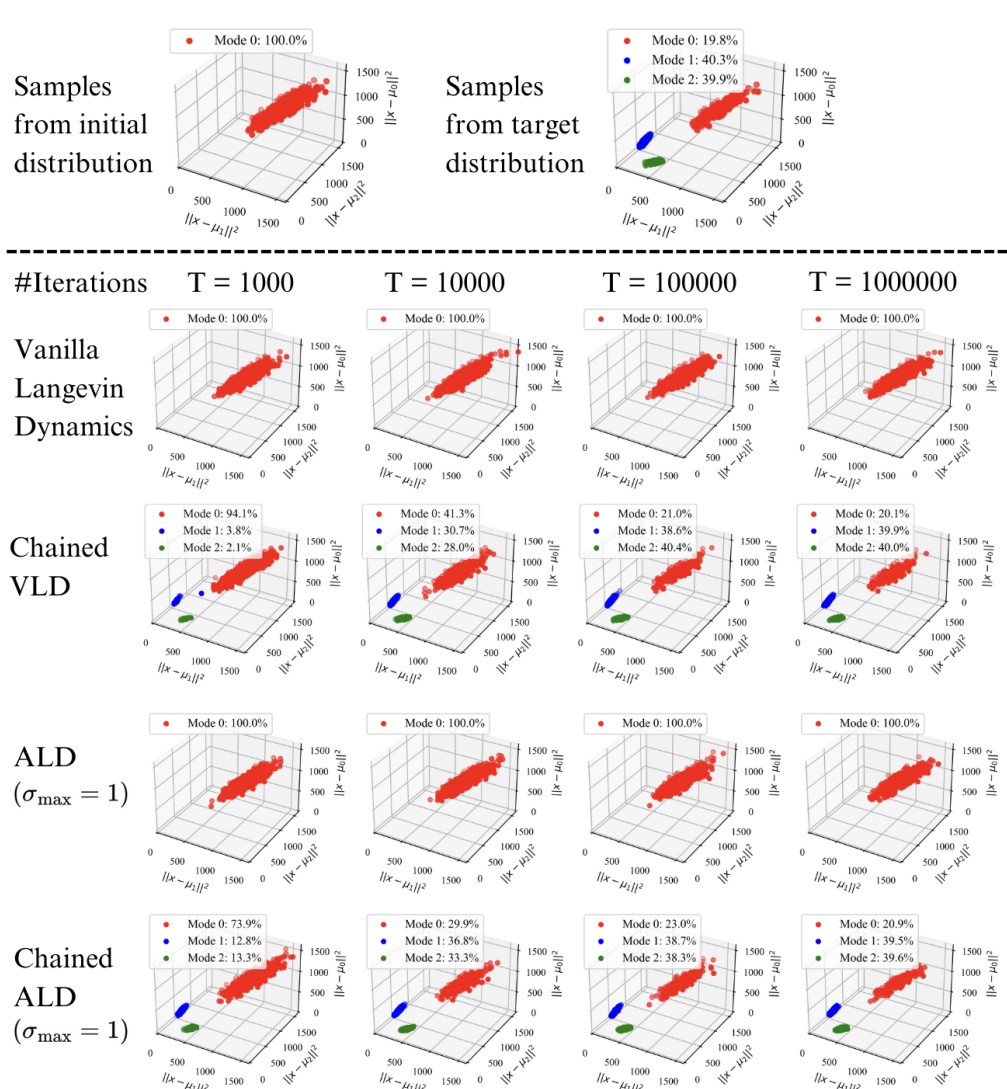

Figure 10: Samples from a mixture of three Gaussian modes generated by Langevin dynamics and chained Langevin dynamics with patch size $Q = 10$. Three axes are $\ell_2$ distance from samples to the mean of the three modes. The samples are initialized in mode 0.

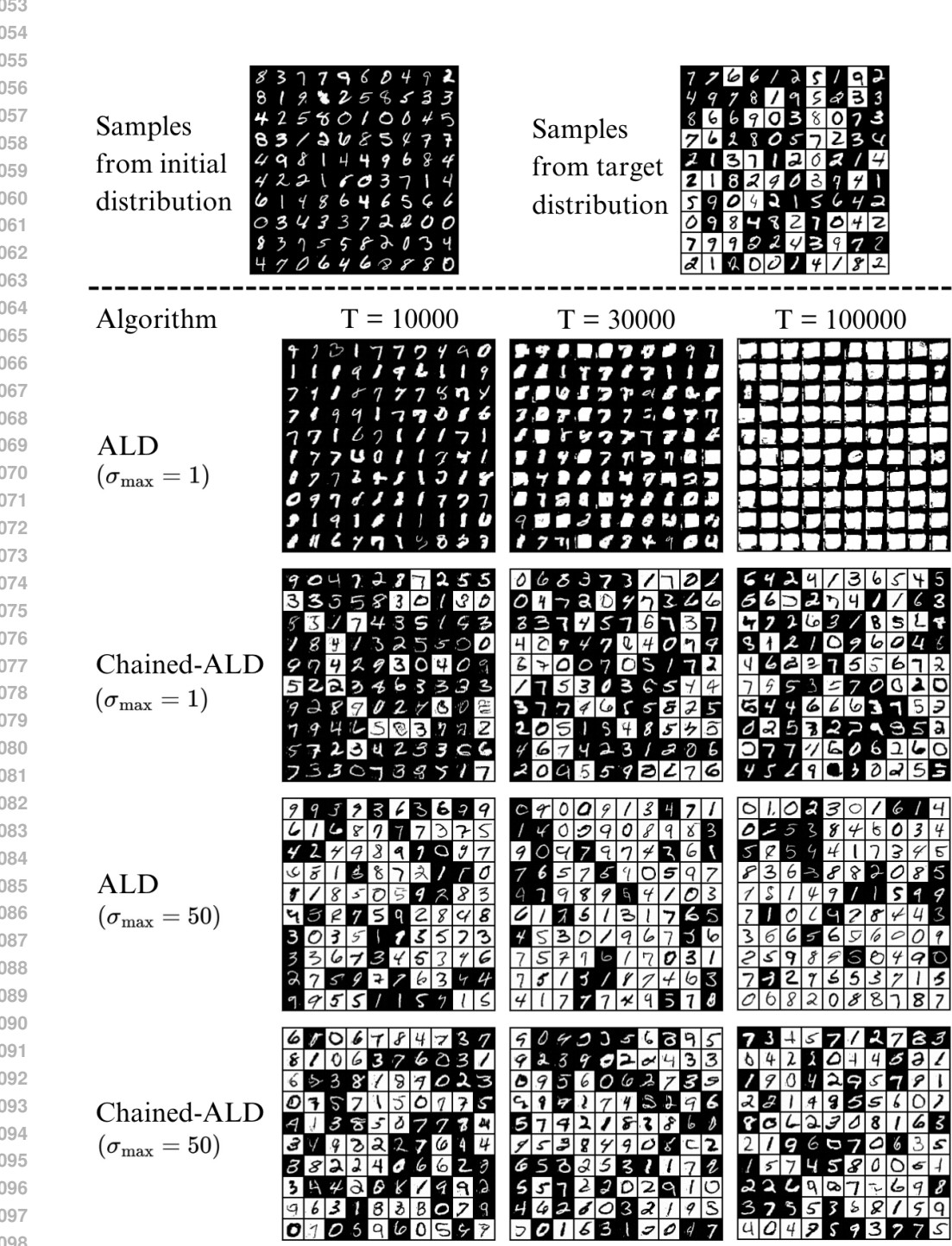

Figure 11: Samples from a mixture distribution of the original and flipped images from the MNIST dataset generated by annealed Langevin dynamics (ALD) and chained annealed Langevin dynamics (Chained-ALD) for different numbers of iterations. The maximum noise level $\sigma_{\max}$ is set to be 1 or 50. The samples are initialized as original images from MNIST.

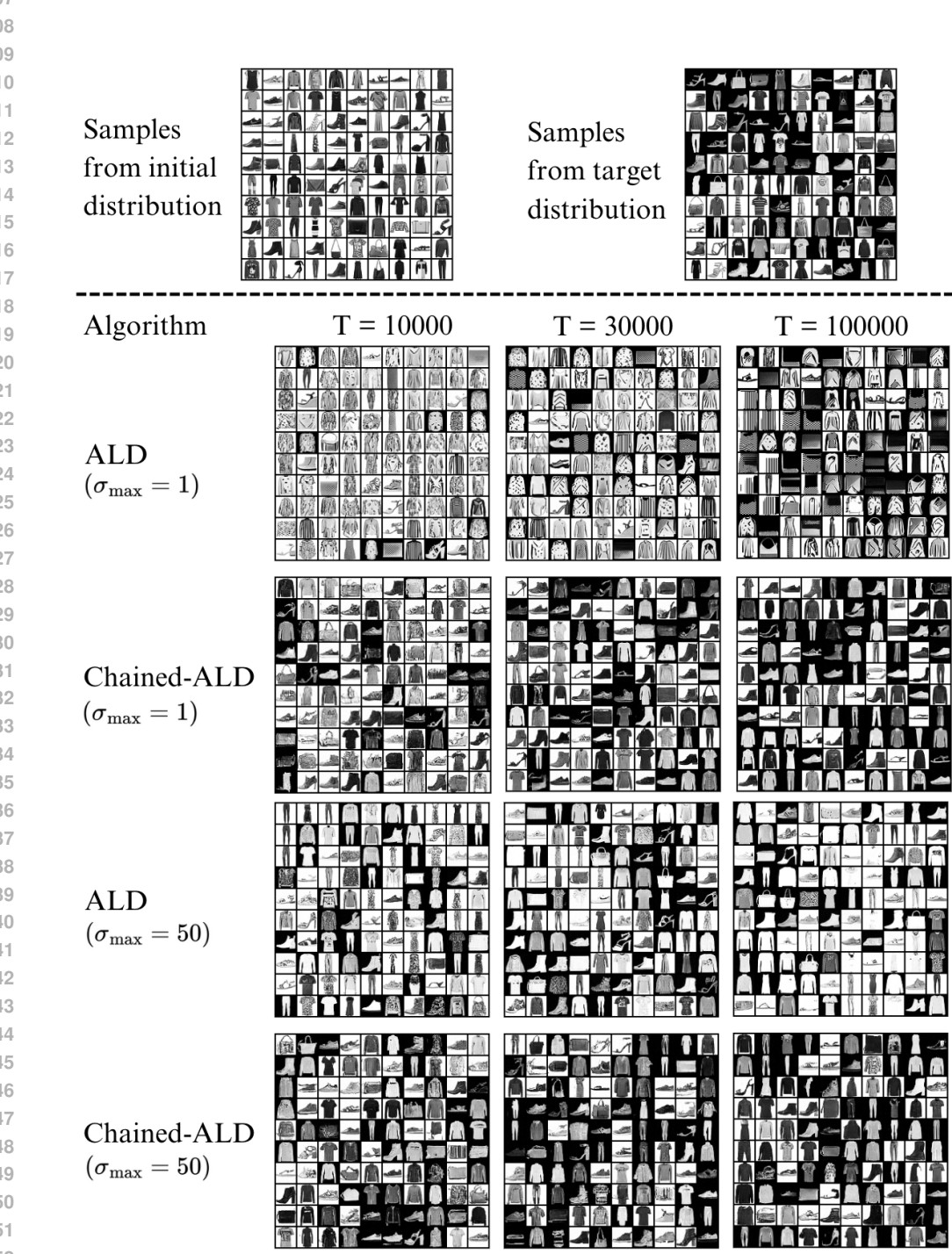

Figure 12: Samples from a mixture distribution of the original and flipped images from the Fashion-MNIST dataset generated by annealed Langevin dynamics (ALD) and chained annealed Langevin dynamics (Chained-ALD) for different numbers of iterations. The maximum noise level $\sigma_{\max}$ is set to be 1 or 50. The samples are initialized as flipped images from FashionMNIST.

