# OpenReview forum: "On the Mode-Seeking Properties of Langevin Dynamics"
_ICLR.cc/2025/Conference — Submitted to ICLR 2025_

### Official Review · Reviewer_2K9H · 2024-11-01

**Soundness:** 2
**Presentation:** 1
**Contribution:** 1
**Rating:** 3
**Confidence:** 4

**Summary:**

The paper considers the task of sampling from a multimodal density using Langevin-like SDEs. The authors considers the standard Euler discretisation of the over damped Langevin diffusion and show that the chain does not discover the modes with high probability after a number of iterations that is exponential in the dimension when the target is a mixture of (sub)-gaussian distributions. To improve over this slow mixing, the paper proposes chained Langevin dynamics, essentially a Gibbs-type approach where Langevin dynamics are applied to batches of the chain instead of the full vector.

**Strengths:**

The paper considers an interesting problem, since slow mixing in multimodal settings is an issue. The authors also propose an algorithm to obviate such slow mixing.

**Weaknesses:**

- The paper is hard to read and contains several paragraphs that are of dubious relevance (e.g. lines 141-144 mentioning GANs, several paragraphs on score-based diffusion models, lines 150-161, 179-200, where it is not clear these are relevant for this article). The introduction is long and overall unclear.

- The authors do not seem comfortable with the MCMC literature, e.g. referring to the Euler discretisation of the overdamped Langevin diffusion as "stochastic gradient Langevin Dynamics", citing a paper that introduces a completely different approach based on subsampling.

- It is not clear if the authors consider the usual sampling context vs. the generative modelling context: is the score of the data distribution known? Moreover the paper contains plenty of references to the generative modelling literature, but it is not clear what is the purpose, and I can only imagine it is to increase their chances of publication.

- It is unclear if the chained algorithm that is proposed has even the right stationary distribution, nor if it has any useful property for sampling purposes. Moreover, one needs to have the conditional densities to run the algorithm in the first place.

-  The experiments, especially those on the MNIST data, miss explanations on the obtained results, so it is unclear how to interpret them.

- Overall, I find the paper should go through a substantial rewriting and thus should be rejected.

**Questions:**

- Is the idea behind the chained algorithm (i.e. decomposing the process in blocks) based/influenced on previous works? If so, the authors should mention the relevant papers.

-Could the authors give any justification for their algorithm?

- Lines 170-171: why is there a minus sign in front of the drift term?

---

> ### Author Response · Authors · 2024-11-25
> **Response to Reviewer 2K9H (Part 1)**
>
> We thank Reviewer 2K9H for his/her time and thoughtful feedback on our work. Below is our response to the questions and comments in the review.
>
>
> **1- Connections with score-based generative modeling**
>
> As we have explained in the global response, our work focuses on sampling from a known distribution using variants of Langevin dynamics. To address the reviewer’s comment, we have clarified in the introduction that our paper studies the sampling task exclusively and does not focus on generative modeling. We have relocated all the citations for related references on generative modeling to the related work section, and the revised introduction solely focuses on the sampling task addressed by Langevin dynamics and Chained-LD algorithms. We have also included Remarks 1 and 2 to clarify the sampling nature of our proposed Chained-LD algorithm.
>
>
> **2- Connections with score-based sampling and GANs**
>
> Langevin dynamics and their properties have been extensively studied in the literature. In our work, we specifically focus on the *mode-seeking tendencies* of Langevin dynamics. Therefore, in the related work section, we discussed previous works on mode-seeking tendencies of sampling and generative modeling algorithms, including mode-seekingness of GANs. We think our studied problem is related to these references, because they all study the mode-seeking tendencies of sample generation algorithms.
>
>
> **3- The term “stochastic gradient Langevin dynamics”**
>
> We thank the reviewer for pointing this out. We have clarified the terminology in the introduction, and replaced it with "Langevin dynamics" in the revision.
>
>
> **4- Assumptions on the score function**
>
> As discussed in the global response, we would like to clarify that the main focus of our work is the mode-seeking properties of Langevin dynamics as a **sampling algorithm** from the score function of a given target distribution $P$. In our theoretical results for Langevin dynamics (Theorems 1 and 2), we assume that the sampling algorithm has access to the given score function and follows the Langevin dynamics update to generate a sample from the underlying distribution. Regarding Chained-LD, Algorithm 1 describes the steps of Chained-LD for a given score function (we do not learn or estimate it). Similarly, Proposition 1 analyzes its convergence by assuming access to the score function of the target distribution. We have added Remarks 1 and 2 to clarify this point in the revision.
>
>
>
> **5- Stationary distribution of Chained-LD sampler**
>
> Chained-LD’s stationary distribution is the same as the target distribution $P$. This statement follows from the chain rule in probability. Note that if every patch is sampled from the conditional distribution $P(x^{(q)}|x^{(1)}, \cdots, x^{(q-1)})$, the whole sequence follows from the target distribution $P$ due to the chain rule:
>
> $$P(x) = \prod_{q=1}^{d/Q} P(x^{(q)}|x^{(1)}, \cdots, x^{(q-1)})$$
>
> Also, the conditional densities used in Chained-LD do not require any extra information compared to the target distribution $P(x)$ assumed in Langevin dynamics. In both cases, the algorithms have complete knowledge of the distribution, where Langevin dynamics has access to the joint distribution $P$ and Chained-LD has access to the conditional distributions that, based on chain rule, have the same information as joint distribution $P$. We have added Remark 2 to make this point clear in the revision.
>
>
> **6- Interpreting the numerical results**
>
> In the synthetic Gaussian mixture experiments, we consider a mixture of three Gaussian modes. As shown in Figure 2, vanilla Langevin dynamics (VLD) cannot find the other two modes within $10^6$ iterations if the sample is initialized in mode 0, while chained vanilla Langevin dynamics (Chained-VLD) can find the other two modes in 1000 steps and correctly recover their frequencies as gradually increasing the number of iterations, which demonstrates the mode-seeking properties of VLD and verifies our theoretical results in Theorem 1 and Proposition 1.
>
> In the MNIST and Fashion-MNIST experiments, we consider a mixture of two modes – the original samples from the datasets and the flipped samples. We followed the training algorithm proposed by [2] to estimate the score function from samples. Considering the resulting score function as a given score function, we ran the sampling algorithm of annealed Langevin dynamics (ALD) and Chained-ALD. As shown in Figures 3 and 4, if we initialize the sampling algorithms in one mode, ALD with bounded noise levels could not find the other missing mode, while ALD with a larger initial noise level and Chained-ALD is able to find both modes, which demonstrates the effect of noise levels on the mode-seeking properties of Langevin dynamics and verifies our theoretical results in Theorem 3.

---

> ### Author Response · Authors · 2024-11-25
> **Response to Reviewer 2K9H (Part 2)**
>
> **7- Justification for the proposed algorithm**
>
> Our theoretical results in Theorems 1 and 2 suggest that Langevin dynamics could take an exponentially growing number of iterations to draw samples from low-variance modes in a mixture distribution. This is intuitive because Langevin dynamics may need to explore the entire $\mathbb{R}^d$ space until approaching some low-variance modes. Therefore, we proposed Chained Langevin dynamics, which divides the sample into patches of constant size $Q\ll d$, thus the volume it needs to explore is reduced to $\mathbb{R}^Q$ for each patch, which allows faster convergence.
>
>
> **8-"Is the idea behind the chained algorithm based/influenced on previous works?"**
>
> To the best of our knowledge, a *sampling algorithm* with a Chained-LD-like structure has not been proposed in the existing literature. The closest algorithm we could find in the literature is “autoregressive diffusion models” which is cited and discussed in the paper. However, please note that the autoregressive diffusion algorithm in [3,4,5] is a *learning algorithm* and its goal is to learn a generative model from training data. On the other hand, Chained-LD is a *sampling algorithm* and not a learning algorithm as we discussed in the global response. We have added Remark 3 to discuss the related works.
>
>
>
> **9- Typo at line 170-171**
>
> We thank the reviewer for pointing it out, which we have corrected in the revision.
>
>
> [1] Max Welling and Yee W Teh. Bayesian learning via stochastic gradient langevin dynamics. In Proceedings of the 28th international conference on machine learning (ICML-11), pp. 681–688. Citeseer, 2011.
>
> [2] Yang Song and Stefano Ermon. Generative modeling by estimating gradients of the data distribution. Advances in neural information processing systems, 32, 2019.
>
> [3] Emiel Hoogeboom, Alexey A. Gritsenko, Jasmijn Bastings, Ben Poole, Rianne van den Berg, and Tim Salimans. Autoregressive diffusion models. In International Conference on Learning Representations, 2022
>
> [4] Tong Wu, Zhihao Fan, Xiao Liu, Hai-Tao Zheng, Yeyun Gong, Yelong Shen, Jian Jiao, Juntao Li, Zhongyu Wei, Jian Guo, Nan Duan, and Weizhu Chen. AR-diffusion: Auto-regressive diffusion model for text generation. In Thirty-seventh Conference on Neural Information Processing Systems, 2023
>
> [5] Kashif Rasul, Calvin Seward, Ingmar Schuster, and Roland Vollgraf. Autoregressive denoising diffusion models for multivariate probabilistic time series forecasting. In International Conference on Machine Learning, pp. 8857–8868. PMLR, 2021.

---

### Official Review · Reviewer_nwd8 · 2024-11-03

**Soundness:** 3
**Presentation:** 2
**Contribution:** 2
**Rating:** 5
**Confidence:** 4

**Summary:**

The paper coniders Langevin Dynamics for multimodal distributions (actually mixtures of gaussians, with a small extension to the case of of a large-variance gaussian plus some localized subgaussians). They prove in some toy model cases that Langevin Dynamics does not reach good approximation to these distributions in short time. As a "solution" to that problem, they lower the dimensionality by partitioning the data vector in equal pieces and sequentially generating the pieces rather than generating them all together. This improves on the "curse of dimension" part of the difficulty.

**Strengths:**

The topic of the paper (rate of approximation of multimodal distributions) is of interest and the problem of finding a good scalable strategy in these problems is largely open.

**Weaknesses:**

The setting of study is mainly a toy model, whose assumptions may or may not hold in realistic scenarios.
A discussion of limitations of the core/crucial Assumptions 1 and 2 of the paper is dearly missed.

I think that this is a good step in that direction, but I'm not sure if the paper is already valuable as is, mainly due to the doubts I have as to how general Assumptions 1 and 2 actually are, compared to typical applications.

**Questions:**

1) Why is the setup of Assumption 1 general enough to deserve a paper?

More precisely, can one say that there are other kinds of multi-modality that appear in applications or not?
If yes, then what are the cases not covered by Assumption 1?
If not, in what sense does Assumption 1 subsume the main difficulties related to multi-modality?

2) Is it correct to say that the underlying principle of Assumption 2 is not that distinct from the case of Assumption 1, other than allowing the "concentrated/independent modes" part of the distribution to be given by more general kinds of measures?

---

> ### Author Response · Authors · 2024-11-25
> **Response to Reviewer nwd8**
>
> We thank Reviewer nwd8 for his/her time and thoughtful feedback on our work. Below is our response to the questions and comments in the review.
>
>
> **1- Justification for Assumptions**
>
> We would like to clarify that our work is a theoretical study of the mode-seeking properties of Langevin dynamics (LD) as a sampling algorithm from a known distribution. In Theorems 1 and 2, our goal is to provide a theoretical setting where we can prove the possible exponentially growing number of iterations for LD to capture all the modes. We prove this result under mixtures of Gaussian and sub-Gaussian, where the latter case applies to random vectors with a bounded support set.
>
> To the best of our knowledge, the existing theoretical results on LD prove satisfactory convergence of the dynamics in sampling from unimodal distributions. This differentiates our work from the existing literature and highlights the setting studied in our work where LD has a *provably* slow convergence. Since our theoretical results show the computational hardness of Langevin dynamics, they motivate finding a sampling algorithm with a lower iteration complexity. We therefore proposed Chained-LD to achieve this goal.
>
>
> **2- Comparison between Assumption 2 and Assumption 1**
>
> Note that Assumption 2 relaxes the Gaussian mode condition in Assumption 1 to a sub-Gaussian mode. This condition is significantly more general which generalizes the theoretical result to more general cases, e.g., where the distribution mode has a bounded support set (known to satisfy the sub-Gaussian condition). As we explained in Item 1 of our response, we do not know any existing computational hardness results for the discretized Langevin dynamic algorithm, which highlights the paper’s theoretical results in Theorems 1 and 2 in revealing settings where standard Langevin dynamics suffers from **provably** slow convergence.

---

> > ### Comment · Reviewer_nwd8 · 2024-11-25
> >
> > Thank you for the reply, I'll try to clarify what about assumptions 1 and its generalization assumption 2 is unsatisfactory to me.
> >
> > Mainly your discussion following assumption 1 says that this represents a situation in which:
> >
> > "the dynamics can only visit the universal mode unless the stochastic noise miraculously leads it to the
> > region of another mode" ( * )
> >
> > What the quoted sentence says sounds reasonable, but it seems to describe a much more general situation than what the specifics of Assumption 1 or Assumption 2 describe. In particular, Assumption 1 (or 2) are the case that there is a 'spread' mode and many 'concentrated modes'.
> >
> > On the other hand the quoted text ( * ) can cover many more situations involving this masking. In what sense is this sentence ( * ) "universally represented" by restricting to the case of assumption 1 (or 2)? Or why should we think that the 'gist' of this masking phenomenon is covered by Assumption 1 (or 2)? Why do we exclude the case of a mixture of more than one "spread" subgaussian distribution, together with some intermediate spreading, together with some concentrated ones?
> >
> > Can you show any evidence that more general cases are somehow subsumed by these assumptions?
> >
> > To say it another way, I believe that there are a lot of real life situations in which Assumption 1 does not hold, and I don't follow why the underlying principle of these assumptions would cover realistic situations. Lacking such justification, it would be plausible that this negative result would not have bearing on actual real cases. Can you prove me wrong? I believe that you may be able to, but I don't see that in the current version of the paper.
> >
> > This is also why I talk about toy model situations, because I don't see how Assumption 1 or Assumption 2 are general enough to cover more general cases, and the experiments directly follow those assumptions.
> >
> > I think that some discussion about how "assumption 1 or 2 actually extend to more general cases", and some experiments beyond the case of those assumptions, i.e. something more realistic than "1 spread mode and many concentrated ones", showing that something similar to the proved slow convergence still holds beyond the assumptions 1 + 2, would allow me to raise my score.

---

> > > ### Author Response · Authors · 2024-11-27
> > >
> > > We thank Reviewer nwd8 for his/her feedback on our response. Below, we provide our intuition for why the theoretical setting in our study can reveal a fairly general mode-seeking tendency of Langevin dynamics when sampling from mixture distributions.
> > >
> > > Here, we discuss a special case of Theorem 2 where the low-variance modes have a **bounded support set**. Therefore, consider the following mixture distribution $P$
> > >
> > > $$P(x) = w_0 P^{(0)}(x) + \sum_{i=1}^k w_i P^{(i)}(x)$$
> > >
> > > in which every mode $P^{(i)}$ (for $i\ge 1$) has a bounded support set, i.e., $P^{(i)}(x) = 0 $ for every $\Vert x - \mu_i \Vert > r$. In other words, $P^{(i)}$ has **zero mass** outside a distance $r$ from its mean $\mu_i$. Note that this assumption *does not impose any restriction* on the total mass of mode $P^{(i)}$, i.e., $w_i$ can take any value in $(0,1)$.
> > >
> > > In the above setting, if the total mass of $P^{(0)}$ (denoted by $w_0$) is negligible, we expect an efficient sampling algorithm to ignore $P^{(0)}$ in the sampling process and focus on the larger-mass modes $P^{(1)},\ldots , P^{(k)}$. However, the Langevin dynamics (LD) performs a **noisy local search** by following the noisy gradient of the log-density $\log P(x)$. Observe that such a noisy local search will completely disregard the modes $P^{(1)},\ldots , P^{(k)}$ *due to their bounded support sets*, unless the sampling variable $x$ is within distance $r$ from one of the mean vectors $\mu_1,\ldots , \mu_k$. On the other hand, the total volume of the support sets of $P^{(1)},\ldots , P^{(k)}$ will be at most $\mathcal{O}(k\cdot r^d)$, implying the exponential iteration complexity of LD in sampling from the larger-mass modes $P^{(1)},\ldots , P^{(k)}$, as long as $r$ is a constant factor smaller than the radius of $P^{(0)}$’s support set.
> > >
> > > The above scenario explains our intuition behind Theorems 1 and 2, where we prove the explained phenomenon will hold even more generally if the support set of modes $P^{(1)},\ldots , P^{(k)}$ has a bounded tail probability vanishing at a sub-Gaussian rate. Note that the discussed scenario already applies to a wide range of mixture distributions whose modes have bounded support sets, where the boundedness assumption only requires the radius of the support set to be a constant factor smaller than the radius of in-between mode $P^{(0)}$.
> > >
> > >
> > > Finally, regarding the reviewer’s question “**Why do we exclude the case of a mixture of more than one "spread" subGaussian distribution**”, we believe the same sampling challenge from the concentrated modes still persists as long as the PDF of spread modes would contain a negligible probability mass in the support sets of the concentrated modes, because Langevin dynamics receive zero guidance from the concentrated modes when they are outside their support sets. However, the theoretical proof for such a case would be extremely challenging because we need to keep track of the distribution of the addition of several Gaussian noise vectors present in LD to a non-Gaussian original distribution, which would lead to a non-Gaussian distribution with an analytically-complex PDF.
> > >
> > > To sum up, as far as we know, our results are the first to **formally prove** the exponentially growing iteration complexity of Langevin dynamics for bounded-support-set concentrated modes and a Gaussian spread mode.

---

### Official Review · Reviewer_7ZLa · 2024-11-03

**Soundness:** 3
**Presentation:** 3
**Contribution:** 3
**Rating:** 5
**Confidence:** 4

**Summary:**

This article examines the behavior of Langevin dynamics (LD), a sampling method commonly used in generative modeling, especially in contexts where data distributions are multimodal (contain multiple clusters or “modes”). The authors highlight a key limitation of traditional Langevin dynamics: its difficulty in sampling from all modes in a multimodal distribution, as it often struggles to transition between modes in high-dimensional settings.

To address this, the authors propose Chained Langevin Dynamics (Chained-LD), a novel approach that divides the data into smaller patches and samples these sequentially, conditioned on previously generated patches. This method reduces the likelihood of mode-seeking (focusing on only a few modes) by lowering the dimensionality for each sampling step, which in turn accelerates convergence to the correct distribution.

Key contributions and findings of the paper include:
1. **Theoretical Analysis:** The study provides a rigorous analysis of examples where the traditional Langevin dynamics can fail to capture all modes, especially under Gaussian and sub-Gaussian mixtures.
2. **Chained Langevin Dynamics:** The paper introduces Chained-LD as a solution and establishes its convergence properties, showing that it can sample all modes in a distribution.
3. **Empirical Validation:** Experiments on synthetic datasets and real image datasets (such as MNIST) successfully conducted showing how Chained-LD can capture all modes compared to standard LD.

**Strengths:**

The paper is well-written, easy to read, and highly structured, with thorough documentation that enhances its clarity. The computational approach is straightforward to follow, and the main message of the work is communicated effectively.

One of the most compelling aspects of the paper is the recursion on the vector $n_t$, which presents an innovative approach to the field; to the best of my knowledge, this recursion is novel and has significant implications on the examples presented. The test setting introduced is particularly interesting, as it highlights how SGMs tend to emphasize majority classes, potentially overlooking minority representations in the data.

Additionally, the authors provide a detailed description of the behavior within a specific test setting, which is later extended to the framework of annealed Langevin dynamics. This progression in the analysis is logical and well-integrated, offering valuable insights into the dynamics of SGMs in multimodal distributions.

**Weaknesses:**

1. **Missing Citations:** The paper does not reference the recent works, like *Conforti, Dumus, and Gentiloni-Silveri (2023)*, which provides critical insights into the KL divergence bounds. Given the significance of these results for KL bounds, integrating this reference would strengthen the theoretical grounding of the paper.
2. **Real-World Applicability and Initial Distribution:** The results are derived based on an initial distribution that is not known in advance, raising questions about their direct applicability to real-world scenarios. In practical applications, data is typically standardized and rescaled, and initializations do not start from a well-defined mode $P^{(0)}$. It would be helpful to understand if and how these findings could be adapted to scenarios where the data distribution and modes are not known beforehand. Moreover, I suggest to include that a direct application of these findings would be finding fair algorithms on less represented group. Could the authors discuss potential approaches for adapting their method to scenarios where the initial distribution is unknown or to explain any limitations in applying their results to such cases?
3. **Handling of Balanced Modes:** The study assumes a setting where one mode is notably dominant. In reality, distributions often lack such stark imbalances. How would the results change when no mode significantly outweighs others? An exploration of how the method performs when modes have similar probabilities would clarify its robustness and generalizability. Could the authors include an additional experiments or theoretical analysis for the case where modes have similar probabilities? This would help demonstrate the method's robustness across different types of multimodal distributions.
4. **Assumption 2 and Gaussian Mixtures:** On page 5, it is claimed that Assumption 2 is automatically satisfied for Gaussian mixtures, yet this is not elaborated upon. This assumption, particularly in points (iv) and (v), introduces technical constraints that lack intuitive explanations for real-world distributions. A more detailed justification for these points would enhance clarity. Could the authors provide a brief proof or explanation for why Gaussian mixtures satisfy Assumption 2, particularly focusing on points (iv) and (v)?
5. **Selection of Patch Size $Q$:** The algorithm’s effectiveness relies on the choice of patch size $Q$. However, guidance on how $Q$ should be selected w.r.t. the dimension $d$ for different applications is not provided. Insights into an optimal or adaptive selection process of this parameter could improve the method’s usability in diverse contexts. Could the authors provide a heuristic or guideline for selecting $Q$ based on d and the characteristics of the data, or to discuss the sensitivity of their method to different choices of $Q$?
6. **Score Function Estimation:** It is unclear how $s_\theta$, the conditional score function estimator, is obtained and whether it is computed offline or dynamically updated. An explanation of the estimation method and its computational cost would help assess the algorithm’s practicality. Could the authors give a brief description of the estimation procedure for $s_\theta$ in the main text or appendix, including whether it's pre-computed or updated during sampling?
7. **Impact on Input Geometry:** The segmentation of the sample into patches for processing may alter the overall geometry of the input vector $x_0$. How does this segmentation affect the coherence and structure of the original data, and does it risk distorting key relationships within the vector? Could the authors discuss or analyze how their patch-based approach preserves or affects important structural relationships in the data?
8. **Proposition 1 and Convergence Bounds:** On page 7, Proposition 1 references a TV distance condition bounded by $\varepsilon Q/d$ but does not justify why this specific bound is chosen. Furthermore, the criteria for calling this result a “convergence bound” are unclear and would benefit from further clarification on its theoretical significance. Could the authors share intuition or justification for the specific bound $\varepsilon Q/d$ and clarify what it is meant by "convergence bound" in this context?
9. **Computational Efficiency:** The proposed approach, particularly with Chained Langevin Dynamics, appears more computationally intensive than traditional methods. Can the authors quantify the additional computational load required for convergence in comparison to previous approaches? Understanding the trade-offs between computational cost and performance is essential for evaluating the practical feasibility of this algorithm. Could the authors include a computational complexity analysis or empirical runtime comparison between their method and traditional Langevin dynamics?

I’m open to improving the evaluation score, as this paper shows real potential in highlighting biases in SGMs, especially around minority representation. Addressing the weaknesses—such as clarifying assumptions, real-world applicability, and computational efficiency—would greatly enhance its impact and make it a valuable reference for understanding and mitigating biases in generative models.

**Questions:**

See the **Weaknesses** section.

---

> ### Author Response · Authors · 2024-11-25
> **Response to Reviewer 7ZLa (Part 1)**
>
> We thank Reviewer 7ZLa for his/her time and thoughtful feedback on our work. Below is our response to the questions and comments in the review.
>
>
> **1- Related Reference**
>
> We thank Reviewer 7ZLa for pointing out the related work on the KL convergence guarantees of score-based diffusion models. We added a discussion about this paper in the revised paper.
>
>
> **2- Real-World Applicability and Initial Distribution**
>
> In scenarios where the underlying target distribution is unknown, as illustrated in Appendix D.2 in the revision, we can train an estimator to approximate the score function from the training dataset. To test the mode-seeking tendencies of the sampling algorithms, in our experiments, we initialize the samples from one mixture component and examine whether they can mix with other modes within a limited number of iterations. The experimental results indicate that Chained-LD is able to mix between modes even if it is initialized inside a mode. Therefore, we believe that Chained-LD is also capable of outputting samples from underrepresented groups.
>
>
> **3- Handling of Balanced Modes**
>
> We are unsure if this comment applies to our results, because we **have not assumed** that the in-between mode $P^{(0)}$ has a higher probability mass (parameter $\omega_0$ in the mixture representation) than the other modes. In fact, our theoretical results on the iteration complexity of Langevin Dynamics (LD)  remain true even if the probability mass of in-between mode $P^{(0)}$ is small. Despite a minor mass, the higher variance of $P^{(0)}$ can still mislead the LD in the in-between space, due to the weaker influence of the other modes with lower variance but higher probability.
>
>
> **4- Assumption 2 and Gaussian Mixtures**
>
> We would like to clarify that Assumptions 2.(i)-(ii) automatically hold for Gaussian component $P^{(i)} = \mathcal{N}(\mu_i, \sigma_i^2 I)$. Assumption 2.(iii) is also true since the score function of Gaussian distribution $\nabla \log P^{(i)}(x) = -\frac{x}{\nu_i^2}$ is $\frac{1}{\nu_i^2}$-Lipschitz. Assumptions 2.(iv) and (v) are specific assumptions on the mean and variance of $P^{(i)}$, which is similar to Assumption 1 for our Gaussian mixture setting. We have made this point more clear in the revision.
>
>
> **5- Selection of Patch Size $Q$**
>
> In Proposition 1, our theoretical results suggest that the iteration complexity of Chained-LD scales as $\mathcal{O} \left( \exp(\mathcal{O}(LQ)) \frac{d^3}{\varepsilon^2\sqrt{Q}} \log \frac{d^2\sqrt{Q}}{\varepsilon^2} \right)$, which takes its minimum value when the patch size $Q$ is a constant not growing with $d$. In practice, the numerical results are insensitive to moderate values of constant $Q$, while for overly large $Q$, Chained-LD has mode-seeking tendencies similar to LD. We added Figures 7-9 in our revision, which demonstrates the effect of $Q$ in the synthetic Gaussian mixture case. We can observe that for dimension $d=100$, $Q \in \{1,4,10\}$ has similar performance, $Q=20$ needs more steps to find the other two modes, $Q=50$ almost cannot find other modes, and $Q=100$ reduces to vanilla Langevin dynamics.
>
>
> **6- Score Function Estimation**
>
> We would like to clarify that the score estimator is computed before running Chaine-LD, and in our numerical analysis, Chained-LD applies to a fixed score estimator. More precisely, we follow reference [1] which provides an algorithm for estimating the score function. More details can be found in Appendix D.2 of the revised draft.
>
> **7- Impact on Input Geometry**
>
> We believe the structural relationships in the data are maintained in Chained-LD due to the conditional distribution $P(x^{(q)}|x^{(1)}, \cdots, x^{(q-1)})$. Although the sample is divided into patches, the structural information in the previous patches can be reflected by the conditional distribution, therefore Chained-LD faithfully models the target distribution according to the chain rule $P(x) = \prod_{q=1}^{d/Q} P(x^{(q)}|x^{(1)}, \cdots, x^{(q-1)})$.
>
>
> **8- Proposition 1 and Convergence Bounds**
>
> We note that Proposition 1 is obtained from a divide-and-conquer method. Since the sample is divided into $d/Q$ patches, to achieve $\varepsilon$ total variation error for the sample, each patch achieves $\frac{\varepsilon}{d/Q} = \varepsilon Q/d$ total variation error in expectation.

---

> ### Author Response · Authors · 2024-11-25
> **Response to Reviewer 7ZLa (Part 2)**
>
> **9- Computational Efficiency**
>
> We would like to highlight that the time complexity of chained Langevin dynamics was significantly lower than vanilla/annealed Langevin dynamics in our experiments. Our comparison between Chained-LD and vanilla/annealed Langevin dynamics in the paper is based on **iteration complexity**, i.e., the number of queries to the score function $\nabla \log P(x^{(q)}|x^{(1)}, \cdots, x^{(q-1)})$ or $\nabla \log P(x)$. However, since Chained-LD only updates one patch at every iteration while vanilla/annealed LD updates the whole image, Chained-LD’s time complexity is significantly lower than vanilla/annealed Langevin dynamics. We have explained this point in Appendix D in the revision.
>
>
> [1] Yang Song and Stefano Ermon. Generative modeling by estimating gradients of the data distribution. Advances in neural information processing systems, 32, 2019.

---

### Official Review · Reviewer_Egfk · 2024-11-03

**Soundness:** 3
**Presentation:** 2
**Contribution:** 3
**Rating:** 5
**Confidence:** 3

**Summary:**

This paper investigates the mode-seeking tendency of stochastic gradient Langevin dynamics (SGLD) and proposes Chained Langevin dynamics with the objective of sampling from multi-modal distributions more accurately. The paper first studies the convergence of SGLD on a multi-modal target with a high-variance, low density, background mode, and shows hardness results of sampling from the sub Gaussian mixture in sub-exponential time in the data dimension. Chained Langevin dynamics seek to alleviate this issue by dividing the input vector into patches and applying SGLD to the conditional distributions $p(x^q \vert x^1, \dots, x^{q-1})$, only, thus reducing the effective dimension of the sampled variable. The proposed chained Langevin dynamics are then combined with annealing schemes and used to train a score-based generative model on toy-datasets and MNIST image datasets.

**Strengths:**

- The paper makes theoretical contributions on the mode seeking properties of Langevin dynamics with and without annealing and makes methodological contributions to improve SGLD.
- The paper investigates an interesting, and to me novel test-case of having a high-variance low-density background mode. Such settings could be useful benchmarks for sampling and generative modelling, particularly for approaches that have scientific applications in mind.
- The chained Langevin diffusion is a simple, easy to implement concept with potentially high impact in sampling and score-based generative modelling.
- To me it is quite impressive that the LSTM architecture (somewhat untypical for diffusion models) achieves the translation from initial samples to target samples on the image dataset example. However, I don't fully understand how this works, since annealed Langevin is normally initialised from a Gaussian distribution. Some explanations on how the generative model is applied, here, would be helpful.

**Weaknesses:**

- To me, the paper appears to be two different papers in one. It is hard to make out a story and it is unclear whether the paper attempts to make contributions in Langevin sampling or score-based generative modelling. I think the paper should make a clearer distinction between sampling from a known target distribution (for which we know the Stein score but don't have samples) and generative modelling (for which we have access to samples but not the Stein score). These two settings are quite different, even though they are mathematically related. This also influences how the theoretical and empirical contributions are perceived as a reader.
- The theory is supposed to motivate the chained Langevin algorithm. However, I see a gap between the motivating failure case of SGLD and the proposed chained LD, at least in the main text of the paper. Firstly, chained LD is introduced in Algorithm 1 as a training algorithm of a generative model trained on data. The theoretical results, on the other hand, regard a sampler from a fixed target distribution. However, the paper does not explain why vanilla score-based generative models would fail on mixtures of sub-Gaussian distributions in the same way as SGLD, and thus why chained LD is needed for the training of the generative model. Secondly, the theory of chained LD is not fully developed, and I guess the mixture of sub-Gaussians would not satisfy the implied assumptions of m-strongly log concavity in Proposition 1.
- If the papers main focus is a new sampling methodology, I think it would be good to justify chained LD more thoroughly from a theory point of view, or to make more thorough evaluations of the sampler, comparing it to other adaptions of Langevin sampling and annealing methods. For example, can something be said about the optimal number of patches between $Q=1$ and $Q=d$, or is this a tuning parameter? I am also curious if related strategies to chained Langevin have been developed before. It seems a reasonable approach, as it looks like a mixture of an autoregressive sampler and a Langevin sampler. Putting chained Langevin into context with related works on other variations of vanilla Langevin would be helpful.
- If the paper proposes a new method for score-based generative modelling, it would be interesting to see how it compares to vanilla diffusion models on standard benchmarks. For example, reporting FID scores of annealed chained LD on standard image datasets like CIFAR10 would typically be expected for a paper at ICLR, even if the scores are not state of the art. Of course it is hard to make improvements, here, since score-based generative models are so optimised, but I could see how slicing the sample into chunks could speed up convergence in some way or another.

**Questions:**

- I am not familiar with the recent literature on convergence results for Langevin dynamics. I would expect that the difficulty of sampling from multi-modal distributions with unadjusted Langevin algorithms is well-known. While I consider the background noise an insightful novel test-case, how are the theoretical contributions in this paper on the convergence of Langevin algorithms different from existing results on unadjusted Langevin algorithms?
- What is the trade-off that one needs to make when choosing the number of patches for chained Langevin? Is there an optimal patch number between $Q=1$ and $Q=d$?
- For the score-based generative model, could you clarify what forward process you used? In diffusion models, one typically picks a forward process that converges to a Gaussian distribution and learns the reverse process from Gaussian samples to data. How did you initialise the reverse diffusion in the image translation experiments show-cased in figure 3, where the diffusion is seemingly not initialised from a Gaussian?

---

> ### Author Response · Authors · 2024-11-25
> **Response to Reviewer Egfk (Part 1)**
>
> We thank Reviewer Egfk for his/her time and thoughtful feedback on our work. Below is our response to the questions and comments in the review.
>
>
> **1- Connection between hardness results and Chained-LD sampling algorithm**
>
> We believe that the two parts of our paper, the first on hardness results on iteration complexity of Langevin dynamics (LD) and the second on Chained-LD, are strongly connected. Note that the theoretical results in Theorems 1 and 2 show that LD can have an exponentially growing iteration complexity in dimension $d$ for the characterized mixture distribution. The complexity result implies that if the LD sampler could draw the coordinates sequentially, the exponential iteration complexity would be addressed. A natural way to implement this idea is to apply the Chain Rule in probability:
>
> $$ P([x_1,\ldots , x_d])  = P(x_1)\cdot P(x_2|x_1)\cdots P(x_d|x_1,\ldots x_{d-1})$$
>
> Therefore, we propose Chained-LD to separate the sampling process across the $d$ coordinates and avoid the exponentially growing complexity of LD shown in Theorems 1 and 2. As we explained, the two parts of the paper are highly connected, and Chained-LD would not be motivated without showing the exponentially growing iteration complexity of LD in sampling from mixture distributions.
>
>
> **2- "I think the paper should make a clearer distinction between sampling from a known target distribution and generative modeling."**
>
> As we have explained in the global response, our work focuses on sampling from a known distribution using variants of Langevin dynamics. To address the reviewer’s comment, we have clarified in the introduction that our paper studies the sampling task exclusively and does not focus on generative modeling. We have relocated all the citations for related references on generative modeling to the related work section, and the revised introduction solely focuses on the sampling task addressed by LD and Chained-LD algorithms. We have also included Remarks 1 and 2 to clarify the sampling nature of our proposed Chained-LD algorithm.
>
>
> **3- “Firstly, chained LD is introduced in Algorithm 1 as a training algorithm of a generative model trained on data.”**
>
> We think this point is related to the reviewer’s previous comment. We believe the comment is due to a misinterpretation of Chained-LD as a learning algorithm. Note that the Chained-LD in Algorithm 1 is not a learning algorithm to learn a model from training samples. As we also discussed in the global response, Chained-LD is a sampling algorithm from an already known distribution $P$, which aims to address the exponentially growing iteration complexity of regular Langevin dynamics in sampling from mixture distributions (Theorems 1 and 2). We have clarified the sampling nature of Algorithm 1 in Remark 2 in the revised text.
>
>
> **4- “However, the paper does not explain why vanilla score-based generative models would fail on mixtures of sub-Gaussian distributions in the same way as SGLD”**
>
> As we responded to the previous points, Chained-LD is not a learning algorithm and it is only a sampling method similar to standard Langevin dynamics. Therefore, we did not compare it with generative modeling methods that learn the distribution from data. We hope that this clarification helps with the reviewer’s comment.
>
>
> **5- Log-concavity assumption in Proposition 1**
>
> We would like to clarify that as suggested by the reference [1], Proposition 1 only requires $P$ to be $m$-strongly log-concave **outside a region of radius $R$**, i.e., for $||\mathbf{x}^{(q)}|| > R$. Therefore, sub-Gaussian mixtures satisfying Assumption 2 would also satisfy this assumption for $R \gtrsim \max_i ||\mu_0-\mu_i||$. We have made this point clear in the revision.
>
>
> **6- Selection of patch size $Q$**
>
> In Proposition 1, our theoretical results suggest that the iteration complexity of Chained-LD scales as $\mathcal{O} \left( \exp(\mathcal{O}(LQ)) \frac{d^3}{\varepsilon^2\sqrt{Q}} \log \frac{d^2\sqrt{Q}}{\varepsilon^2} \right)$, which takes its minimum value when the patch size $Q$ is a constant not growing with $d$. In practice, the numerical results are insensitive to moderate values of constant $Q$, while for overly large $Q$, Chained-LD has mode-seeking tendencies similar to LD. We added Figures 7-9 in our revision, which demonstrates the effect of $Q$ in the synthetic Gaussian mixture case. We can observe that for dimension $d=100$, $Q \in \{1,4,10\}$ has similar performance, $Q=20$ needs more steps to find the other two modes, $Q=50$ almost cannot find other modes, and $Q=100$ reduces to vanilla Langevin dynamics.

---

> ### Author Response · Authors · 2024-11-25
> **Response to Reviewer Egfk (Part 2)**
>
> **7- Related works on autoregressive diffusion models**
>
> To the best of our knowledge, a *sampling algorithm* with a Chained-LD-like structure has not been proposed in the existing literature. The closest algorithm we could find in the literature is “autoregressive diffusion models” which is cited and discussed in the paper. However, please note that the autoregressive diffusion algorithm in [2,3,4] is a *learning algorithm* and its goal is to learn a generative model from training data. On the other hand, Chained-LD is a *sampling algorithm* and not a learning algorithm as we discussed in the global response. We have added Remark 3 to discuss the related works.
>
>
> **8- Comparison of our theoretical contributions with the existing literature**
>
> In this work, we study sub-Gaussian mixtures with an in-between distribution component $P^{(0)}$, which we have not seen in the existing literature. We prove that under such a mixture model, Langevin dynamics tends to be mode-seeking and may only find a subset of modes. To the best of our knowledge, mode-seeking properties of diffusion models under imbalanced modes have not been investigated in the literature. We also note that our analysis can be extended to annealed Langevin dynamics with bounded noise levels as we showed in Theorems 3 and 4 in Appendix B, indicating the effect of annealing noise levels on the mode-seeking tendencies of Langevin dynamics.
>
>
> **9- Forward process of Langevin dynamics**
>
> As discussed in [5], Langevin dynamics is the discretization of variance-exploding SDE, so the forward process converges to a Gaussian distribution with infinity variance, which is infeasible in practice. On the other hand, [6] suggests that Langevin dynamics asymptotically mix between modes and converge to the target distribution, regardless of the initialization. In our experiments, as we aim to study the mode-seeking tendencies of Langevin dynamics, we initialize the samples from one mixture component and examine whether they can mix with other modes within a limited number of iterations.
>
>
> [1] Yi-An Ma, Yuansi Chen, Chi Jin, Nicolas Flammarion, and Michael I Jordan. Sampling can be faster than optimization. Proceedings of the National Academy of Sciences, 116(42):20881–20885, 2019.
>
> [2] Emiel Hoogeboom, Alexey A. Gritsenko, Jasmijn Bastings, Ben Poole, Rianne van den Berg, and Tim Salimans. Autoregressive diffusion models. In International Conference on Learning Representations, 2022
>
> [3] Tong Wu, Zhihao Fan, Xiao Liu, Hai-Tao Zheng, Yeyun Gong, Yelong Shen, Jian Jiao, Juntao Li, Zhongyu Wei, Jian Guo, Nan Duan, and Weizhu Chen. AR-diffusion: Auto-regressive diffusion model for text generation. In Thirty-seventh Conference on Neural Information Processing Systems, 2023
>
> [4] Kashif Rasul, Calvin Seward, Ingmar Schuster, and Roland Vollgraf. Autoregressive denoising diffusion models for multivariate probabilistic time series forecasting. In International Conference on Machine Learning, pp. 8857–8868. PMLR, 2021.
>
> [5] Yang Song, Jascha Sohl-Dickstein, Diederik P Kingma, Abhishek Kumar, Stefano Ermon, and Ben Poole. Score-based generative modeling through stochastic differential equations. In International Conference on Learning Representations, 2020c.
>
> [6] Max Welling and Yee W Teh. Bayesian learning via stochastic gradient langevin dynamics. In Proceedings of the 28th international conference on machine learning (ICML-11), pp. 681–688. Citeseer, 2011.

---

> > ### Comment · Reviewer_Egfk · 2024-11-25
> >
> > Thank you for the detailed response to my questions. I have a quick follow up question regarding the distinction between sampling and generative modelling in your work:
> >
> > **We believe the comment is due to a misinterpretation of Chained-LD as a learning algorithm. Note that the Chained-LD in Algorithm 1 is not a learning algorithm to learn a model from training samples.**
> >
> > Thank you for clarifying algorithm 1 and making it more general by replacing the noise conditional score network with the score $\nabla\log P_\sigma(\mathbf x)$. However, my concern is that the denoising score $\nabla\log P_\sigma$ is typically intractable when only the target distribution is known. This is the reason why standard Langevin does not use the denoising score to run the reverse diffusion, but the regular Stein score $\nabla \log P_0$ instead. I would expect that a sampling algorithm that uses $\nabla\log P_\sigma$ instead of $\nabla\log P_0$ would already improve standard Langevin significantly in theory and practice, even without chaining.
> >
> > Since  $\nabla\log P_\sigma$ is typically intractable even when P is known, you replace, as far as I understand, $\nabla\log P_\sigma$ with a neural surrogate $s_\theta$ such as a noise conditional score network. However, this denoising scores need to be learned *from data*. This, in my opinion, turns the original sampling algorithm into a sampling algorithm for diffusion-based generative models, since at some point the score network needs to be fitted. This is also the reason why I understood algorithm 1 as a training algorithm for generative models. Could you clarify the following questions?
> > - Are you proposing chained Langevin as a method to sample from pre-trained diffusion models?
> > - If not, can you implement chained LD even when you don't have access to data sampled from the target distribution? How do you estimate the denoising score $\nabla\log P_\sigma$ in these cases?

---

> ### Author Response · Authors · 2024-11-27
>
> We thank Reviewer Egfk for his/her time and feedback on our response. Regarding the raised points in the feedback,
>
> **1- "my concern is that the denoising score $\nabla \log P_\sigma (x)$ is typically intractable when only the target distribution is known"**
>
> First, we would like to clarify that in our experiments on the Gaussian mixture in Figure 2, we applied Algorithm 1 with $\sigma=0$, i.e. chained version of **vanilla** Langevin dynamics. Therefore, the improvement of Chained-LD over vanilla Langevin dynamics in Figure 2 does not require a choice of positive $\sigma>0$.
>
> Also, in the *specific case of Gaussian mixtures*, such as the experiment in Figure 2, the score function $\nabla \log P_\sigma (x)$ is indeed tractable for a general $\sigma\ge 0$. To compare the performance between $\sigma=0$ and non-zero $\sigma \le 1$, we have run the experiment of Figure 2 with both vanilla LD (where $\sigma=0$) and annealed LD with $\sigma_{\max}=1$, as well as the chained version of both the algorithms. As shown in Figure 10 in the revised Appendix, the non-chained versions of both LD and ALD (with $\sigma_{\max}=1$) did not find the missing Gaussian modes until iteration $10^6$. On the other hand, their chained versions, i.e. Chained-VLD and Chained-ALD, could find the missing modes in only 1000 iterations.
>
> As the experiments on the Gaussian mixture setting suggest, the gain by chaining Langevin dynamics is present under both zero and non-zero $\sigma$ parameters. We agree with the reviewer that the annealed LD with a general $\sigma>0$ can lead to an intractable score function in a general case; however, it does not affect our experiments on Gaussian mixtures where the score function under any $\sigma$ value can be efficiently computed.
>
> **2- "I would expect that a sampling algorithm that uses $\nabla \log P_\sigma$ instead of $\nabla \log P_0$ would already improve standard Langevin significantly in theory and practice, even without chaining."**
>
> As we explained in our response to Item 1, we ran the experiment on Gaussian mixtures and observed that the convergence speed gain by chaining Langevin dynamics is present under both zero and non-zero $\sigma$ parameter. Please refer to Figure 10 of the revised Appendix for the numerical results. Note that “vanilla” in the figure means $\sigma=0$.
>
>
> **3- "Are you proposing chained Langevin as a method to sample from pre-trained diffusion models?"**
>
> Please note that our work is a theoretical study of Langevin dynamics and their mode-seeking properties. Also, we would like to clarify that our proposed Chained-LD is intended to be a *counterpart* of the well-known Langevin dynamics. Therefore, as in practice, Langevin dynamics are not intended to be used for drawing samples from pre-trained models, we do not primarily intend to apply the Chained-LD algorithm to sample from pre-trained models.
>
> To be clear about our theoretical study, our main point is that the (vanilla) Langevin dynamics can *theoretically* lead to exponentially-growing iteration complexity to sample from mixture distributions, while the counterpart Chained-LD could circumvent the exponential complexity in dimension $d$ by chaining the sampling process using the Chain Rule.
>
>
> **4- "If not, can you implement chained LD even when you don't have access to data sampled from the target distribution?"**
>
> If we correctly understood the reviewer’s question, the reviewer asks whether we can use Chained-LD if we only have access to $\nabla \log P_0(x)$ (where $\sigma =0$). The answer is yes, because as we explained the chaining approach is applicable to vanilla Langevin dynamics. For example, in Figure 2, the "Chained-VLD" refers to the case where $\sigma=0$, which we refer to as “Vanilla Langevin dynamics”.

---

> > ### Comment · Reviewer_Egfk · 2024-12-01
> >
> > I thank the authors for their detailed responses and the additional experimental results/figures. I think the modifications improved the paper and made it clearer what algorithm 1 intends to do and how it ties in with the rest of the paper.
> >
> > I think the writing of this paper remains a weak point. To give an example: After reading the comments of other reviewers, it seems to me that I was not the only one who was confused about the combination of chained Langevin with annealing, and how the score function is estimated in practice. It seems to me that combining these two ideas without fully exploring chained vanilla Langevin first made the paper harder to follow. This problem also exists in the revision, where algorithm 1 requires some type of estimate for the score function of the annealed distribution.
> >
> > I think the paper could be improved by first discussing chained vanilla Langevin in theory and practice (thus also changing algorithm 1 to chained vanilla Langevin), and discussing the application to sample from noise conditional score networks only as a nice downstream task, once the main contribution of the paper has been made clear.
> >
> > I would like to retain my score.

---

### Author Response · Authors · 2024-11-25
**Global Response**

We thank the reviewers for their thoughtful feedback on our work.

In this global response, we would like to clarify on our proposed *sampling algorithm* Chained Langevin Dynamics (Chained-LD), because we think Reviewers Egfk and 2K9H have misinterpreted the Chained-LD method as a learning algorithm from training data. Regarding the comments in the reviews,

- We would like to clarify that the proposed Chained-LD is **not** a generative modeling algorithm to learn the distribution of training data, and it only aims to draw samples from a known distribution $P$ with score function $s(x) = \nabla \log P(x)$.

- The task handled by Chained-LD is **identical to** the sampling task of well-known Langevin dynamics (LD). Both algorithms assume access to distribution $p$ and aim to draw samples from a target distribution $P$. In Langevin dynamics, the sampler has access to the score function $\nabla \log P(x)$. In Theorems 1 and 2, we prove that the iteration complexity of Langevin dynamics to converge to an underlying mixture distribution can exponentially grow with the dimension of data $d$.

- Our results on the exponential growth of LD’s iteration complexity in dimension $d$ motivate the Chained-LD algorithm. Chained-LD follows a divide-and-conquer strategy which breaks the sampling task of the $d$-dimensional vector $x=[x_1,x_2,\ldots , x_d]$ into sampling each coordinate $x_i$ sequentially. The design of Chained-LD follows naturally from the chain rule in probability:
$$ P([x_1,\ldots , x_d])  = P(x_1)\cdot P(x_2|x_1)\cdots P(x_d|x_1,\ldots x_{d-1})$$

- As we explained above, there is no learning from data in the process of either Langevin dynamics or Chained-LD. Both the algorithms have complete knowledge of the probability model $P$, and aim to draw samples distributed according to $P$. The only difference is that Langevin dynamics performs the iterations for the entire vector $x$, while Chained-LD performs the sampling sequentially over coordinates according to the conditional distribution $P(x_k | x_1,\ldots , x_{k-1})$ given the previously drawn coordinates $x_1,\ldots, x_{k-1}$ prior to each iteration $k$.

- In the revision, we have relocated all the citations for related references on generative modeling to the related work section, and the revised introduction solely focuses on the sampling task addressed by LD and Chained-LD algorithms. We have also included Remarks 1 and 2 to clarify the sampling nature of our proposed Chained-LD algorithm.

---

### Meta-Review · Area_Chair_tK5q · 2024-12-20

**Metareview:**

The authors studied the property of Langevin dynamics being unable to explore multiple modes, and suggested a solution by sampling conditionally based on partitioning the dimension of the problem. This is an interesting idea, however, this requires sampling from the conditional distribution of one patch given previous sampled patches, and this distribution is often unknown. In general if we are given only an unnormalized density to sample from, I'm not sure how we will be able to compute the required condition densities.

Reviewer Egfk made another good point, because the authors then moved onto generative models given a dataset. I agree that this is not the same problem setting, as the goal of generate models is not to sample from a known unnormalized density. In fact, it's not clear at all if diffusion models suffer from the metastability issues as Langevin dynamics. Intuitively, the given samples would lead to the trained score function to identify the modes already.

All together, it's very confusing to interpret what the goal of this paper is supposed to be, as the two parts do not connect into one central theme. In combinations with the scores from the reviewers, I would have to recommend reject.

I would recommend the authors take a step back to think about the high level story of this paper, and choose to either work on the sampling from a known unnormalized density setting, or sampling as a generative model given a dataset setting. These are not directly related via Langevin dynamics.

**Additional Comments On Reviewer Discussion:**

The most helpful discussion was from Egfk, where the confusion was emphasized but not completely resolved in my opinion. Given that I had the same confusion when reading this paper, and following the discussion helped me confirm that I was not the only one feeling this way. This helped me confirm my decision.

---

### Decision · Program_Chairs · 2025-01-22

Reject